# Sp1-regulated expression of p11 contributes to motor neuron degeneration by membrane insertion of TASK1

Victoria García-Morales [1,2,5], Guillermo Rodríguez-Bey [1,5], Laura Gómez-Pérez [1,5], Germán Domínguez-Vías [1,2,5], David González-Forero[1,2], Federico Portillo[1,2], Antonio Campos-Caro [2,3], Ángela Gento-Caro[1,2], Noura Issaoui[1], Rosa M. Soler [4], Ana Garcera[4] & Bernardo Moreno-López [1,2]

Disruption in membrane excitability contributes to malfunction and differential vulnerability of specific neuronal subpopulations in a number of neurological diseases. The adaptor protein p11, and background potassium channel TASK1, have overlapping distributions in the CNS. Here, we report that the transcription factor Sp1 controls p11 expression, which impacts on excitability by hampering functional expression of TASK1. In the SOD1-G93A mouse model of ALS, Sp1-p11-TASK1 dysregulation contributes to increased excitability and vulnerability of motor neurons. Interference with either Sp1 or p11 is neuroprotective, delaying neuron loss and prolonging lifespan in this model. Nitrosative stress, a potential factor in human neurodegeneration, stimulated Sp1 expression and human p11 promoter activity, at least in part, through a Sp1-binding site. Disruption of Sp1 or p11 also has neuroprotective effects in a traumatic model of motor neuron degeneration. Together our work suggests the Sp1-p11-TASK1 pathway is a potential target for treatment of degeneration of motor neurons.

[1] Grupo de Neurodegeneración y Neurorreparación (GRUNEDERE), Área de Fisiología, Facultad de Medicina, Universidad de Cádiz, 11003 Cádiz, Spain. [2] Instituto de Investigación e Innovación Biomédica de Cádiz (INiBICA), Cádiz, Spain. [3] Unidad de Investigación, Hospital Universitario Puerta del Mar, Cádiz, Spain. [4] Unitat de Senyalització Neuronal, Dept. Medicina Experimental, Universitat de Lleida-IRBLLEIDA, Lleida, Spain. [5] These authors contributed equally: Victoria García-Morales, Guillermo Rodríguez-Bey, Laura Gómez-Pérez, Germán Domínguez-Vías. Correspondence and requests for materials should be addressed to B.M.-L. (email: bernardo.moreno@uca.es)

D espite their different clinical manifestations, neurodegenerative diseases share a multitude of common underlying mechanisms. Of special relevance is intrinsic membrane excitability (IME), determinant of neuronal responsiveness to incoming stimuli, which can condition specific neuronal subpopulations to become differentially vulnerable in neurodegenerative disorders[1–4].

Both passive and active membrane properties are involved in establishing neuronal IME. Passive properties, such as resting membrane potential (Vm) and membrane resistance, are determined by the density of background ion channels opened at rest, whereas active electrical signaling mainly relies on the activity of voltage-gated and ligand-gated ion channels. IME alterations interact with other pathogenic mechanisms, such as nitrosative stress (RNS) and excitotoxicity, that contribute to neuronal degeneration in many human neuropathologies[1–6]. For instance, RNS, a major factor mediating excitotoxic neurodegeneration[6–9], enhances IME by inhibiting background potassium ($K^+$) currents[10], thus depolarizing Vm and increasing membrane resistance. On the other hand, neuronal excitotoxicity arising from disruption of $Ca^{2+}$ homeostasis by overstimulation of glutamate receptors (GluRs)[7,11] might be amplified by an increased IME. Depolarization of Vm exacerbates $Ca^{2+}$ influx into the neuron by promoting voltage-sensitive $Ca^{2+}$ channels (VSCCs) opening and by relieving N-methyl-D-aspartate receptors from the magnesium blockage[7,12,13]. $Ca^{2+}$ overload can in turn trigger RNS, closing a vicious cycle with many still-unknown partners and leading to neuronal degeneration and cell death.

The adaptor protein p11 (S100A10) could be a firm candidate to fine-tune neuronal IME by regulating plasma membrane insertion of TASK1 (TWIK-related acid-sensitive $K^+$ subunit 1)[14], a member of the KCNK family of two-pore-domain $K^+$ channels. This family primarily determines passive membrane properties in mammalian cells, which confers it potential therapeutic impact[15,16]. Expression patterns of p11[17] and TASK1[18] predict their overlapping in many neuronal subpopulations vulnerable to certain neuropathological conditions. Particularly, somatic motor neurons (MNs), which express high levels of p11 and TASK1, are vulnerable to amyotrophic lateral sclerosis (ALS); a fatal neurodegenerative disease coursing with hyper-excitability as one of the earliest manifestations in sporadic and familial ALS patients[3,4]. Notably, enhanced IME, excitotoxicity, and RNS are common hallmarks in MN degeneration models with differing etiology, i.e. ALS[1–6] and traumatic injury of a motor nerve[6,19]. Therefore, we hypothesized that p11 and TASK1 are pivotal partners in a common mechanism determining vulnerability to neurodegeneration.

Our findings support that dysregulation of the ubiquitous transcription factor Sp1, p11, and TASK1 contributes to MN degeneration by impacting on MN IME and vulnerability.

## Results

**p11 determines MN IME by TASK1.** Co-immunoprecipitation studies supported a physical interaction between p11 and TASK1 in the nervous system (Fig. 1a). Subsequently, we examined by whole-cell patch-clamp the impact of p11–TASK1 on MNs IME (Fig. 1b). The strong reduction in $mRNA_{p11}$, resulting from the application of a small interfering RNA against $mRNA_{p11}$ ($siRNA_{p11}$), lowered IME of hypoglossal MNs (HMNs) from brainstem slices of rat pups (postnatal day 7; P7) (Fig. 1b–e). Whether $siRNA_{p11}$-related IME alterations were mediated by a regulatory effect on TASK channel function was addressed by modifying extracellular pH to characterize TASK-dependent changes in Vm and input resistance ($R_N$)[10]. Alkalination of extracellular medium (pH 8.2) from a physiological pH (7.2) opens TASK channels,

enhancing $K^+$ efflux, which leads to Vm hyperpolarization, and reduction of $R_N$ (Fig. 1f). On the contrary, an acidic solution (pH 6.2) retains $K^+$ into the cell by closing TASK channels, which causes Vm depolarization and $R_N$ increase (Fig. 1f). Therefore, the impact of pH on MN IME increases in parallel to the number of functional TASK channels at the plasma membrane. Suggestive of an increase in surface expression of TASK channels, $siRNA_{p11}$ strengthened pH impact on HMNs IME (Fig. 1g). In contrast, dexamethasone, a corticosteroid that stimulates p11 promoter activity, upregulated p11 at the hypoglossal nucleus (HN), increased IME, and reduced pH influence on HMNs IME (Supplementary Fig. 1).

To evaluate whether p11 stablishes IME through TASK1, further experiments were performed in primary cultures of embryonic spinal cord MNs (SMNs). In our experimental conditions, IME of wild-type SMNs ($SMNs^{wt}$) changed over time in culture (Fig. 1h). Thus, whereas Vm and $R_N$ remained steady throughout the first 3–4 days-in vitro (DIV), IME of $SMNs^{wt}$ progressively declined over the 4–6 DIV interval, and subsequently remained stable up to 8 DIV, the last time point tested (Fig. 1h). These outcomes are consistent with a progressive increase in surface expression of functional background $K^+$ channels at the 4–6 DIV interval. Therefore, next experiments were carried out in $SMN^{wt}$ at 3–4 DIV (Fig. 1i), a time window over which functional expression of TASK is still low. Remarkably, p11 knockdown increased TASK1-like immunofluorescence in the neurites of $SMNs^{wt}$ (Fig. 1i–k, Supplementary Fig. 2). Since $siRNA_{p11}$ did not upregulate TASK1 protein (Fig. 1i), our results support that p11 regulates cell distribution rather than whole levels of TASK1 expression. Conclusively, while decline in p11 did not affect SMNs isolated from $task1^{-/-}$ embryos ($SMNs^{task1-/-}$), it lowered IME and increased TASK-dependent changes in $SMNs^{wt}$ and $SMNs^{task3-/-}$ (lacking in TASK3, also highly expressed in MNs) (Fig. 1l, m). It was noteworthy that $SMNs^{wt}$ either untreated or treated with a non-targeting siRNA (cRNA) were almost unresponsive to pH variations despite TASK1 expression (Fig. 1i,m, Supplementary Fig. 2), suggesting a low surface expression of pH-modulated channels at 3–4 DIV. Altogether, these findings support that physical interaction between p11 and TASK1 prevents functional expression of the channel at the plasma membrane, which in turn controls MN IME.

**p11–TASK1 defines MN vulnerability to an excitotoxic insult.** GluR overstimulation induces intracellular $Ca^{2+}$ deregulation and $Ca^{2+}$ homeostasis disruption that finally leads to cell death[7,11]. To evoke this harmfulness mechanism, SMNs were exposed to a toxic dose of glutamate (Fig. 2a, Supplementary Fig. 3a, b). $Ca^{2+}$ imaging showed a glutamate-induced early rapid rise in intracellular $Ca^{2+}$ concentration ($[Ca^{2+}]_i$) in $SMNs^{wt}$ which peaked and then progressively dropped to a lower plateau (Fig. 2a, Supplementary Fig. 3a, b). This plateau was transient in $90.4 \pm 4.6\%$ of cases ($n = 164$ $SMNs^{wt}$). Following this initial plateau, $SMNs^{wt}$ generally experienced "$Ca^{2+}$ deregulation", a delayed and continued increase in $[Ca^{2+}]_i$ that reached a mostly sustained and irreversible plateau (Fig. 2a, Supplementary Fig. 3a, b), indicative of impending neuronal death[20,21]. To characterize $Ca^{2+}$ dynamics in SMNs, resting $[Ca^{2+}]_i$, peak $[Ca^{2+}]_i$ of the early rapid response to glutamate and time of $Ca^{2+}$ deregulation were measured (Supplementary Fig. 3b). $siRNA_{p11}$ strongly attenuated glutamate-evoked $Ca^{2+}$ dynamics (Fig. 2a, Supplementary Fig. 3c–e), although did not alter the fraction of SMNs that underwent $Ca^{2+}$ deregulation with regard to genotype (Supplementary Fig. 3f). Kaplan–Meier cumulative probability curves causally associated $siRNA_{p11}$

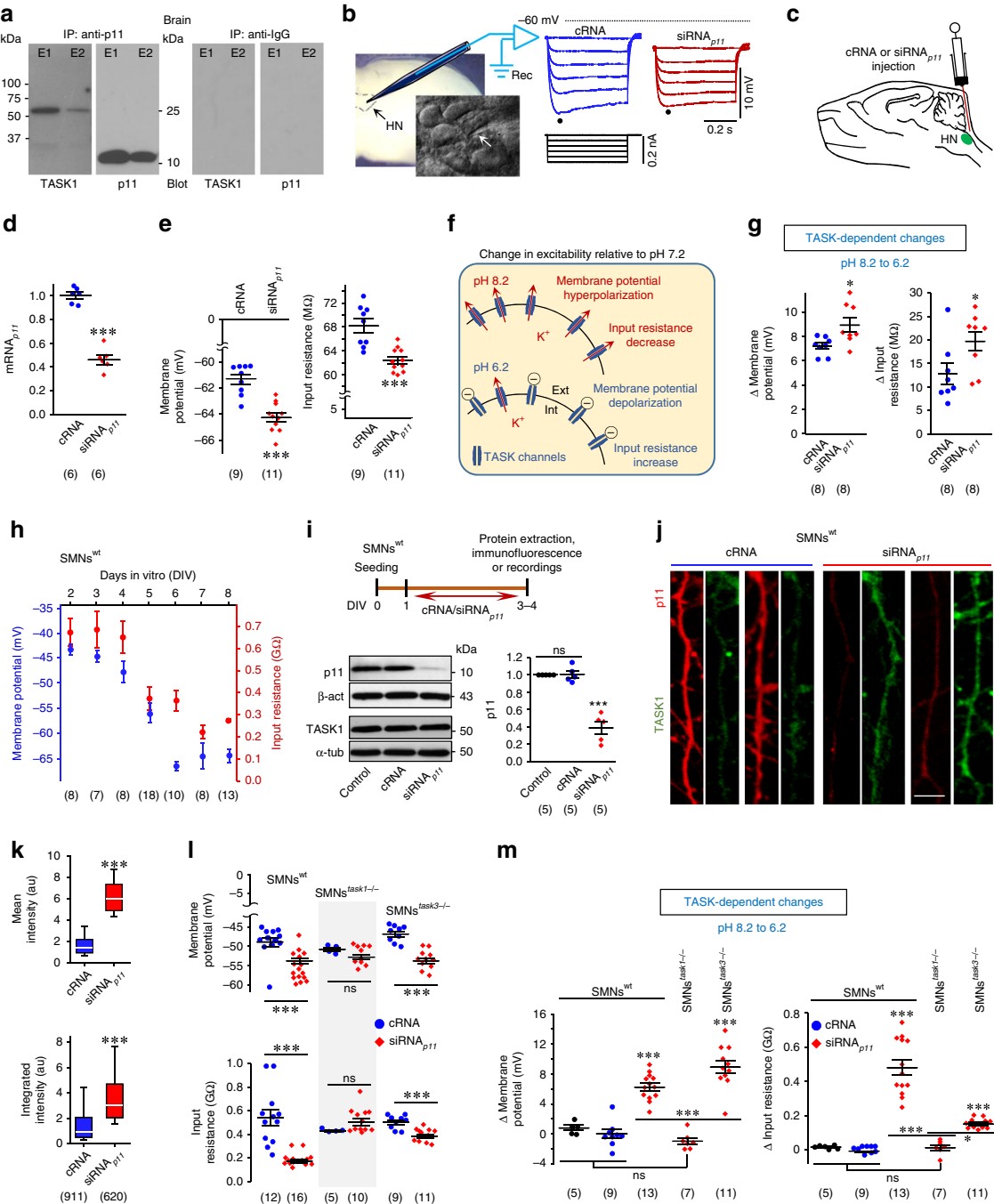

**Fig. 1** p11 controls MN IME through TASK1, but not TASK3 subunits. **a** Co-immunoprecipitation (IP) of p11 and TASK1 in protein fractions of homogenized brain from an adult mouse. An anti-sheep IgG was a control. Elution 1 (E1) and 2 (E2) are displayed. **b** Left, experimental model for whole-cell patch clamp recordings from HMNs in brainstem slices. Inset, a micropipette (white arrow) close to the MN pool is showed. Right, voltage responses (top) to step hyperpolarizing currents (bottom) of two HMNs recorded from P7 rats receiving indicated oligonucleotides at P5. Dots, time points used to measure the peak voltage response to construct the $I$–$V$ plot (see the "Methods" section). **c**, **d** Microinjection of siRNA$_{p11}$ (2 µg/2 µl) into the fourth ventricle of P5 rats reduced mRNA$_{p11}$ levels in the brainstem at P7. **e** Vm and $R_N$ of HMNs recorded under pH 7.2. **f** Schematic summarizing effects of extracellular pH on MN IME and TASK-mediated K$^+$ currents. Int intracellular, Ext extracellular. **g** Changes in Vm and $R_N$ induced by variation of extracellular pH from 8.2 to 6.2 (TASK-dependent changes). **h** Time-course of Vm and $R_N$ obtained from recorded SMNs$^{wt}$ at the indicated days-in vitro (DIV) after seeding. **i** Effect of siRNA$_{p11}$ or cRNA (2 µM) on p11 and TASK1 in SMNs$^{wt}$ by western blot. **j** Confocal images showing that siRNA$_{p11}$ (2 µM) leads to a drastic reduction in p11 immunolabelling (red) together with an increase in TASK1 expression (green) in neuritic processes of SMNs$^{wt}$, as compared to cRNA (2 µM). Scale bar, 5 µm. **k** Box-plots of the mean (top) and integrated (bottom) intensity (in arbitrary units, a.u.) of TASK1 immunolabelling in neurites after treatments. **l**, **m** As in **e**, **g**, respectively, but from SMNs of the indicated genotypes under the specified treatments. Number of independent samples in each group is in parentheses. Error bars, SEM. *$p < 0.05$, **$p < 0.01$, ***$p < 0.001$; ns, not significant; by Student $t$-test **d**, **e**, **g**, **k**, **l**, one-way analysis of variance (ANOVA) with post hoc Holm–Sidak method **i**, **m** (Vm) or ANOVA on Ranks with post hoc Dunn's method **m** ($R_N$). Source data are provided as a Source Data file

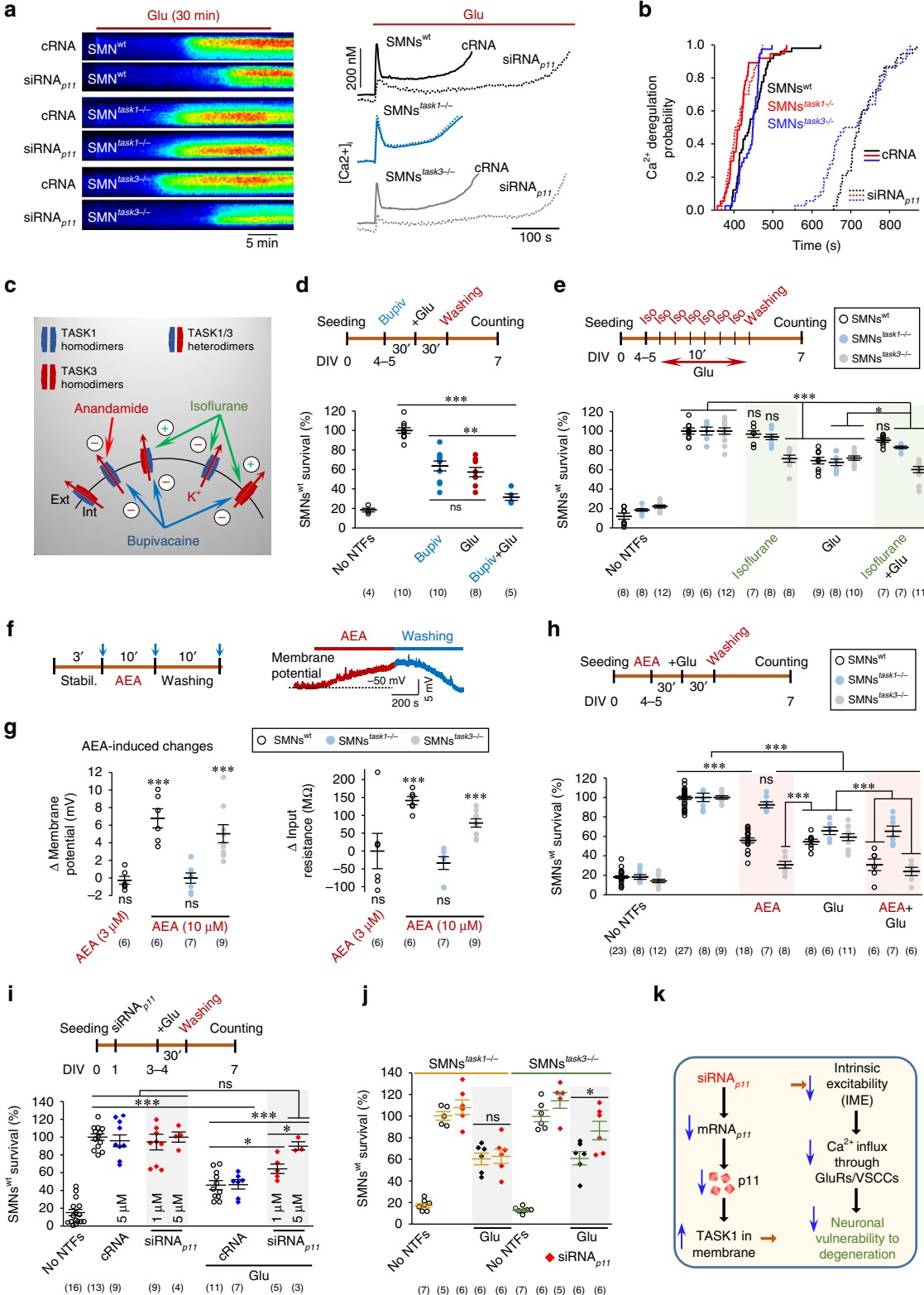

treatment with a noticeable delay in the time of $Ca^{2+}$ deregulation (i.e., likely prolonging the cell lifespan) in SMNs$^{wt}$ and SMNs$^{task3-/-}$ but, remarkably, not in SMNs$^{task1-/-}$ (Fig. 2b). siRNA$_{p11}$-mediated attenuation of the intracellular $Ca^{2+}$ response to glutamate involved reduction of $Ca^{2+}$ entry

through at least GluRs and VSCCs (Supplementary Note 1, Supplementary Fig. 3f–k).

Since MN IME is mainly determined by TASK channels (TASK1/3 heterodimers, as well as TASK1/1 and TASK3/3 homodimers)[18,22], we first investigated their impact on SMNs

**Fig. 2** p11 controls $Ca^{2+}$ signaling and vulnerability to an excitotoxic stimulus through TASK1, but not TASK3 subunits. **a** Dimensional displays (left) and time course of $[Ca^{2+}]_i$ alterations (right) obtained from SMNs treated with cRNA or $siRNA_{p11}$ (2 μM) from 1 DIV. Red horizontal lines indicate the glutamate (Glu, 150 μM) exposure interval. **b** Log-Rank test, Kaplan–Meier analysis reported that $siRNA_{p11}$ (dotted lines) delayed $Ca^{2+}$ deregulation in $SMNs^{wt}$ and $SMNs^{task3-/-}$ ($p < 0.001$), but not $SMNs^{task1-/-}$ ($p = 0.761$) relative to that in the cRNA condition. $n = 36$–49 SMNs. $Ca^{2+}$ imaging studies were performed at 3–4 DIV. **c** Schematic summarizing effects of listed drugs on TASK-mediated currents. **d** Effects of bupivacaine (Bupiv; 40 μM, 60 min) and Glu (150 μM, 30 min), added alone or in combination, on $SMNs^{wt}$ survival. **e** Effects of isoflurane (Iso; 0.8 mM each 10 min for 70 min) and Glu, added alone or in combination, on $SMNs^{wt}$, $SMNs^{task1-/-}$, and $SMNs^{task3-/-}$ survival. **f** Influence of AEA (10 μM) on Vm of a $SMN^{wt}$ at 4–5 DIV. **g** Changes in Vm (left) and $R_N$ (right) of $SMNs^{wt}$, $SMNs^{task1-/-}$, and $SMNs^{task3-/-}$ induced by AEA at the indicated concentrations. Number of SMNs in each group is in parentheses. **h** Effects of AEA (10 μM, 60 min) added alone or in combination with Glu on the survival of the indicated groups of SMNs. **i** Dose-dependent effects of $siRNA_{p11}$ on the survival of $SMNs^{wt}$ exposed to Glu (30 min, 150 μM). cRNA (5 μM) was taken as an additional control. Mean number of $SMNs^{wt}$ in the untreated condition was taken as 100%. **j** Mean effects of $siRNA_{p11}$ (2 μM) on survival of $SMNs^{task1-/-}$ and $SMNs^{task3-/-}$. No NTFs, culture medium not supplemented with neurotrophic factors. Schematics in **d**–**f**, **h**, **i** represent experimental protocols. Error bars, SEM. *$p < 0.05$, **$p < 0.01$, ***$p < 0.001$; ns, not significant; by Student's $t$-test **g** or one-way analysis of variance (ANOVA) with post hoc Holm–Sidak method **d**, **e**, **h**–**j**. **d**, **e**, **h**–**j** $n \geq 3$ independent experiments. **k** Schematic diagram depicting a neuroprotective strategy by interfering with p11 function (with $siRNA_{p11}$ in the example). Source data are provided as a Source Data file

survival. These experiments were performed in SMNs at 4–5 DIV, when an increase in surface expression of TASK channels is expected (see Fig. 1h). Drugs used for this purpose and their effects on TASK-mediated currents are summarized in Fig. 2c. A broad-spectrum TASK blocker, bupivacaine, reduced $SMNs^{wt}$ survival like glutamate (Fig. 2d, Supplementary Fig. 4a, b); however, bupivacaine potentiated glutamate harmfulness (Fig. 2d). Interestingly, the anesthetic isoflurane, a TASK1/3 and TASK3/3 opener, and a TASK1/1 blocker, compromised survival per se and potentiated glutamate toxicity in $SMNs^{task3-/-}$ but blocked and reduced glutamate toxicity in $SMNs^{wt}$ and $SMNs^{task1-/-}$, respectively (Fig. 2e). Furthermore, the selective TASK1 blocker anandamide (AEA, 10 μM), increased IME, reduced survival, and potentiated glutamate toxicity in $SMNs^{wt}$ and $SMNs^{task3-/-}$ but, conclusively, not in $SMNs^{task1-/-}$ (Fig. 2f–h, Supplementary Fig. 4c). However, AEA at a lower concentration (3 μM) did not modify either IME or $SMNs^{wt}$ survival (Fig. 2g, Supplementary Fig. 4c). The detrimental effects of AEA on $SMNs^{wt}$ survival were insensitive to AM281, a cannabinoid receptor-1 antagonist (Supplementary Fig. 4d). Altogether, TASK channels, and among them TASK1 subunits, determine MN IME and survival, as well as MN vulnerability to an excitotoxic insult. Therefore, promoting functional expression of TASK1 by reducing p11 is a reasonable strategy to protect MNs against excitotoxicity. Accordingly, $siRNA_{p11}$-induced p11 downregulation protected $SMNs^{wt}$ and $SMNs^{task3-/-}$ but, again, not $SMNs^{task1-/-}$, from glutamate (Fig. 2i, j, Supplementary Fig. 5a). Overall, p11 downregulation lowers MN IME by promoting TASK1 insertion at plasma membrane, then attenuates $Ca^{2+}$ influx through GluRs and VSCCs, and subsequently reduces vulnerability to excitotoxic degeneration (Fig. 2k).

**p11 knockdown is neuroprotective in the SOD1-G93A model.** The outstanding impact of p11 on MN IME, $Ca^{2+}$ signaling, and vulnerability to an excitotoxic insult suggests a leading role of p11 in neurodegenerative processes and highlights disruption of p11 function as a promising neuroprotective strategy. The superoxide dismutase 1 (SOD1)-G93A transgenic mouse model of familial ALS is the most widely used model in basic and preclinical studies of ALS. In this model, disease progression courses from hind to forelimbs. Hence, we focused our studies on MN pools located at the ventral horn of the lumbar spinal cord. In wild-type adult mice, MNs were the cell type in the ventral horn with highest levels of $mRNA_{p11}$ and p11 (Fig. 3a, b). In contrast, astrocytes within and surrounding MN clusters only rarely and weakly displayed p11 immunoreactivity (Fig. 3c). At the pre-symptomatic stage (1-month-old), $mRNA_{p11}$ and $mRNA_{task1}$ were already upregulated in the spinal cord of transgenic mice (Fig. 3d). Although

$mRNA_{p11}$ and p11 upregulation persisted throughout the symptomatic phase (4-month-old), both $mRNA_{task1}$ and TASK1 declined (Fig. 3d). At the early-symptomatic stage (3-month-old) that precedes MN loss, p11 was upregulated in lumbar SOD1-G93A MNs as compared to those from non-transgenic (Non-Tg) littermates (Fig. 3e). At this stage, p11-positive astrocytes were abundant within the characteristic astroglial reaction observed in the ventral horn of SOD1-G93A mice (Fig. 3f).

Experiments performed on SMNs provide evidence for a MN autonomous role of p11 upregulation in the ALS model. SMNs from SOD1-G93A embryos ($SMNs^{G93A}$) recapitulated p11–TASK1 dysregulation observed in the spinal cord of symptomatic mice (Supplementary Fig. 5b). In agreement with a reduced functional expression of TASK1 channels, $SMNs^{G93A}$ exhibited a more excitable state than that of Non-Tg SMNs ($SMNs^{Non-Tg}$) (Supplementary Fig. 5c). $Ca^{2+}$ imaging shed light on a gain-in-function of p11 for managing $Ca^{2+}$ dynamics in $SMNs^{G93A}$ (Supplementary Fig. 5d–h). Under cRNA, $SMNs^{G93A}$ showed higher resting $[Ca^{2+}]_i$ and peak $[Ca^{2+}]_i$ of the early rapid response to glutamate, but shorter time of $Ca^{2+}$ deregulation than $SMNs^{Non-Tg}$ (Supplementary Fig. 5d–h). Remarkably, neuroprotective effects of $siRNA_{p11}$ on the disturbance of $Ca^{2+}$ dynamics were greater in $SMNs^{G93A}$ (resting: $-39.9 \pm 8.8\%$; peak: $-45.0 \pm 4.1\%$; Δ time of $Ca^{2+}$ deregulation: $+347.8 \pm 15.4$ s) than in $SMNs^{Non-Tg}$ (resting: $-33.4 \pm 6.4\%$; peak: $-34.5 \pm 8.6\%$; Δ time of $Ca^{2+}$ deregulation: $+139.7 \pm 10.1$ s) (Supplementary Fig. 5d–g). Kaplan–Meier analysis of $Ca^{2+}$ deregulation reported that, under cRNA-treatment, the probability function of $SMNs^{G93A}$ was displaced to the left (i.e., likely shortening the cell lifespan) relative to $SMNs^{Non-Tg}$ ($p < 0.001$). Noticeably, no differences ($p = 0.130$) were found between the two groups of $siRNA_{p11}$-treated SMNs (Supplementary Fig. 5h). The probability curves for both of these groups were displaced to the right (i.e., likely prolonging the cell lifespan; $p < 0.001$), compared to those from their respective cRNA-treated pools (Supplementary Fig. 5h). Finally, $SMNs^{G93A}$ were more vulnerable to glutamate than $SMNs^{Non-Tg}$ and, as expected, $siRNA_{p11}$ robustly increased survival of $SMNs^{G93A}$ exposed to glutamate (Supplementary Fig. 5i).

Hence, potential neuroprotective effects of $siRNA_{p11}$ was assessed in the ALS model. We firstly tested the efficacy of a weekly injection of cRNA or $siRNA_{p11}$ into the 4th ventricle to affect lumbar MNs, beginning at the pre-symptomatic stage (2-month-old), once the skull was developed enough to implant the injection system (Fig. 3g). By using this procedure, we observed that $siRNA_{p11}$ consistently reduced $mRNA_{p11}$ and p11 protein in the lumbar spinal cord after 8 weeks of treatment (Supplementary Fig. 6a, b). Quantitative analysis of p11-immunolabelling in lumbar MNs revealed that the decline in labeling intensity was already evident

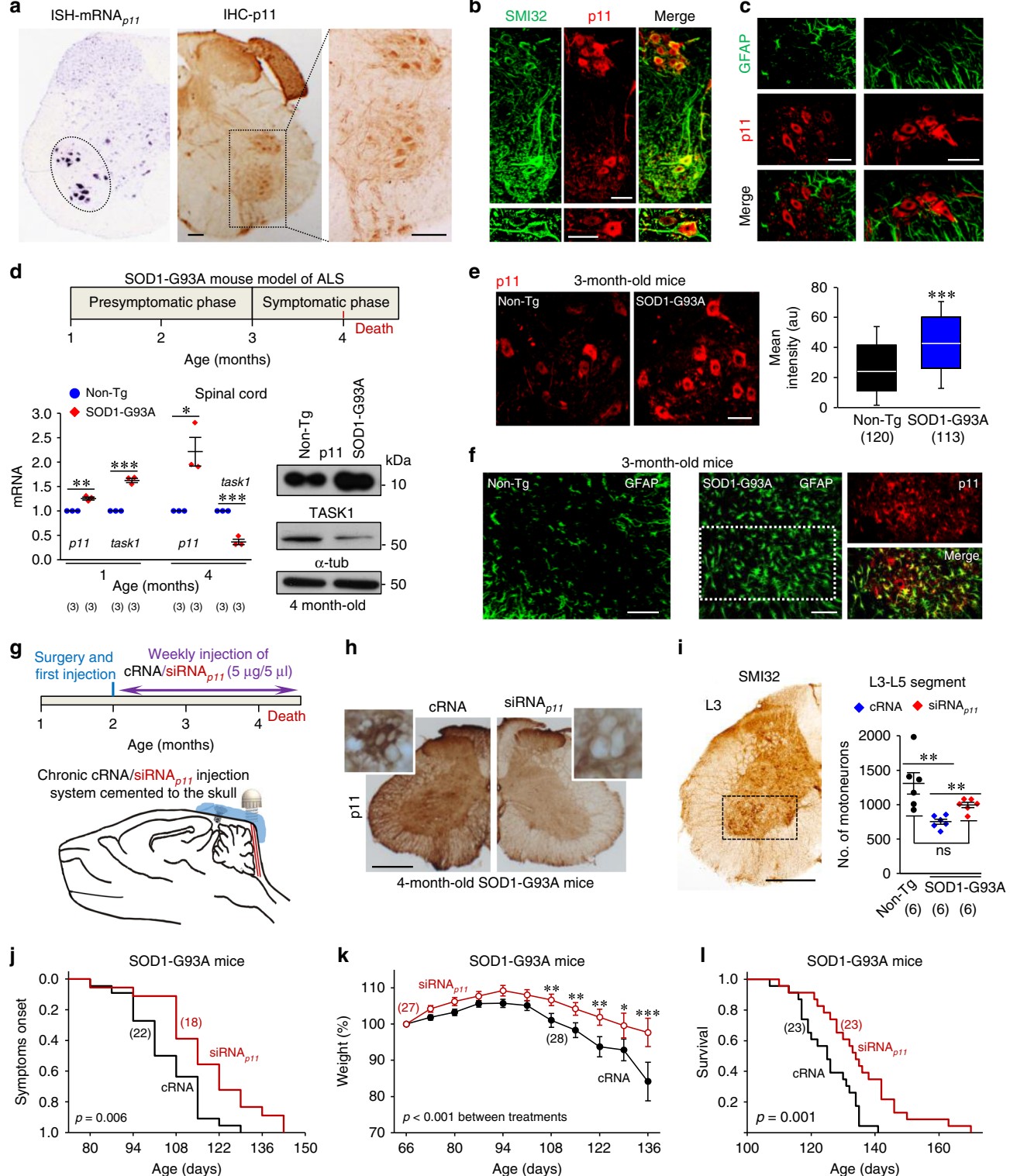

2 weeks after beginning of siRNA$_{p11}$-treatment (Supplementary Fig. 6c). Thereafter, the number of SMI32-positive MNs in the L3–L5 lumbar segment from 4-month-old mice was quantified. Chronic administration of siRNA$_{p11}$ in SOD1-G93A mice, which reduced p11-immunoreactivity in the lumbar spinal cord (Fig. 3h), attenuated MN loss and improved motor performance (rotarod, footprint, runtime, and grip strength) as compared with cRNA-treated animals (Fig. 3i, Supplementary Fig. 6d–h). Correspondingly, siRNA$_{p11}$ delayed symptoms onset, alleviated weight loss, and

prolonged survival (cRNA: $125.0 \pm 1.8$ days; siRNA$_{p11}$: $135.3 \pm 3.0$ days, $p < 0.001$, Student's $t$-test) in SOD1-G93A mice (Fig. 3j–l). Beneficial effects did not result from different levels of transgene expression in both cohorts (Supplementary Fig. 6i).

**p11 knockdown is neuroprotective in the nerve injury model.** In situ hybridization and immunohistochemistry studies clearly showed that HMNs, but not astrocytes, express high levels of p11 in the adult mouse (Fig. 4a, b). Bidirectional trophic

**Fig. 3** Neuroprotective effects of siRNA$_{p11}$ in the SOD1-G93A mouse model of ALS. **a** In situ hybridization (ISH, from the Allen Brain Atlas) and immunohistochemistry (ICH) showing the expression pattern of p11 in the lumbar spinal cord of adult wild-type mice. **b, c** Confocal images of lumbar spinal cords immunolabeled for p11 and the MN marker SMI32 **b** or the astroglial marker GFAP **c. d** mRNA (left) and protein (right) levels of p11 and TASK1 in the spinal cord of SOD1-G93A mice and their Non-Tg littermates, at the indicated stages (see schematic on top). **e** p11 immunofluorescence (left) and mean intensity (arbitrary units, a.u.; right) of labeling in lumbar MNs of indicated mice and age. MNs were identified as in **b. f** Confocal images showing astroglial reaction in the ventral horn of SOD1-G93A mice. Double immunolabeling reported p11-immunoreactive astrocytes in the boxed area. **g** Schematic of timing for injection system implantation and cRNA/siRNA$_{p11}$ administration. **h** p11 immunohistochemistry performed in lumbar spinal cord sections obtained from indicated mice which received cRNA or siRNA$_{p11}$ from P60. Insets, examples of two vacuolated MNs with different levels of p11 immunostaining. **i** Number of SMI32-identified MNs in the L3–L5 spinal cord segments of 4-month-old mice with the indicated genotypes that received the stated treatments beginning at P60. The box indicates the area of the ventral horn containing the MN pools that were quantified. Scale bars, **b, c, e**, 50 μm; **a, c**, 100 μm; **h, i**, 300 μm. **j–l** Cumulative probability curves of symptom onset **j**, survival **l**, and time course of mean body weight **k** for cRNA-treated and siRNA$_{p11}$-treated SOD1-G93A mice. Number of independent samples in each group is in parentheses. Statistic outputs are displayed in plots. Error bars, SEM. *$p < 0.05$, **$p < 0.01$, ***$p < 0.001$; n.s., not significant; by Student $t$-test **d, e, i** or two-way ANOVA with post hoc Holm–Sidak method **k**. Log-Rank test, Kaplan–Meier analysis was applied to **j** and **l**. Source data are provided as a Source Data file

communication between MNs and their target myocytes is necessary to keep MNs alive. Therefore, resection of a segment of the XIIth nerve, in which the possibility of nerve regeneration was minimized, led to significant loss of neurons in the HN (Fig. 4c, d). Neuronal loss was still undetectable one week post-lesion. However, a significant drop in the number of neurons was observed 2 weeks post-injury and, progressively declined up to ~40% from 4th to 8th week post-lesion (Fig. 4d). Remarkably, p11 was upregulated in the HN preceding MN loss (Fig. 4e). p11 upregulation was evident as soon as one day after injury and rose at 7th days post-lesion (Fig. 4e). p11 induction in the HN was mainly associated with upregulation in HMNs. p11-positive astrocytes were not observed in the injured side, even though the characteristic astroglial reaction occurred in the injured side (Fig. 4f, g). It was also interesting to note that, one week after nerve resection, mRNA$_{task1}$ declined in the injured HN (Fig. 4e), just as in the spinal cord of symptomatic SOD1-G93A mice.

To explore cell-specificity of p11 to MN loss, neuron-specific lentiviral constructions (LVVs) (Fig. 4c), which direct the expression of a small hairpin RNA against *p11* (LVV-shRNA$_{p11}$) or against bacterial *lacz* (LVV-shRNA$_{lacz}$), were used. LVV-shRNA$_{p11}$ reduced mRNA$_{p11}$, enhanced TASK-dependent changes, and protected SMNs$^{wt}$ against glutamate (Supplementary Fig. 7). Furthermore, a single intramedullar microinjection of LVV-shRNA$_{p11}$, led to a drastic reduction of mRNA$_{p11}$ and p11 in both HNs 24 days after viral administration (Fig. 4h). Interestingly, LVV-shRNA$_{p11}$ microinjection in the injured HN reduced both p11 expression in injured, but not in intact HMNs, and cell loss (Fig. 4i, j). Therefore, neuron-specific p11 knockdown was neuroprotective for axotomized MNs.

The reached outcomes in both degenerative models highlight p11 upregulation as a key event in an overall mechanism contributing to MN degeneration in diverse motor pathologies regardless of causal origin.

**RNS upregulates p11 and inhibits TASK-like currents**. In order to obtain a more complete picture of this pathogenic mechanism, we looked for a widespread pathological event upstream p11 upregulation. In this context, large quantities of nitric oxide (NO) lead to RNS, a potential factor in excitotoxic neurodegeneration[6–9]. Interestingly, NO has been reported to upregulate p11 in human bronchial epithelial cells via protein kinase G (PKG)[23]. Finally, RNS inhibits TASK-mediated currents in MNs throughout PKG[10] (Fig. 5a). Here, RNS was simulated by long-term action (≥4 h) of DETA/NO (1 mM). At this concentration, this NONOate releases pathological amounts of NO, as estimated in vivo[6,7,24,25], over a 24-h period. NO inhibited TASK-like currents in HMNs independently of short-term mechanisms such as direct Gα$_q$ co-association[26] and/or Rho-kinase (ROCK)-mediated phosphorylation[27], since a Gα$_q$

inhibitor or short-term presence (15 min) of a ROCK inhibitor did not prevent/revert long-term effects of DETA/NO (Supplementary Fig. 8a–c). However, the long-term mechanism involved RhoA-ROCK signaling, but not dysregulation of TASK1 and TASK3 expression (Supplementary Fig. 8d–g).

As reported for smooth muscle cells[28], NO, via PKG, increased both, the amount of the phosphorylated form of the transcription factor ATF-1 and RhoA in the HN of neonatal rats (Fig. 5b, c). Conclusively, DETA/NO, via PKG-ROCK, upregulated mRNA$_{p11}$ and p11, but not mRNA$_{task1}$ or mRNA$_{task3}$ (Fig. 5c–e). Furthermore, DETA/NO-evoked p11 upregulation and pH-responsiveness reduction were fully blocked by retinoic acid, which posttranslationally reduces p11 levels (Supplementary Fig. 9a–c), strengthening the notion that RNS inhibits TASK-like channels in a p11-dependent way. Several additional findings support a pathological NO-p11-TASK association in damaged MNs: (i) XIIth nerve crushing at P3 induces neuronal NO synthase (nNOS) expression in MNs[6], and MN loss at P10, but not at P7 (Fig. 5f; Supplementary Fig. 9d, e); (ii) RhoA and p11 upregulation in the injured HN, together with hyper-excitability and reduction in TASK-mediated currents of injured HMNs, precede neuron loss (Fig. 5g–j); (iii) daily administration of the NOS inhibitor L-NAME fully prevents RhoA and p11 upregulation, and substantially preserves TASK-mediated currents in damaged HMNs (Fig. 5f–j); (iv) intracerebroventricular microinjection of siRNA$_{p11}$ increases TASK-dependent changes in IME in axotomized HMNs (Fig. 5f, j); and (v) treatment of brainstem slices (P7) with retinoic acid down-regulates p11 in the injured side and partially recovers pH-responsiveness of insulted HMNs (Supplementary Fig. 9f–h).

Therefore, in axotomized MNs, RNS resulted in the recruitment of the PKG–RhoA–ROCK pathway to upregulate p11, thus reducing TASK-mediated currents underlying hyper-excitability that precedes MN loss (Fig. 5k).

**RNS stimulates the human p11 promoter via a Sp1-binding site**. To strengthen clinic and translational potential, we tested RNS effects on luciferase activity coupled to the human p11 promoter (region −1436 to +89) in the transfected and differentiated MN-like cell line NSC34 (dNSC34). In preliminary experiments, DETA/NO induced mRNA$_{p11}$ and p11 in dNSC34s (Supplementary Fig. 10a), indicating their capability to couple p11 expression to NO. Only DETA/NO concentrations (1–2 mM), releasing pathological amounts of NO, increased luciferase activity (Fig. 6a), in a time-dependent manner (Supplementary Fig. 10b). We chose a 48-h treatment with DETA/NO 1 mM as the optimal condition for subsequent experiments. DETA/NO stimulated luciferase activity through the NO target and PKG-activator soluble guanylyl cyclase (sGC), PKG, and ROCK

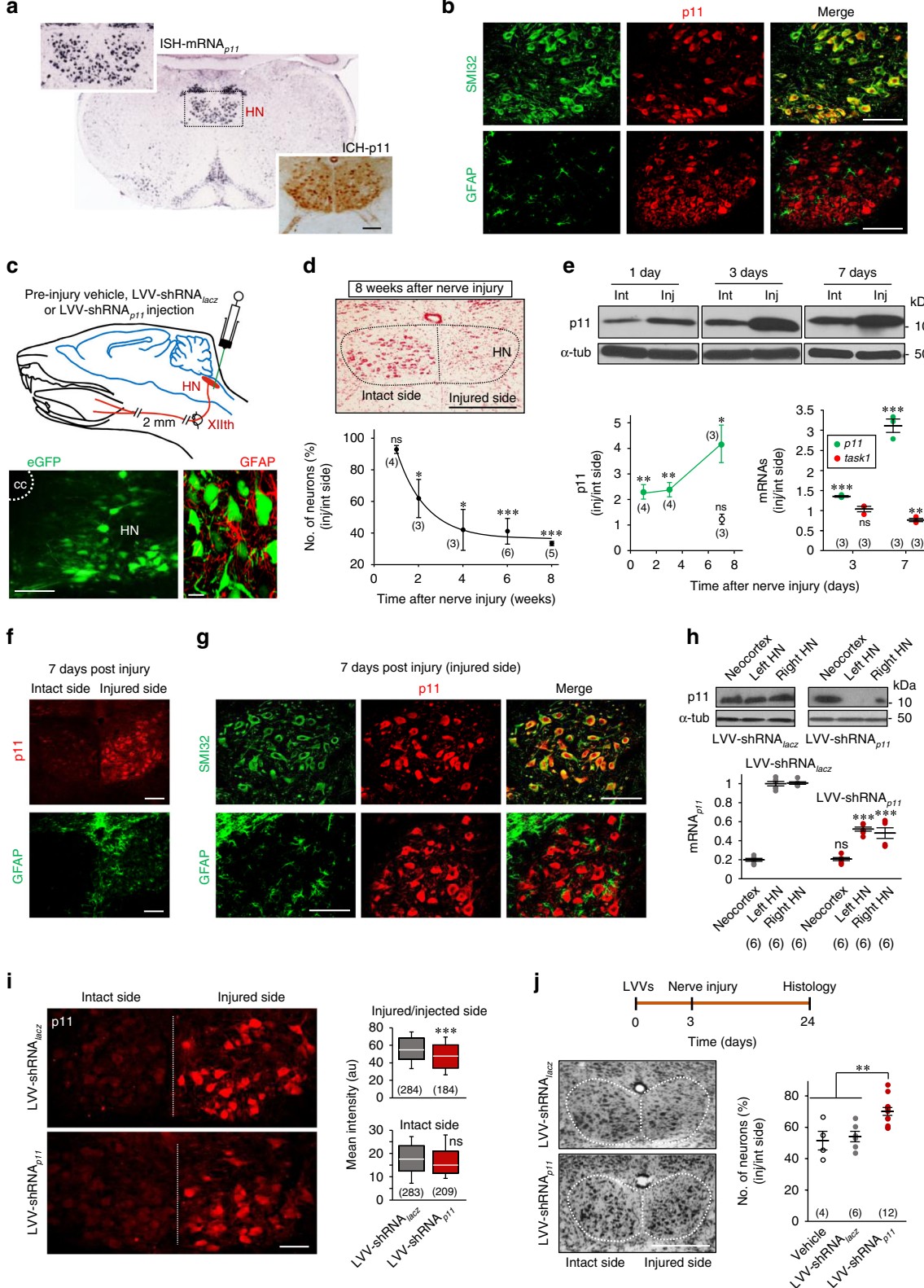

(Fig. 6b). Furthermore, serial 5′ promoter deletions isolated a minimal NO-responsive element at the −96 to −70 segment (Fig. 6c). Finally, site-directed mutagenesis identified a potential binding site for Sp1 (GC-box) essential for NO-responsiveness (Fig. 6d). Since, three additional Sp1 and a NO-responsive AP-1-binding sites upstream −96 have been identified[23,29], whether p11

promoter is regulated by other NO-responsive elements and/or by still unrevealed factors remains to be investigated.

Electrophoretic mobility shift assays (EMSA) displayed two well-defined complexes when the wild-type −96 to −70 sequence was incubated with nuclear protein extracts from dNSC34s. Both complexes were absent if the sequence comprised the

**Fig. 4** Cell-type specific p11 dysregulation is pivotal in a traumatic model of MN degeneration. **a** In situ hybridization (ISH, from the Allen Brain Atlas) and immunohistochemistry (ICH) showing the expression pattern of p11 in the HN of adult wild-type mice. **b** Confocal images showing p11 in SMI32-labeled HMNs (top) but not in GFAP-positive astrocytes (bottom). **c** Top, experimental design. Bottom, images illustrating neuronal-specificity of a lentivirus, which direct the expression of eGFP. **d** Effect of resection of a 2-mm segment of the XIIth nerve and immediate proximal stump ligation on the number of HMNs in adult mice. A neutral red-stained coronal section at the indicated post-injury time is shown on top. **e** Western blots (top) and plot (bottom, left) showing p11 upregulation in the injured (Inj) relative to the intact (Int) HN. Empty circle, sham condition. Bottom, right, qRT-PCR analysis. **f** p11 upregulation (top) and astroglial reaction (bottom) in the insulted HN after nerve injury. **g** Double immunohistochemistry showed p11 expression in SMI32-identified HMNs but not in GFAP-positive astrocytes in the damaged side. **h** Effect of microinjection of indicated LVVs in the midline between both HNs **c** on protein and mRNA levels for p11 in HNs. Neocortex was a control for systemic transduction. gapdh and α-tub were internal controls for qRT-PCR and western blotting, respectively. **i** LVV-shRNA$_{p11}$ microinjection, 3 days before nerve injury (see schematic in **j**), reduced p11 labeling in SMI32-identified HMNs relative to LVV-shRNA$_{lacz}$. Mean intensity (arbitrary units, au) of p11 labeling in injured/intact HMNs (top) or in the intact/non-injected side (bottom) after indicated LVVs injection. **j** LVV-shRNA$_{p11}$ reduced neuronal loss, as assessed by Nissl staining in the injured versus the intact HN. In **d**, **e**, **i** (top), **j**, data values in the injured side are displayed relative to those obtained for the intact side. Scale bars, **c** (right), 20 μm; **a**–**c** (left), **f**, **g**, **i**, 100 μm; **d**, **j**, 500 μm. Number of independent samples is in parentheses. Error bars, SEM. *$p < 0.05$, **$p < 0.01$, ***$p < 0.001$; n.s., not significant; by Mann–Whitney test **d**, **e**, **h**, Student t-test **i** or one-way ANOVA with post hoc Holm–Sidak method **j**. Source data are provided as a Source Data file

NO-irresponsive mutation (Fig. 6e). Competitive inhibition, by increasing the concentration of the canonical Sp1 sequence, reduced complex formation in a dose-dependent manner (Fig. 6f). In addition, incubation with antibodies against Sp1 or Sp3 either alone or in combination clearly reduced the intensity of the complex with lower electrophoretic mobility (Fig. 6g). Altogether, these outcomes support a binding of Sp1, and probably Sp3, to the NO-sensitive GC-box within the −96 to −70 segment of p11 promoter. Finally, DETA/NO at 1 mM, but not at 0.1 mM, upregulated mRNA$_{Sp1}$ and mRNA$_{p11}$ in SMNs$^{wt}$ via the PKG-ROCK signaling (Fig. 6h, i), further supporting that DETA/NO effects on promoter activity result in an increase in transcriptional activity of the p11 gene in murine MNs.

Therefore, we hypothesize that RNS, established under neurological conditions by pathological concentrations of NO, recruits sGC–PKG–ROCK signaling and subsequently upregulates the transcription factor Sp1, which stimulates p11 promoter activity and promotes p11 expression (Fig. 6j).

**Mit-A, a feasible neuroprotector that dysregulates p11–TASK1.** Based on the preceding framework, we tested the viability of interference with Sp1 function as a neuroprotective strategy. Transient transfection of a plasmid directing the expression of a truncated form of Sp1, which acts as a dominant negative (dnSp1), strongly reduced p11 in dNSC34s (Supplementary Fig. 11a, b). Furthermore, a neuron-specific LVV directing the expression of dnSp1 (LVV-dnSp1) reduced mRNA$_{p11}$, strongly promoted functional expression of TASK-like channels, and almost fully blocked glutamate toxicity on SMNs$^{wt}$ (Supplementary Fig. 11c–e). These data support that Sp1 controls constitutive expression of p11 in dNSC34s and SMNs$^{wt}$. Nevertheless, given the current limitations of LVVs for medical use, we tested a Sp1-interfering drug with higher clinical potential. In this context, Mit-A, a FDA-approved anticancer antibiotic that binds to GC-rich sequences, is usually used to interfere with Sp1 binding to its target sites.

Mit-A fully avoided NO-triggered stimulation of the minimal NO-responsive element of the human p11 promoter (Supplementary Fig. 12a). This antibiotic strongly reduced mRNA$_{p11}$ and, unexpectedly, increased mRNA$_{task1}$ levels in SMNs$^{wt}$, both of which would be expected to result in additive functional effects (Supplementary Fig. 12b). Thus, Mit-A drastically lowered IME and promoted TASK-mediated changes in SMNs$^{wt}$ and SMNs$^{task3−/−}$, but, conclusively, not in SMNs$^{task1−/−}$ (Supplementary Fig. 12c–g). Mit-A also attenuated glutamate-evoked Ca$^{2+}$ homeostasis disruption in a TASK1-dependent manner (Supplementary Fig. 12h–j). Kaplan–Meier analysis revealed that Mit-A caused a strong delay in the time of Ca$^{2+}$ deregulation (i.e.,

likely prolonging the cell lifespan) in SMNs$^{wt}$ and SMNs$^{task3−/−}$, but again not in SMNs$^{task1−/−}$ (Supplementary Fig. 12k). Finally, Mit-A protected SMNs$^{wt}$ and SMNs$^{task3−/−}$ but, once again, not SMNs$^{task1−/−}$, against glutamate toxicity (Supplementary Fig. 12l). These results strengthen the therapeutic interest of Mit-A by reducing neuronal hyper-excitability and vulnerability against excitotoxicity.

**Mit-A is neuroprotective in the SOD1-G93A model.** Both Sp1 upregulation in the spinal cord from symtomatic (4-month-old) SOD1-G93A mice, and the increase of nuclear immunolabelling for Sp1 in lumbar MNs preceding cell loss (3-month-old) underpin a key role for Sp1 in the ALS model (Fig. 7a, b). Sp1 was also upregulated in SMNs$^{G93A}$ (Supplementary Fig. 13a). Accordingly, Mit-A lowered IME, promoted TASK-dependent changes and strongly attenuated glutamate-triggered Ca$^{2+}$ disturbance in SMNs$^{G93A}$ (Supplementary Fig. 13b–g). The Kaplan–Meier analysis of Ca$^{2+}$ deregulation showed that Mit-A displaced to the right the probability function of SMNs$^{G93A}$ (Supplementary Fig. 13h), and lowered their vulnerability to glutamate (Supplementary Fig. 13i).

Therefore, neuroprotective action of Mit-A is underpinned by the additive action on p11 and TASK1 expression, which lowers IME and delays glutamate-evoked Ca$^{2+}$ deregulation, at least in part, by promoting TASK1 expression at the plasma membrane (Fig. 7c). Then, we tested preclinical efficacy of Mit-A in SOD1-G93A mice. Initially, optimal dosage of Mit-A was determined in wild-type mice. Chronic and daily administration of growing doses of Mit-A in the drinking water for 3 weeks, reduced p11 and increased TASK1 expression in the spinal cord (Supplementary Fig. 14a, b). Given that p11 reduction after all tested doses was similar, we primarily selected the lower one (30 μg kg$^{−1}$ day$^{−1}$) for chronic treatment. This dose for 2 weeks reduced and increased, respectively, p11 and TASK1-like subunits in lumbar MNs (Supplementary Fig. 14c, d). Furthermore, daily administration of Mit-A from P30 kept p11–TASK1 appropriately dysregulated in the spinal cord and attenuated the loss of lumbar MNs in 4-month-old SOD1-G93A mice (Fig. 7d, e). Mit-A delayed symptoms onset, alleviated weight loss, prolonged survival, and improved motor performance in SOD1-G93A mice (Fig. 7f–h; Supplementary Fig. 14e–g). Antibiotic-induced neuroprotective effects were substantiated by functional benefits at the muscle. Myocytes denervation is a characteristic hallmark of ALS, identified in needle electromyography by spontaneous discharges of skeletal muscle fibers or fibrillations (Fig. 7i). Thus, the frequency of fibrillations gradually increased in transgenic mice from week 13 to week 16 of age but these were generally absent in Non-Tg animals (Fig. 7j–m). Interestingly, the rate of fibrillations

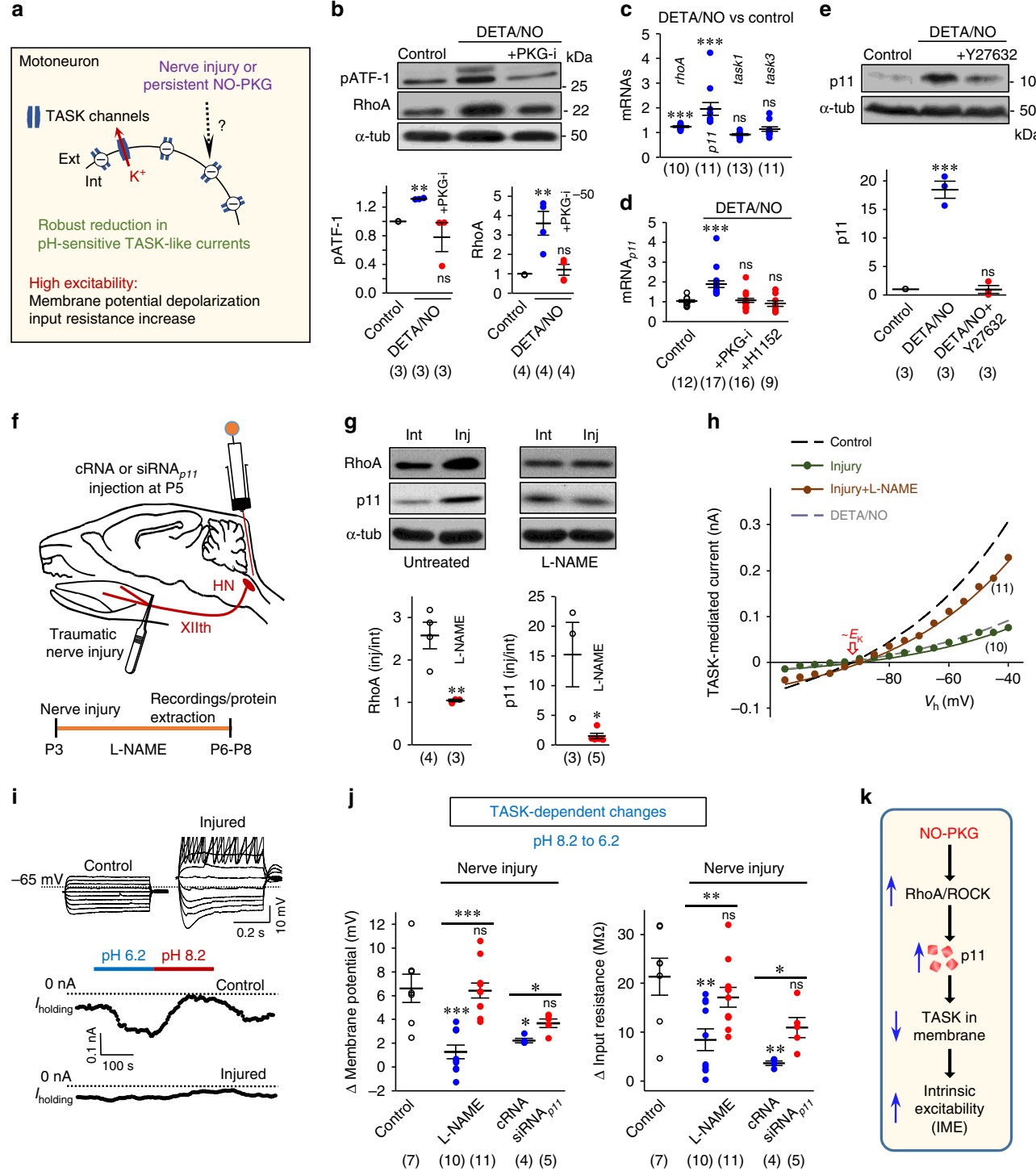

under Mit-A treatment was drastically decreased at all ages tested (Fig. 7j–m).

Similar outcomes were observed when Mit-A treatment ($30 \, \mu g \, kg^{-1} \, day^{-1}$) began at 2 months of age in ALS mice (Supplementary Fig. 15a–f). Although two additional drug regimens (30 or $150 \, \mu g \, kg^{-1} \, day^{-1}$), beginning at P90, prolonged survival and improved rotarod performance of SOD1-G93A mice (Supplementary Fig. 15c, d), the results were not conclusive on the beneficial effects of these treatments in the remaining parameters (Supplementary Fig. 15a, b, e, f). The higher dose tested even worsened body weight, footprint, and

runtime outcomes over the time-course (Supplementary Fig. 15b, e, f).

Finally, Mit-A ($30 \, \mu g \, kg^{-1} \, day^{-1}$) and Riluzole ($25 \, mg \, kg^{-1} \, day^{-1}$), the most clinically used drug for ALS patients, resulted in similar beneficial outputs in SOD1[G93A] mice when parallel treatments began at 2 months of age (Supplementary Fig. 15g–m).

**Mit-A is neuroprotective in the motor nerve injury model**. One week after resection of a 2-mm-segment of the XIIth nerve of adult mice, Sp1 was increased in the HN of the injured side

**Fig. 5** RNS upregulates p11 and inhibits TASK channels in a traumatic model of MN degeneration. **a** Effects of nerve injury or persistent action of the NO-PKG pathway on MN IME and TASK-mediated currents. **b-e** Expression of phosphorylated ATF-1 (pATF-1) and RhoA proteins **b**, indicated mRNAs **c**, **d** and p11 **e** in HNs from neonatal brainstem slices incubated for 6 h in aCSF alone (control), or supplemented with either DETA/NO (1 mM), DETA/NO+PKG-i (10 μM), DETA/NO+H1152 (20 μM), or DETA/NO+Y27632 (10 μM). In **c**, mRNA levels of DETA/NO versus control condition are presented. In **b-e**, $n \geq 3$ independent experiments. **f** Experimental model. Crushing was inflicted to the XIIth nerve of P3 rats. Pups were untreated, L-NAME-treated, or they received a single intracerebroventricular microinjection of cRNA or siRNA$_{p11}$. **g** Immunoblots for RhoA and p11 of HNs from P7 rats treated as shown. *gapdh* and α-tub were the internal controls for qRT-PCR and western blotting, respectively. **h** Averaged I–V relationships of the TASK-mediated current for the indicated treatments (see experimental design in **f**) (see Supplementary Fig. 9b). Data were well fitted ($r > 0.9$) with the Goldman–Hodgkin–Katz equation (solid line). Current reversed close to −90 mV, near the reversal potential predicted for K$^+$ ($E_K$). Goldman–Hodgkin–Katz fits for control and DETA/NO (dashed lines), previously reported[10], are shown for comparison. **i** Top, current-clamp recordings of the voltage responses to a series of depolarizing and hyperpolarizing current pulses (0.5 s duration, 0.04 nA increments) from a control and a damaged HMN recorded 4 days after axonal injury. Bottom, time courses illustrating alterations in current holding ($I_{holding}$) recorded in response to extracellular pH changes for two P7 HMNs held at −65 mV. **j** Plots represent TASK-dependent changes in Vm and $R_N$ for each listed treatment group. Number of independent samples is in parentheses. Error bars, SEM. *$p < 0.05$, **$p < 0.01$, ***$p < 0.001$; n.s., not significant; by Student's t-test **c**, **g** or one-way ANOVA with post hoc Holm–Sidak method **b**, **d**, **e**, **j**. **k** Schematic diagram depicting the mechanism by which RNS impacts on MN IME. Source data are provided as a Source Data file

(Fig. 8a). Sp1-immunolabelling was enhanced in the injured side at 7 days after injury (Fig. 8b, c), thus preceding HMNs loss (see Fig. 4d). Interestingly, lesion-induced Sp1 upregulation was avoided by 2 days treatment with the NOS inhibitor L-NAME (Fig. 8d). Furthermore, chronic administration of the NOS inhibitor or Fasudil (10 mg kg$^{-1}$ day$^{-1}$), an inhibitor of the downstream NO-acting ROCK (see Fig. 6j), reduced the loss of injured HMNs (Fig. 8e). All these pharmacological treatments began at 5 days post-lesion, when de novo expression of nNOS in injured MNs was evidenced[30]. These outcomes indicate that Sp1 upregulation, downstream de novo NO synthesis, is part of a mechanism that precedes MN death after motor nerve injury.

Mit-A administration in the drinking water at two doses (30 or 300 μg kg$^{-1}$ day$^{-1}$) drastically reduced neuron loss observed 21 days after nerve resection (Fig. 8f). Neuroprotection was confirmed in *task3*$^{-/-}$, but was absent in *task1*$^{-/-}$ mice (Fig. 8f). Intraperitoneal administration of the higher dose of Mit-A (300 μg kg$^{-1}$ day$^{-1}$) showed slight but significant neuroprotective effects (Fig. 8f). Therefore, Mit-A delays MN degeneration after nerve injury, by a mechanism involving, at least, TASK1 subunits.

Therefore, Sp1 is also a feasible partner in an overall mechanism contributing to MN degeneration and point to the therapeutic interest of Mit-A for the treatment of diverse motor pathologies regardless of causal origin.

**Cell-specific Sp1–p11 knockdown protects SOD1-G93A MNs.** p11 upregulation occurs in astrocytes and MNs before MN loss in SOD1-G93A mice (see Fig. 3e, f). Then, we studied a cell autonomous role of Sp1–p11 in MN degeneration. In this context, most used MN-specific conditional knockouts mice target the subset of MNs that is mainly spared in the disease (VAChT:Cre model)[31–35], or other cells types besides MNs (Hb9:Cre model)[36–38], including spinal cord interneurons involved in rhythm generation during locomotor activity. Therefore, here we performed intraspinal microinjections of neuron-specific LVV-dnSp1 or LVV-shRNA$_{p11}$ (see Fig. 4c) into the ventral horn of the lumbar segments in 2-month-old SOD1-G93A mice (Fig. 9a). A careful inspection of the L1–L3 ventral horns from 4-month-old animals showed the presence of the reporter protein GFP in p11-immunoreactive MNs, but not in glial-like structures (Fig. 9b–e). Strikingly, p11-immunolabelling was reduced in GFP-positive MNs after LVV-dnSp1 or LVV-shRNA$_{p11}$ injection (Fig. 9c–f). Furthermore, the frequency of vacuolated/degenerating MNs was higher in LVV-shRNA$_{lacz}$- compared to LVV-dnSp1/LVV-shRNA$_{p11}$-treated SOD1-G93A mice (Fig. 9g, h). Accordingly, the number of apparently healthy MNs was higher in mice receiving LVV-dnSp1 or LVV-shRNA$_{p11}$ than in those

treated with LVV-shRNA$_{lacz}$ (Fig. 9g–i). Therefore, these outcomes, together with those from SMNs, strongly support a cell autonomous role of Sp1/p11 in MN degeneration. However, whether p11 upregulation in glial cells also contributes in some degree to disease progression merits further investigation.

## Discussion

Our study identify a mechanism that regulates neuronal IME, Ca$^{2+}$ signaling, and vulnerability to degeneration. The ubiquitous transcription factor Sp1 controls the expression of p11, a retention factor at the endoplasmic reticulum for TASK1[14], which regulates channel expression at the cell surface, thus fine-tuning neuronal IME and Ca$^{2+}$ influx via GluRs and VSCCs (Fig. 10a). Interference with Sp1 function reduces p11 expression and attenuates MN IME, Ca$^{2+}$ dynamics, and vulnerability via TASK1. These effects were similar to those revealed by disrupting p11. Hence, we conclude that modulation of the Sp1–p11–TASK1 triad offers a promising strategy to minimize excitotoxic damage occurring in a multitude of neuropathologies associated with dysregulated membrane excitability and/or excitotoxic degeneration, including both acute and chronic neurodegenerative diseases[7,8].

Of pathological relevance, Sp1–p11–TASK1 dysregulation occurs in two motor neurodegenerative models with very different causal origins, which shared excitotoxicity and hyper-excitability as pathophysiological hallmarks[3,4,6]. Sp1–p11–TASK1 strongly impacts on IME of MNs. Interfering with either Sp1 or p11 lowers MNs IME by promoting functional expression of TASK1, then protecting MNs against excitotoxicity. However, whether hyper-excitability is harmful or neuroprotective for MNs in ALS is a current subject of debate. Intrinsic hyper-excitability of MNs has been assumed to participates in the excitotoxic process[39]. In contrast, hyper-excitability is proposed as a transitory state[40,41] that even might be beneficial[42,43] in ALS, and which might underlie survival of S-type MNs in the ALS model[42,43]. Here, at early-presymptomatic (P30) stage, p11 and TASK1 upregulation could be part of a homeostatic and compensatory process to neutralize deficits during early phases of disease[43]. Nevertheless, persistent p11 upregulation and TASK1 down-expression occurring at the symptomatic stage might potentiate excitotoxic processes[39] by strengthening, for example, the incoming synaptic excitation to MNs that might result from inhibitory bouton loss and excitatory synapse gain[44]. In this line, chronic interference with Sp1 and/or p11 beginning at presymptomatic (P30 or P60) or early-symptomatic (P90) stages resulted neuroprotective for SOD1-G93A MNs and/or mice.

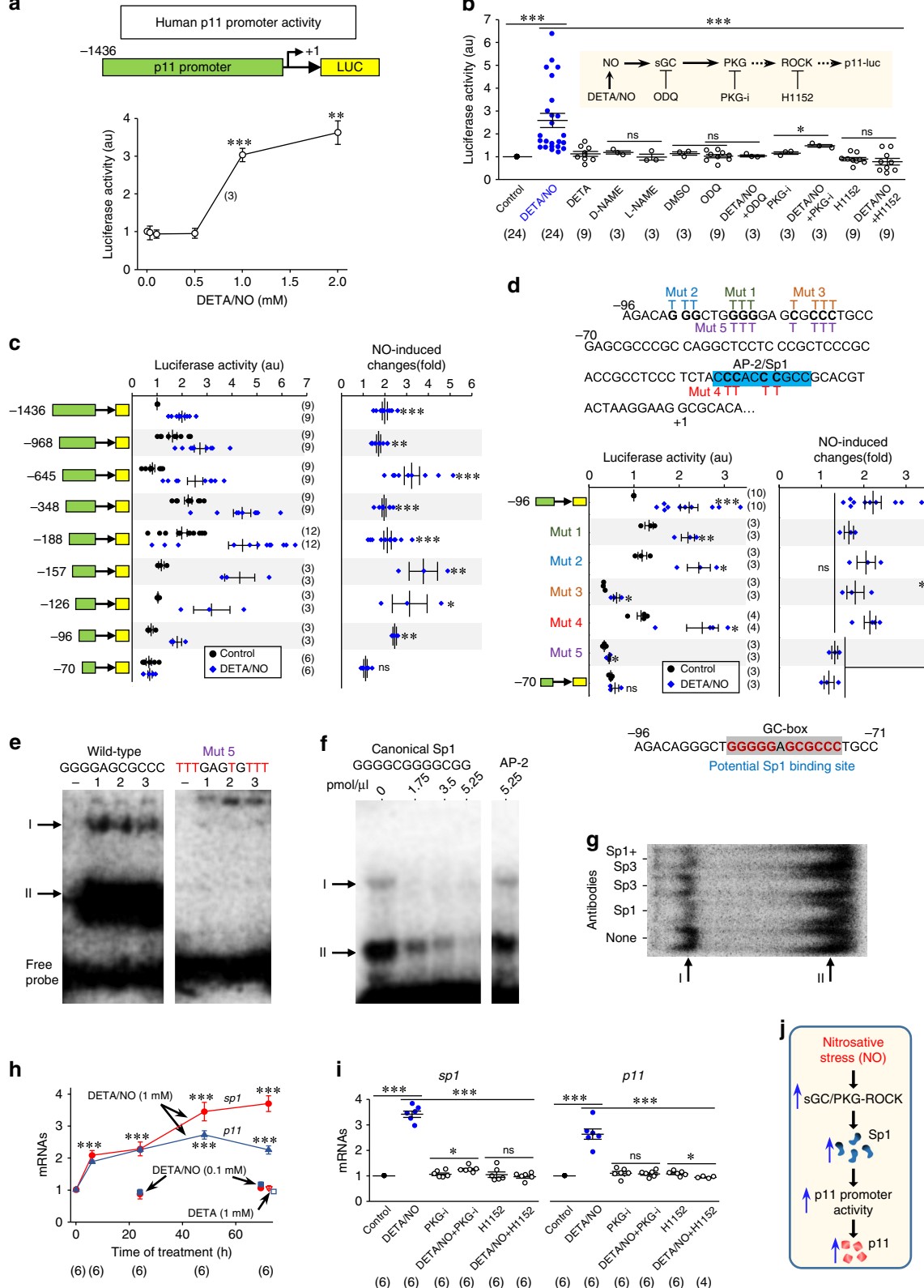

Additional controversy comes from passive membrane properties that lie behind hyper-excitability in SOD1-G93A MNs. While embryonic (E17.5)[45] and early-postnatal (P6–P10)[46] spinal MNs displayed $R_N$ increase or Vm depolarization, respectively, no changes in Vm nor $R_N$ accompanied hyper-excitability in HMNs$^{G93A}$ (P4–P10)[47]. Here, hyper-excitability of SMNs$^{G93A}$ (E12.5), recorded at 4–5 DIV, involved Vm depolarization and $R_N$ increase. Interestingly, hyper-excitable SMNs$^{G93A}$ (E12–E15)[48,49], recorded at 10–30 DIV, did not displayed Vm and $R_N$ alterations. This agrees with an early transitory state in which these passive membrane properties of SMNs$^{G93A}$ are affected. All these discrepancies

**Fig. 6** RNS stimulates the human p11 promoter through a Sp1-binding site. **a**, **b** Luciferase activity (in arbitrary units, a.u.) coupled to the human p11 promoter from transfected dNSC34 incubated for 48 h either with the indicated DETA/NO concentrations **a** or with drugs modulating the NO/sGC/PKG-ROCK signaling pathway **b**. **c** Luciferase activity displayed by the indicated serial 5′ promoter deletions in untreated (control) or DETA/NO-treated transfected dNSC34s. NO-induced changes, in folds relative to the mean value for untreated (control) condition, are presented on the right plot. **d** Top, sequence of the human p11 promoter from −96 to +7. A regulatory AP-2/Sp1-binding site is highlighted (blue). G/C to T site-directed mutations are also shown (Mut 1–Mut 5). Middle, as in **c** for the −96 to +89 segment of the promoter including each of mutations. The −70 to +89 segment was used as a NO-unresponsive control. As a negative control, mutation of the AP-2/Sp1-binding site downstream −70[29] did not alter NO-responsiveness. Bottom, NO-responsive GC-box element identified by site-directed mutagenesis. **e–g** EMSAs obtained by incubating $^{32}$P-labeled −96 to +89 wild-type or Mut 5 probes with three independent nuclear protein extracts (1, 2, 3) from dNSC34s or in the absence of nuclear extracts (−) **e**. Competition assays by co-addition of canonical sequences for Sp1 and AP-2 at the indicated concentrations **f**, or super-shift analysis with antibodies (2 µg µl$^{-1}$) against Sp1 and/or Sp3 **g**. **h** Time course of changes in mRNA$_{Sp1}$ and mRNA$_{p11}$ expression in SMNs$^{wt}$ incubated with the stated drugs beginning at 6 DIV relative to that obtained from untreated SMNs$^{wt}$ at the respective time points. DETA/NO (0.1 mM) for 24 or 72-h did not affect mRNA levels. DETA (1 mM, 72 h) was a negative control. $n = 3$ independent experiments. **i** Levels of mRNA$_{Sp1}$ and mRNA$_{p11}$ in SMNs$^{wt}$ that received the indicated treatments for 48 h. Error bars, SEM. $*p < 0.05$, $**p < 0.01$, $***p < 0.001$; n.s., not significant; by Student's $t$-test **c**, **d**, one-way ANOVA with post hoc Holm–Sidak method **a**, **b**, **h**, **i**. **j** The schematic represents the molecular pathway subserving p11 upregulation under RNS. Source data are provided as a Source Data file

might be due to different embryonic origin (spinal cord vs. brainstem), developmental stage (embryonic vs. neonatal), and/or experimental preparations.

Several findings support causality between RNS and Sp1–p11–TASK1 dysregulation. First, Sp1 expression and its DNA-binding activity increased in rat hippocampus exposed to excitotoxic insults[50,51]. Second, nNOS induction preceded MN death in ALS and after nerve injury[6]. Third, DETA/NO (0.1 mM) did not alter human p11 promoter activity or Sp1–p11 expression. However, DETA/NO (1 mM), which releases pathological concentrations of NO[6,24,25], increases MN IME by inhibition of TASK-like channels, upregulates Sp1–p11, and stimulates human p11 promoter via at least a Sp1-binding GC-box. Consistently, NO upregulated p11 in human epithelial cells[23] and promoted Sp1-binding activity[52–54]. Fourth, Sp1-interfering agent Mit-A fully avoids RNS-triggered stimulation of the minimal NO-responsive element of the human p11 promoter. Fifth, inhibition of endogenous NO synthesis attenuates the Sp1–p11 upregulation and the TASK-dependent increase of MN IME in the nerve injury model. Therefore, Sp1–p11–TASK1 dysregulation downstream of RNS seems a key event in a positive feedback loop leading to excitotoxic degeneration (Fig. 10b).

Comparison of the timing of NOS-p11 upregulation after injury[30] and in the ALS model[6], suggests that early p11 induction was RNS-unrelated in the two models (Fig. 10b). Thus, early slight p11 upregulation after nerve injury (1–3 days) and in the ALS model (early-presymptomatic) precede NOS induction after nerve injury (3–5 days)[30] and in SOD1-G93A mice (late-pre-symptomatic)[6]. Furthermore, given that NOS induction occurs in MNs and glial cells in early-symptomatic SOD1-G93A mice preceding MN loss[6], whether p11 expression at this stage is coupled to NO from MN and/or glia origin merits further investigation (Fig. 10b). Sp1 ubiquity and broad co-expression of p11 and TASK1 all along the CNS[17,18], suggest the need for further investigation on this triad to ascertain feasible biomarkers for diverse psychiatric and neurological disorders. One of the three, p11, has recently been proposed as a biomarker for monitoring the severity of Parkinson's disease[55].

The Sp1-binding inhibitor Mit-A is neuroprotective in the ALS mouse model. The oral lower dose we administered (30 µg kg$^{-1}$ day$^{-1}$) is in the lower range for its intravenous application in humans[56–58]. We would expect that concentration of Mit-A that might be reached within the organism after oral administration was lower than that obtained with equivalent intravenous doses in chemotherapy. In this line, characteristic side-effects after intravenous Mit-A administration (bleeding, behavioral alterations, and weight loss)[56–58] were absent in treated mice. Strikingly, beneficial outcomes of Mit-A treatment on SOD1-G93A mice were similar to those achieved under treatment with Riluzole, the most used drug for ALS patients in clinical settings. Currently, Mit-A is the subject of clinical trials for several cancers (http://www.ClinicalTrials.gov; Identifiers: NCT02859415, NCT01624090), thus supporting a renewed interest in the clinical use of this kind of compounds. The present results along with the emergence of Mit-A analogs with higher activity and less toxicity[59–61] encourage the use of these drugs in clinical trials for neurodegenerative disorders[60] such as ALS.

In summary, RNS-triggered induction of Sp1–p11 decreases background K$^+$ currents and switches neuron to a more vulnerable configuration in neurodegenerative disorders. In contrast, a reduction in Sp1–p11 places neurons in a more resistant state against degeneration due to an increase in the functional expression of TASK1. The broad expression of Sp1–p11–TASK1 throughout the organism suggests that interplay between these proteins might not only be circumscribed to the CNS but may also impact functioning of the immune, cardiovascular, respiratory, digestive, excretory, and reproductive systems, both in health and in sickness[62–64]. Indeed, dysregulation of either Sp1, p11, or TASK1 has been independently reported in multiple pathological conditions, including inflammatory processes and cancer, affecting different organs and tissues.

## Methods

**Animals.** Neonatal (P3–P9) wistar rats of either sex, young (1–2-month-old) and adult (>2-month-old) male mice, either CD1, SOD1-G93A (Jackson Laboratory, Bar Harbor, ME, USA) and Non-Tg littermates; and, CD1, C57BL/6J, task1$^{-/-}$, task3$^{-/-}$, and SOD1-G93A pregnant mice (12.5 days gestation). All these animals were provided by the local Animal Supply Services (SEPA, University of Cadiz). Animal care and handling followed the guidelines of the European Union Council (2010/63/EU, 86/609/UE) on the use of laboratory animals. Experimental procedures were approved by the local Animal Care and Ethics Committee (University of Cadiz, Cadiz, Spain) and the Ministry of Agriculture, Fisheries and Rural Development (Junta de Andalucía, Spain). task1$^{-/-}$ and task3$^{-/-}$ mice, a gift from Dr. Douglas A. Bayliss (University of Virginia, VA, USA), became stablished colonies and were bred in the local Animal Supply Services. Animals were individually housed in cages with water and food pellets available ad libitum, at 21 ± 1 °C, with a 12-h light/dark cycle. Neonatal animals were housed with their mother. Efforts were made to minimize the number of animals used and their suffering. All surgical processes were carried out under aseptic conditions.

**Electrophysiology.** Whole-cell patch-clamp recordings were performed in HMNs from coronal brainstem slices (P6–P9 rats) or cultured SMNs (4 DIV)[10,65–67]. In summary, neonatal rats were decapitated under anesthesia by hypothermia (10–15 min at 4 °C). Brainstems were rapidly removed and dissected in artificial cerebrospinal fluid (aCSF) enriched with sucrose at 4 °C (in mM: 26 NaHCO$_3$, 10 glucose, 3 KCl, 1.25 NaH$_2$PO$_4$, 2 MgCl$_2$, and 218 sucrose), and bubbled with 95% O$_2$ and 5% CO$_2$. Transverse slices (300–400 µm-thick), obtained using a vibroslicer (NVSL; WPI), were transferred to normal oxygenated aCSF (in mM: 26 NaHCO$_3$,

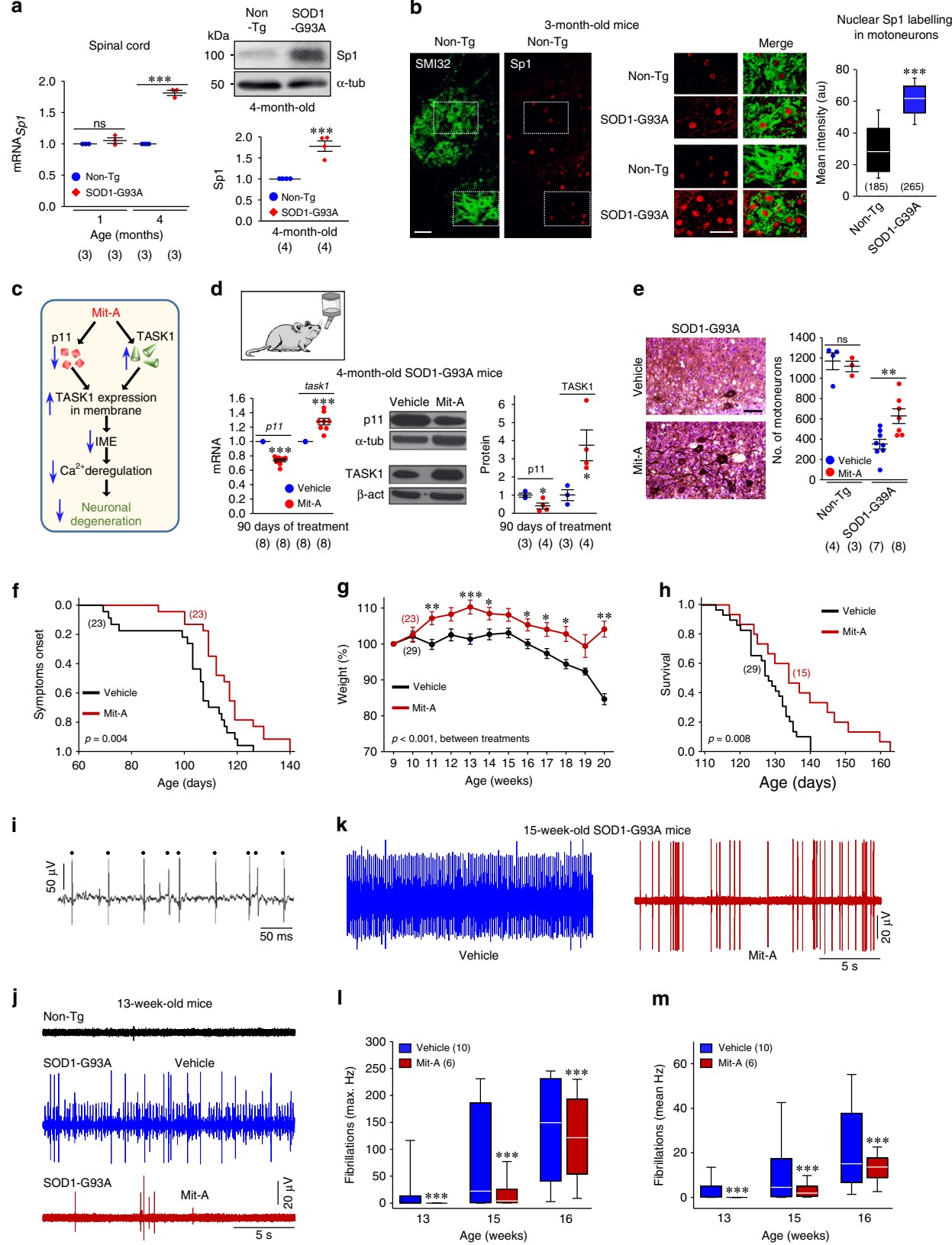

10 glucose, 3 KCl, 1.25 NaH$_2$PO$_4$, 6 MgCl$_2$, 130 NaCl, and 0.5 CaCl$_2$) and stabilized at ~37 °C for 1 h. Slices were then transferred to a recording chamber for whole-cell patch-clamp recordings or to an incubation chamber for treatment with different drugs before HN microdissection. SMN cultures were prepared from embryonic day 12.5 (E12.5) mouse spinal cords, as described below.

Recordings of HMNs and SMNs were performed under constant perfusion (~3–4 ml min$^{-1}$) with normal oxygenated aCSF at 31 °C. Recordings were obtained from motoneuron somata, using a Nikon (Tokyo, Japan) Eclipse CFI60 microscope equipped with infrared differential interference contrast. Patch electrodes (1.5–3 MΩ resistance) were filled with the following internal solution (in mM): 17.5 KCl,

**Fig. 7** Neuroprotective effects of the Sp1-interfering agent Mit-A in the SOD1-G93A mouse model of ALS. **a** mRNA (left) and protein (right) levels for Sp1 in the spinal cord of SOD1-G93A mice and their Non-Tg littermates at the indicated ages. **b** Confocal images showing Sp1-immunolabeled nuclei of SMI32-positive lumbar MNs from 3-month-old Non-Tg and SOD1-G93A mice. Boxed areas in low magnification images for the Non-Tg condition are displayed at higher magnification together with similar areas obtained from transgenic mice. Box plot represents the mean intensity of nuclear Sp1-labeling of Non-Tg or SOD1-G93A MNs. **c** Mechanism of action by which Mit-A might protect neurons against excitotoxic degeneration. **d** Daily administration of Mit-A (30 μg kg$^{-1}$ day$^{-1}$) in the drinking water, beginning at P30, dysregulated mRNA (left) and protein (right) levels of p11 and TASK1, in the lumbar spinal cord of SOD1-G93A mice at P120. **e** Number of SMI32-positive MNs (left), in the L3–L5 spinal cord segment of SOD1-G93A mice and their Non-Tg littermates, treated with Mit-A or vehicle. Scale bars, 50 μm. **f–h** Cumulative probability curves of symptoms onset **f** and survival **h**, and time course of body weight **g**, against the age of SOD1-G93A mice receiving the indicated treatments. Mit-A treatment as in **d**. **i–k** Representative electromyographic recordings from the gastronecmius muscle of Non-Tg and SOD1-G93A mice at weeks 13 **i, j** and 15 **k**. Transgenic mice received treatments as in **f–h**. In **i**, fibrillation potentials (dots) were recorded by needle electromyography from a 13-week-old transgenic mouse. These spontaneous muscle discharges were absent in Non-Tg mice **j**. **l, m** Maximal (**l**) and mean (**m**) frequency of fibrillations in SOD1-G93A animals at the indicated weeks treated with vehicle or Mit-A. Number of independent samples is in parentheses. Error bars, SEM. *$p < 0.05$, **$p < 0.01$, ***$p < 0.001$; n.s., not significant; by Student's $t$-test **a, b, d, e, l, m** or two-way ANOVA with post hoc Holm–Sidak method **g**. Statistic outputs determined by Log-Rank test, Kaplan–Meier analysis **f, h** or two-way ANOVA **g** between treatments are stated on plots. Source data are provided as a Source Data file

122.5 K-gluconate, 9 NaCl, 1 MgCl$_2$, 10 HEPES, 0.2 EGTA, 3 Mg-ATP, and 0.3 GTP–Tris at pH 7.4. Recordings were obtained and low-pass Bessel filtered at 10 kHz with a MultiClamp 700B amplifier. Data were digitized at 20 kHz with a Digidata 1332A analog-to-digital converter and acquired using pCLAMP 9.2 software (Molecular Devices, Foster City, CA). Analysis was only performed for recordings with access resistance between 5 and 20 MΩ. Recordings were discarded if access resistance changed by >15% during the trial. Series resistance was routinely compensated 65–75%. The pipette offset potential was neutralized just before motoneurons were patched. Recordings were not corrected for liquid junction potential.

In the current-clamp configuration, resting Vm and $R_N$ were measured for the assessment of MN IME. $R_N$ was calculated from the current–voltage ($I–V$) plots obtained by injecting a series of depolarizing and hyperpolarizing current pulses (0.5 s; −0.2 to 0.2 nA). In Fig. 1b only the hyperpolarizing square pulses are exemplified. The resulting data points were then fitted with a regression line, and $R_N$ was estimated as the slope of the lines. Voltage-clamp recordings were all carried out in presence of tetrodotoxin (TTX, 1 μM; Tocris Cookson, Bristol, UK). Initially, the holding current ($I_{holding}$) required to keep Vm at −65 mV was measured (Supplementary Fig. 8a). $I–V$ relationships (Supplementary Fig. 8e, right) were obtained by applying voltage steps in 5 mV increments ranging from −50 to −120 mV from a baseline holding potential of −65 mV. In this case, the instantaneous component was measured ~10 ms after the onset of the step (Supplementary Fig. 8b, e, triangles). In the experiments designed to evaluate TASK-like pH-sensitive currents, MNs were sequentially recorded at varying pH levels (pH 6.2, 7.2, and 8.2). Extracellular pH was adjusted adding either HCl or NaOH to aCSF. TASK-dependent currents (pH-sensitive currents) were obtained by subtraction of the $I–V$ relationship obtained at pH 8.2 minus that measured at pH 6.2 (Fig. 5h, Supplementary Figs. 8b and 9b).

**Quantitative real-time reverse transcriptase PCR (qRT-PCR).** Total RNA was extracted from medulla, spinal cord, microdissected HNs, dNSC34s, or SMNs (100,000 cells per well) using the TRIsure Isolation Reagent (BioLine). This procedure allowed the isolation of total RNA, DNA, and protein fractions from a single sample. Subsequent treatment with the RNase-free DNase set (Invitrogen), following the manufacturer recommendations, was performed to reduce DNA contamination. The concentration and purity of RNA samples were determined by spectrophotometry at 260 and 280 nm. Retrotranscription (RT) was done using random hexamers, 500 ng of total RNA as template and iScriptTM cDNA Synthesis Kit (Bio-Rad). For real-time RT-PCR, each specific gene product was amplified with the MiniopticonTM System (BIO-RAD), using iQ SYBR Green Supermix (Bio-Rad). The PCR primers sequences are stated in the Supplementary Table 1. The cDNA levels for the different samples were determined using the ΔΔCT method, being *gapdh* the housekeeping gene. All analyses were performed in triplicate, with each experiment repeated at least twice.

**Western blot analysis.** Tissue samples from the medulla, spinal cord or microdissected HNs were homogenized in TRIsure, as above, or in lysis buffer [50 mM Tris/HCl, pH 7.4, 1% (v/v) Triton X-100, 0.5% (w/v) sodium deoxycholate] supplemented with protease (Sigma-Aldrich) and phosphatase (Pierce) inhibitors using a 1 ml syringe. dNSC34s cells and SMNs (100,000 cells per well) were rinsed with ice-cold PBS and immediately transferred to the lysis buffer with protease and phosphatase inhibitors. Protein concentrations were determined by Bradford-protein assay (BioRad) or by micro BCA protein assay kit (Thermo Scientific). Proteins were separated by SDS–PAGE and blotted to polyvinylidene difluoride membrane (Amersham). Membranes were blocked with BSA 5%, 0.1% Tween for 1 hr at room temperature and subsequently blotted with specific antibodies against pATF-1 (1:200, Santa Cruz Biotechnology, Cat# sc-7978, RRID:AB_2086020) and p11 (1:1000, R&D systems, Cat# AF2377, RRID:AB_2183469) developed in goat,

HA (12CA5) (1:50, Sigma-Aldrich, Cat# 11583816001, RRID:AB_514505), RhoA (1:200, Santa Cruz Biotechnology, Cat# sc-418, RRID:AB_628218), β-actin (1: 2,500,000, Sigma-Aldrich, Cat# A5441, RRID:AB_476744) and α-tubulin (1:1,000,000, Sigma-Aldrich, Cat# T9026, RRID:AB_477593) developed in mouse, and Sp1 (1:1,000, Abcam, Cat# ab124804, RRID:AB_10974611), TASK1 (1:200, Alomone Labs, Cat# APC-024, RRID:AB_2040132; 1:200, Sigma-Aldrich, Cat# P0981, RRID:AB_260876) or TASK3 (1:200, Alomone labs, Cat# APC-044, RRID: AB_2039953) developed in rabbit. Proteins were visualized by using horseradish peroxidase-conjugated secondary antibodies (Pierce) and the enhanced chemoluminiscence system (Amersham). Analysis was performed using the ImageJ 1.48 v software from the National Institutes of Health. Uncropped and unprocessed scans of all blots are supplied in the Source Data file.

**Immunoprecipitation.** Immunoprecipitation assays were performed using Pierce® co-immunoprecipitation kit (Thermo Scientific) following manufacturer's instructions with subtle modifications. Briefly, the brain of an adult mouse (CD1) was mechanically homogenized in immunoprecipitation assay buffer (5 μl/mg of tissue; 50 mM Tris, 150 mM NaCl, 1 mM EDTA, 1% NP40, and 5% glycerol, pH 7.4 supplemented with a cocktail of protease inhibitors) and cleared by centrifugation. To adjust the protein concentration in samples prior to immunoprecipitation, it was measured using a BioRad protein assay kit according to the manufacturer´s instructions. First, Pierce spin columns containing 25 μl resin were incubated with 10 μg of anti-p11 antibody (R&D systems) for 3 h in a rotator at room temperature. As negative control, other spin columns were incubated with 10 μg of an irrelevant antibody (anti-sheep IgG, Sigma-Aldrich) in the same conditions. The homogenates were diluted with immunoprecipitation assays buffer to have a final volume of 300 μl per column, and incubated with the resins overnight in a rotator at 4 °C. Resin centrifugation steps were performed at 2000×g for 45 s. Thereafter, a volume of 60 μl of the elution buffer was passed through every resin to collect p11 together with their associated-proteins in a first elution (E1) fraction. A second elution (E2) of 60 μl was collected to ensure that most p11 and p11-associated proteins were rescued from the resin. For protein denaturalization 100 mM DTT 5× sample buffer was added to eluted fractions which were subsequently heated at 95 °C for 5 min. Finally, fractions were subjected to electrophoretic and immunoblot analysis for p11 and TASK1 detection.

**Primary cultures of SMNs and survival assays.** Primary cultures of MNs were prepared from the spinal cord of mouse embryos at 12.5 days of gestation (E12.5), following an established protocol[65,68]. Briefly, desiccated ventral cords were first chemically (0.025% trypsin in glucose–HEPES buffer solution supplemented with 20 i.u. ml$^{-1}$ penicillin and 20 mg ml$^{-1}$ streptomycin) and mechanically dissociated to successively be collected under a 4% bovine serum albumin cushion. Centrifugation (10 min, 520×g) on an Iodixanol density gradient (OptiPrep, Axis-Shield, Oslo, Norway) allowed to isolate the largest cells, which were once more centrifuged through a bovine serum albumin cushion. Isolated cells were pooled in a tube containing culture medium and plated at various densities depending on the type of experiment. Cultured SMNs were clearly identified by SMI32 immunofluorescence or by morphological criteria. SMNs were cultured in Neurobasal medium (Gibco, Invitrogen, Paisley, UK) supplemented with B27 (Gibco; Invitrogen), horse serum (2% v/v), L-glutamine (0.5 mM), and 2-mercaptoethanol (25 μM; Sigma-Aldrich), and a cocktail of neurotrophic factors (NTFs): 1 ng ml$^{-1}$ brain-derived neurotrophic factor, 10 ng ml$^{-1}$ glial cell-line-derived neurotrophic factor, 10 ng ml$^{-1}$ ciliary neurotrophic factor, and 10 ng ml$^{-1}$ hepatocyte growth factor (PreProtech, London, UK). Isolated SMNs were plated either in four-well tissue culture dishes (Nunc, Thermo Fisher Scientific, Roskilde, Denmark) for survival experiments (2000 SMNs per well) or on 24-mm Corning glass coverslips (Corning, NY) placed in 35-mm culture dishes for electrophysiological experiments (30,000 SMNs per well), Ca$^{2+}$ imaging (30,000 SMNs per well) or

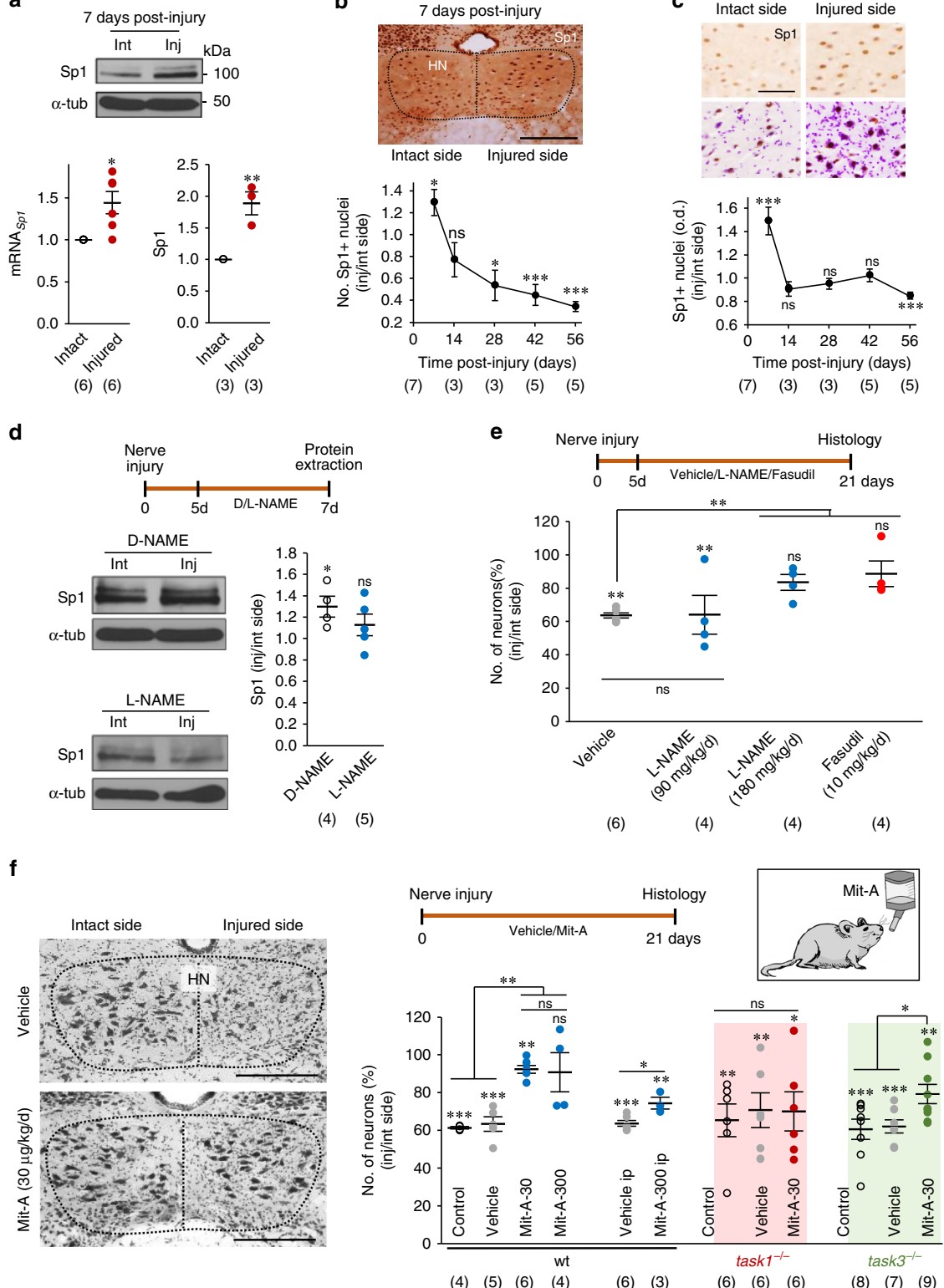

immunocytochemistry (15,000 SMNs per well). For survival evaluation of SMNs, the cells received complete medium on the seeding day, either containing or lacking (no NTFs) the combined NTF cocktail. One and four days after seeding, culture medium with NTFs was replaced with the same medium supplemented with the experimental conditions (see treatments below). To assess neuronal survival, the number of large phase-bright cells with long neurites in the total area of triplicate wells for each condition was counted 72 h after the last treatment. Data were obtained from at least three independent experiments. Individual symbols in figures represent a single well. The mean number of surviving cells (in percent) was normalized taking the "NTF medium alone" treatment as 100%.

**Ca²⁺ imaging**. For intracellular $Ca^{2+}$ measures, isolated SMNs, plated on laminin-coated coverslips, were washed in Krebs solution (140 mM NaCl; 5 mM KCl; 2 mM CaCl₂; 1 mM MgCl₂; 10 mM HEPES; and 11 mM glucose, pH 7.4). Afterwards, SMNs were loaded with a non-ratiometric or single wave dye that measures free cytosolic $Ca^{2+}$. For that purpose, coverslips were incubated (30 min, 37 °C, darkness) in Krebs solution supplemented with Fluo-4 AM (1 μM, Molecular Probes, Invitrogen) and pluronic acid (0.02%). Confocal images were obtained by means of a direct Olympus FV1000 confocal laser scanning microscope using a ×40 water immersion objective. Fluo-4 AM was excited at a wavelength of 488 nm and the emission fluorescence at a wavelength of 518 nm was acquired for offline analysis.

**Fig. 8** TASK1-dependent neuroprotective effects of Mit-A in the traumatic model of MN degeneration. **a–c** Effects of resection of a 2-mm-segment of the XIIth nerve on $mRNA_{Sp1}$ and Sp1 levels **a**, number of Sp1-immunoreactive nuclei **b**, and optical density of nuclear staining **c** in the HN of the injured (inj) versus the intact (int) side at the stated days post-lesion. Coronal sections immunolabelled for Sp1 **b**, **c**, some of them counterstained by Nissl (**c**, bottom), obtained 7 days after nerve injury. At least 50 Sp1-immunoreactive nuclei per animal and side were analyzed in **c**. **d** Immunoblots for Sp1 from HNs of the intact and injured sides at 7 days post-lesion in mice treated with L-NAME (180 mg kg$^{-1}$ day$^{-1}$, i.p.) or its inactive stereoisomer D-NAME from day 5th after nerve resection. *gapdh* and α-tub/β-act were internal controls for *q*RT-PCR and western blotting, respectively. **e**, **f** Plots represent the number of neurons (in percent) in the injured versus the intact side taken as 100%. In **e**, administration of vehicle and drugs were intraperitoneal (i.p.). Mit-A was orally administered except when indicated as i.p. applied in **f**. Nissl stained coronal sections obtained from mice sacrificed 3 weeks after nerve injury receiving indicated treatments are shown to the left in **f**. Dotted lines delimit the HN in each section. Animals treated with vehicle in **e** and vehicle i.p. in **f** are the same. **f** Control, untreated animals; vehicle, drinking water supplemented with 1% sucrose; Mit-A-30 (30 µg kg$^{-1}$ day$^{-1}$); Mit-A-300 (300 µg kg$^{-1}$ day$^{-1}$). Note that Mit-A was neuroprotective for *task3*$^{-/-}$, but not for *task1*$^{-/-}$ mice. Scale bars, **b** 350 µm; **c** 100 µm; **f** 500 µm. Number of independent samples is indicated in parentheses. Error bars, SEM. *$p < 0.05$, **$p < 0.01$, ***$p < 0.001$; n.s., not significant; by paired (injured vs. intact side within a treatment, **a–f**), unpaired (comparison between two groups, **f**) Student *t*-test or one-way ANOVA with post hoc Holm–Sidak method (comparison between more than two treatments, **e**, **f**). Source data are provided as a Source Data file

Time-lapse images were taken at 5-s intervals for at least 50 min. All experiments were performed at 31 °C. For the analysis of Ca$^{2+}$ dynamics, regions of interest including overall SMN somata were used and the background was subtracted from each selected SMN. SMN Ca$^{2+}$ concentrations were estimated using the maximal fluorescence equation:

$$[Ca^{2+}]_i = K_d \times (F/F_{max} - 1/R_f)/(1 - F/F_{max})$$

where $K_d$ is the Fluo-4 AM dissociation constant, that in our experimental conditions was 273.2 nM, and $R_f$ is the dynamic range of Fluo-4 AM for each analyzed SMN ($F_{max}/F_{min}$). $F$ was the recorded fluorescence. $F_{max}$ and $F_{min}$ values were determined, at the end of each experiment, by exposing SMNs to the Ca$^{2+}$ ionophore ionomycin (10 µM) and subsequent addition of the Ca$^{2+}$ chelator, EGTA (15 mM), respectively. $K_d$ and $R_f$ values for our experimental conditions and cell types were estimated in vitro by using standard methods[69].

**Nerve injury models.** Rat pups (P3) and adult mice (>2-month-old) were subjected to XIIth nerve crushing or partial resection, respectively. Both modalities of nerve injury induce neuronal hyperexcitability preceding motoneuron loss mainly by glutamate-induced excitotoxicity[10]. Anesthetized animals (1.5–3% isoflurane mixed with 100% O$_2$) received intramuscularly atropine (0.2 mg kg$^{-1}$) and dexamethasone sodium phosphate (0.8 mg kg$^{-1}$). The absence of withdrawal reflexes was considered a signal of sufficient deep anesthesia. In P3 rats, the right XIIth nerve was isolated from the surrounding tissue and the common nerve trunk was thoroughly crushed with microdissecting tweezers, applied for 30 s (Fig. 5f, Supplementary Fig. 9f). In adult mice, following nerve isolation, a 2-mm segment was surgically removed (Fig. 4c). Both types of lesion were performed just proximal to the bifurcation of the main XIIth nerve trunk into lateral and medial branches. In adult mice, axonal regeneration, from the proximal stump of the injured nerves, was minimized by a tied suture thread (3/0). The incision was sutured and cleaned with an aseptic solution (povidone-iodine). The animals were sacrificed at different time points after nerve injury, depending on the experimental paradigm. All animals received one post-operative injection of penicillin (20,000 i.u. kg$^{-1}$; i.m.) to prevent infection and pirazolone (0.1 mg kg$^{-1}$; i.m.) for post-surgery analgesia.

**Histological analysis.** Animals, under deep anesthesia (7% chloral hydrate), were intraventricularly injected with heparin and immediately perfused transcardially with phosphate buffered saline (PBS), followed by 4% paraformaldehyde (PFA) in 0.1 M phosphate buffer (PB), pH 7.4, at 4 °C. The medulla and/or lumbar spinal cords (L1–L3 or L3–L5) were rapidly dissected and postfixed for 2 h in 4% PFA. Cryoprotection was performed by immersion in 30% sucrose in 0.1 M PB (4 °C, overnight). Serial coronal sections (30 µm-thick), obtained by means of a microtome, were kept at −20 °C in a cryoprotectant solution (glycerol/PBS, 1:1 v/v) until processing.

For immunohistochemistry, after PBS washing, sections were immersed in a blocking solution [2.5% (w/v) bovine serum albumin, 0.25% (w/v) sodium azide, and 0.1% (v/v) Triton X-100 in PBS] for 30 min, followed by incubation (4 °C, overnight) with the polyclonal primary antibodies against SMI32 (1:8,000; Covance Research Products Inc, Cat# SMI-32R-500, RRID:AB_509998), GFAP (1:250; Santa Cruz Biotechnology, Cat# sc-58766, RRID:AB_783554) developed in mouse, Sp1 (1:50, Abcam, Cat# ab124804, RRID:AB_10974611), TASK1 (1:200, Alomone Labs, Cat# APC-024, RRID:AB_2040132) developed in rabbit, and p11 (1:50, R&D systems, Cat# AF2377, RRID:AB_2183469) developed in goat. Next, the tissue was rinsed with PBS and incubated for 1 h at room temperature with Cy2-conjugated, Cy3-conjugated, and/or Cy5-conjugated, or biotinylated anti-goat, anti-mouse, and/or anti-rabbit IgGs (1:400; Jackson ImmunoResearch Laboratories) as secondary antibodies. When applicable, biotin was detected by means of the avidin–biotin–peroxidase system (Pierce, Rockford, IL, USA) using as chromogen 3,3-diaminobenzidine tetrahydrochloride. Sections were mounted on slides, dehydrated, covered with DePeX, and visualized under light microscopy. Sections

were analyzed using an Olympus IX81 inverted microscope for light microscopy or an Olympus FV1000-MPE confocal microscope for fluorescence microscopy (Olympus, Japan). Images were acquired through a z-plane in which maximum antibody penetration was evidenced. The pinhole opening was 1 Airy unit. For comparison between different experimental conditions, acquisition setting was kept identical. Animals and tissue were processed in parallel. Images were processed for background subtraction to obtain the maximum dynamic range of greyscale (from 0 to 250) and analyzed using the software provided by Olympus. In all cases, the area delimiting p11-positive or TASK1-positive somata or Sp1-immunoreactive nuclei were manually traced, and for each cell, the mean optical density (o.d.) or fluorescence intensity was measured. MNs were acquired in sections obtained from at least three animals per condition. Treatments were blinded for researcher who performed measures.

For Nissl or neutral red staining, brainstem sections containing the HN were mounted on slides, re-hydrated, and immersed in Nissl (10 min, 0.5% cresyl violet) or red neutral (5 min, 1% neutral red in 0.2 M acetate buffer, pH 3.3) staining solutions. The sections were then rinsed with distilled water, dehydrated in ethanol, cleared in xylene, and covered with DePeX.

The number of neurons with a well-defined nucleus was quantified in one of each three serial sections through the whole rostrocaudal extent of the HN or in one of each three–six serial sections through the L1–L3 or L3–L5 segments of the lumbar spinal cord. SMI32-identified MNs were counted in 10 horizontal or 30 coronal sections of the lumbar spinal cord from 4-month-old mice. All quantifications were carried out by researchers blinded to the lesion side, treatment and/or genotype groups.

SMNs, plated in four-well tissue culture dishes (15,000 cells per well), were processed by double-immunostaining for p11 and TASK1. At 3–4 DIV, cRNA-treated and siRNA$_{p11}$-treated SMN cultures were fixed in 4% PFA (15 min, room temperature) and 100% methanol (10 min, 4 °C) and incubated overnight at 4 °C with polyclonal antibodies against p11 (1:100) and TASK1 (1:100) developed in goat and rabbit, respectively. SMNs were further incubated with goat and rabbit anti-IgG secondary antibodies conjugated with Cy3 and Cy2 (1:200; Jackson ImmunoResearch Laboratories), respectively. Dishes were analyzed using a Leica (Nussloch, Germany) confocal fluorescence microscope. Images were acquired through the z-plane containing the highest density of large diameter SMNs and well defined neurites as above. The pinhole opening was 1 Airy unit. Acquisition settings were identical for cRNA and siRNA-treated cultures, which were processed and acquired in parallel. Images were captured and processed for background subtraction to obtain the maximum dynamic range of intensities (from 0 to 250) and analyzed using the software provided by Leica. In all cases, the area delimiting somata and neurites was automatically traced by applying a macro, and for each image, the integrated and mean immunofluorescence intensities of SMN cell bodies and neurites were obtained in both the cRNA-treated and siRNA$_{p11}$-treated cultures.

**Lentiviral constructions.** All neuron-specific LVVs used in this work were designed, constructed, and produced under the supervision of Dr. S. Kasparov (Bristol University, Bristol, UK) during a research stay of G.R.B. in his lab. All the lentiviral constructions were made based on the vector pTYF-mCMV/SYN. This system uses a transcriptional amplification strategy (TAS) to bidirectionally enhance the transcriptional activity of human synapsin-1 (SYN) promoter, by joining upstream a minimal core promoter (65 bp) derived from the human cytomegalovirus (mCMV) in the opposite orientation. This construction creates a compact TAS-amplified neuron-specific promoter useful for the development of powerful gene expression systems[70]. Two cloning steps were necessary to generate the LVV-shRNA$_{lacz}$ plasmid. First, the shRNA$_{lacz}$ sequence (5′-AAATCGCTGA CTTGTGTAGTCGTTTTGGCCACTGACTGACAACCACACATCAGCGAT TT-3′) was cloned into the vector pcDNA6.2-WemGFP (Invitrogen) to obtain the shRNA$_{lacz}$-GFP (green fluorescent protein) plasmid. Finally, a PCR product containing the shRNA$_{lacz}$-GFP was cloned into the vector pTYF-CMV/SYN[70] to

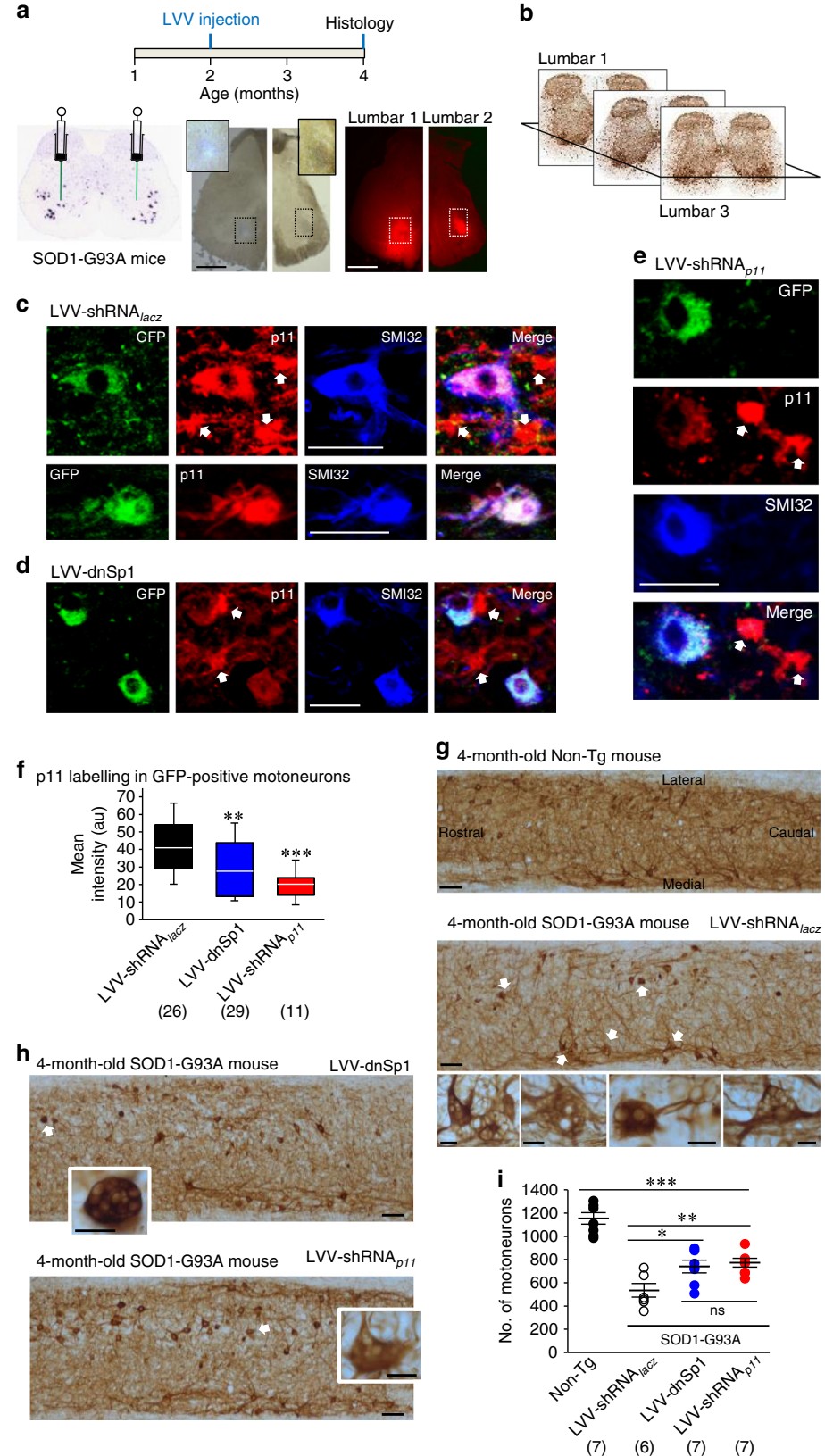

**f** p11 labelling in GFP-positive motoneurons

**g** 4-month-old Non-Tg mouse

**h** 4-month-old SOD1-G93A mouse    LVV-dnSp1

**i**

generate the final lentiviral plasmid. This same procedure was also followed to obtain the LVV-shRNA$_{p11}$ plasmid using the shRNA$_{p11}$ sequence (5′-TAGTCA TTGCATGCAATGGTGGTTTTGGCCACTGACTGACCACCATTGTGCAATG ACTA-3′).

A different strategy was used to generate the LVV-dnSp1 vector. The full cDNA sequence for the Sp1 dominant negative[71] corresponding with the C-terminal

domain of the consensus sequence (622–788; a gift from Dr. Johann Rotheneder, Medical University of Vienna, Austria) was cloned into the pIRES2-eGFP. A PCR product containing the dnSp1-eGFP sequence was cloned into pTYF-mCMV/SYN plasmid to generate the final LVV-dnSp1 construction.

Transient co-transfection of HEK293FT cells with shuttle plasmids, the packaging vector pNHP, and the envelope plasmid pHEF-VSVG was performed

**Fig. 9** Cell-type specific p11 knockdown is neuroprotective for MNs in the SOD1-G93A mouse model of ALS. **a** Bilateral microinjections of neuron-specific LVVs were performed in the ventral horn of 2-month-old SOD1-G93A mice (see schematic on top). Reference section (from the Allen Brain Atlas) illustrating injection protocol (left) is the same than in Fig. 3a. Bright-field (middle panels) and fluorescence (right panels) images of the same coronal sections showing injection places (boxed areas) of Chicago sky blue at the same coordinates than LVVs. Insets, details of injection places. **b** Representative acetyl-cholinesterase (AChE)-immunostained L1–L3 sections (from the Allen Brain Atlas) showing horizontal plane sectioning performed to analyze LVV-treated mice. **c–e** Confocal images showing p11 immunolabelled GFP-positive SMI32-identified lumbar MNs from 4-month-old SOD1-G93A mice after intraspinal microinjection of LVV-shRNA$_{lacz}$ **c**, LVV-dnSp1 **d** or LVV-shRNA$_{p11}$ **e** performed in 2-month-old animals. Note that while glial-like structures (arrows) show similar p11 labeling intensity in all conditions, transduced MNs in **d** and **e** display considerably less intensity than in **c**. **f** Mean fluorescence intensity (arbitrary units, a.u.) of p11 staining in GFP−/SMI32-positive lumbar MNs. **g**, **h** Representative SMI32-immunostained ventral horn (L1–L3) horizontal sections obtained from 4-month-old mice of the stated genotypes receiving indicated treatments. High magnification photomicrographs of some vacuolated MNs (arrows) are shown. Scale bars, **a** 500 μm; **c–e** 50 μm; **g**, **h** 20 and 100 μm, high and low magnification images, respectively. **i** Number of MNs in the L1–L3 spinal cord segment of 4-month-old mice with the indicated genotypes that received the stated treatments. Number of independent samples is in parentheses. Error bars, SEM. *$p < 0.05$, **$p < 0.01$, ***$p < 0.001$; n.s., not significant; by one-way ANOVA with post hoc Holm–Sidak method. Source data are provided as a Source Data file

to produce the LVV plasmid system derived from HIV-1 used for this study[70]. The virus-containing medium was harvested 48 or 72 h after transfection and filtrated using a 0.45 μm filter (Millipore). Filtrated medium was carefully added on the top of a 20% sucrose solution and subsequently centrifuged at 75,000×$g$ for 2 h. After removal of the supernatant, the pellet was re-suspended in PBS.

Finally, the titer of the virus (>10$^9$ iu μl$^{-1}$) was determined using the ELISA kit Lenti-X p24 rapid Titer (Clontech) which measures the amount of lentiviral capsid protein p24. Cells were transduced with the different LVVs at a five times multiplicity of infection.

**Plasmid constructions, deletions, and site-directed mutagenesis.** The promoter region of the p11 human gene (about 1.4 kb of the gene 5′ flanking region) was amplified using primers designed to add complementary overhangs in BglII-HindIII and the bacterial artificial chromosome (BAC) RP11-139D23 as genomic DNA template (developed by Dr. Kazutoyo Osoegawa, and acquired through the BACPAC Resource Center, Children's Hospital Oakland Research Institute, California, USA). PCR conditions used were 95 °C 5 min, (95 °C 30 s, 52 °C 30 s, 72 °C 1 min) × 35 cycles, and 72 °C 5 min for final extension. The PCR amplicon was purified using a PCR Purification Kit (Qiagen), digested with BglII-HindIII and cloned into the pGL4.10 luciferase Firefly reporter plasmid (Promega) in BglII-HindIII to generate −1436 + 89p11-Fir (pGL4.10-p11-Fir) construct. The human promoter sequence was confirmed by sequencing.

To narrow down NO-responsive regions in the promoter sequence, we performed serial 5′ deletions of the promoter by PCR, changing the 5′ sense primer. The common+89 HindIII 3′ primer was 5′-AAGCTTACCTTGGCCG AGGCGCGGCG-3′. Primers used for 5′ deletions of the p11 promoter were as indicated in Supplementary Table 1. The amplicons were cloned into pGL4.10 as above. When these regions were identified, we looked for transcription factors with ability to bind to the delimited promoter segments using a virtual transcription element search system. Site-directed mutagenesis of the transcription factors' binding sequences were then performed by PCR.

For site-directed mutagenesis, modified versions of the −96 fragment 5′ primer were used to develop PCR. The related primers used were as indicated in Supplementary Table 1.

**Luciferase reporter assay.** dNSC34 cells were transiently co-transfected with 5 μg of the different constructs cloned in pGL4.10-Fir and 0.5 μg of the wild-type Renilla luciferase control reporter vector pGL4.74-Ren (Promega) to determine the transfection efficiency, using the calcium phosphate precipitation method. 40,000 dNSC34s per well were transfected. After 24 h in the transfection medium, this medium was washed out and replaced by culture medium supplemented with diverse drugs. Finally, luciferase activity was measured using the Dual-luciferase Reporter Assay System (Promega). Promoter activity was measured in relative units of luciferase (RUL) of Firefly versus Renilla activities and normalized to the control.

Undifferentiated NSC34s (CELLutions Biosystems Inc., Toronto, Canada) were cultured in high glucose Dulbecco's modified Eagle's medium (DMEM), supplemented with 10% (v/v) fetal calf serum (FCS), 2 mM L-glutamine and 1% (v/v) penicillin/streptomycin. To induce maturation, NSC34s were plated at low density (4,000 cells cm$^{-2}$) in 1:1 DMEM plus Ham's F12 medium supplemented with 1% (v/v) FCS, 1% penicillin/streptomycin, and 1% modified Eagle's medium non-essential amino acids (Sigma-Aldrich)[65]. Differentiated phenotype of these cells was visually verified after 7–9 days in differentiating medium and transfections were carried out as above.

**Electrophoretic mobility shift assays.** Nuclear extracts were isolated from ~4 × 10$^6$ dNSC34 cells. dNSC34s were gently washed in PBS and collected by centrifugation in a microcentrifuge at 8,600×$g$. Cells were re-suspended in isotonic

buffer (10 mM Tris–HCl pH 7.5, 2 mM MgCl$_2$, 3 mM CaCl$_2$, 0.3 M sucrose, 1 mM DTT) plus a protease inhibitors cocktail (Sigma). Next, plasma membrane was disaggregated in a 10% NP40 solution and the total mix was centrifuged at 8,600×$g$ (30 min, 4 °C). Subsequently, the pellet was re-suspended in extraction buffer [20 mM Hepes pH 7.9, 1.5 mM MgCl$_2$, 0.42 M NaCl, 0.2 mM EDTA, 25% Glycerol (v/v)]. After further centrifugation (14,200×$g$, 10 min, 4 °C), the resultant supernatant, enriched in nuclear proteins, was aliquoted.

Double stranded probes were obtained by mixing equal volumes of both complementary primers (wild-type and Mut 5; Supplementary Table 1) at equimolar concentration (25 μM). We next placed the tube in a standard heat-block at 90–95 °C for 3–5 min followed by cooling at room temperature overnight. Then, the probes were 5′-labeled by a T4 Polynucleotide Kinase (Promega Gel Shift Assay System) using [γ$^{32}$P]-ATP (3,000 Ci mmol$^{-1}$ at 10 mCi ml$^{-1}$). Labeled probes were purified by spin-column chromatography (Microspin™ G-50 Columns, GE Healthcare).

Binding reactions between the nuclear extracts and radiolabeled probes were performed by incubating 1 μl of nuclear extract, 4 μl of binding buffer (50 mM Tris–HCl pH 7.5, 250 mM NaCl, 5 mM MgCl$_2$, 2.5 mM EDTA, 2.5 mM DTT, 20% Glicerol), and 1 μl of poly (dI-dC)/(dI-dC) (1 μg μl$^{-1}$ in 10 mM Tris–HCl, pH 7.5, 1 mM EDTA) in the presence or absence of competitor oligonucleotides or antibodies [anti-Sp1 (2 μg μl$^{-1}$), Abcam, Cat# ab124804, RRID:AB_10974611, or/ and anti-Sp3 (2 μg μl$^{-1}$), Santa Cruz Biotechnology, Cat# sc-13018, RRID: AB_2194471]. Nuclear extracts were pre-incubated for 1 h at 4 °C with antibodies or with oligonucleotides comprising the canonical sequence for Sp1 (5′-ATTCGATCGGGGGCGGGGCGAGC-3′; 1.75, 3.5 and 5.25 pmol μl$^{-1}$) or AP-2 (5′-GATCGAACTGACCGCCCGCGGCCCGT-3′; 5.25 pmol μl$^{-1}$) before adding the radiolabeled probe for 25 min at RT. Products were electrophoresed at 30 mA for 3 h on 4.8% polyacrylamide gels in high ionic strength buffer (50 mM Tris, 380 mM glycine, 2 mM EDTA, pH 8.5). Dried gels were analyzed by autoradiography [Personal Molecular Imager™ (PMI™) System, Biorad].

**Evaluation of disease progression in SOD1-G93A mice.** Transgenic males were received at pre-symptomatic stage (P30). Then, habituation sessions for rotarod and grip strength began at P45. One or two valid measures were obtained before cannula implantation for chronic oligonucleotide administration at P60 to discard surgical effects on motor performance. Only animals with no evident motor deficits at one week after surgery were included in the study. Researchers blinded for treatments did all behavioral studies.

Surgery to implant the cannula infusion system (Plastics One Inc., Roanoke, VA, USA) for intracerebroventricular chronic administration of oligonucleotides was performed in 2-month-old animals. Briefly, under general anesthesia (as above), mice were fixed with ear bars in a stereotaxic frame and the skin was incised longitudinally after shaving and disinfecting. The guide cannula and its respective dummy cannula were carefully advanced throughout a drilled round window (0.5 mm in diameter) in the midline of the interparietal–occipital bones junction. Cannula was placed in parallel and close to the inner portion of the occipital bone (Fig. 3g). The infusion system was finally attached to the skull by means of acrylic resin once the end of guide cannula was positioned near to the fourth ventricle. To gain in stability and endurance, all the system was cemented with the same resin to two screws fixed in the left and right parietal bones. Post-operative cares were as above.

The body weight for each mouse was weekly recorded, beginning one week after surgery. Individual weights were normalized taking the mean value of initial measures as 100%. At the beginning of the study (66 days), mean body weight was similar for the two cohorts (cRNA: 23.4 ± 0.5 g; siRNA$_{p11}$: 24.2 ± 0.4 g; $p = 0.124$, Student $t$-test). Survival times were analyzed by means of Kaplan–Meier survival analysis with Log-Rank tests for statistical significance. The end stage was defined as the day when mice were unable to right themselves 30 s after being placed on a side. At this time point, animals were euthanized using a carbon dioxide-enriched atmosphere.

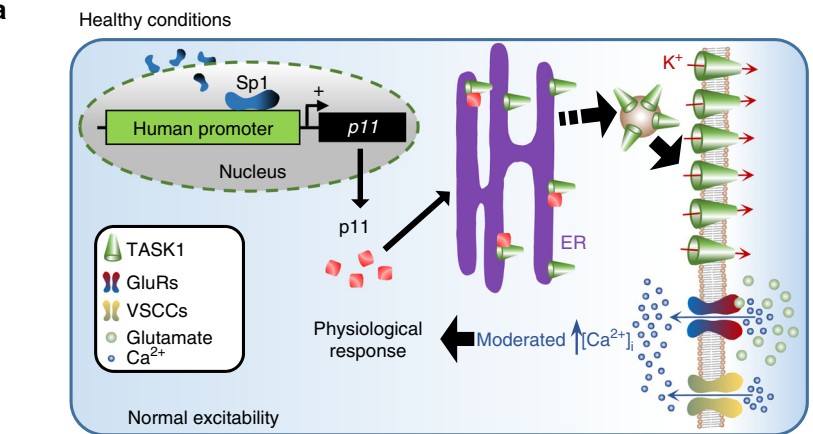

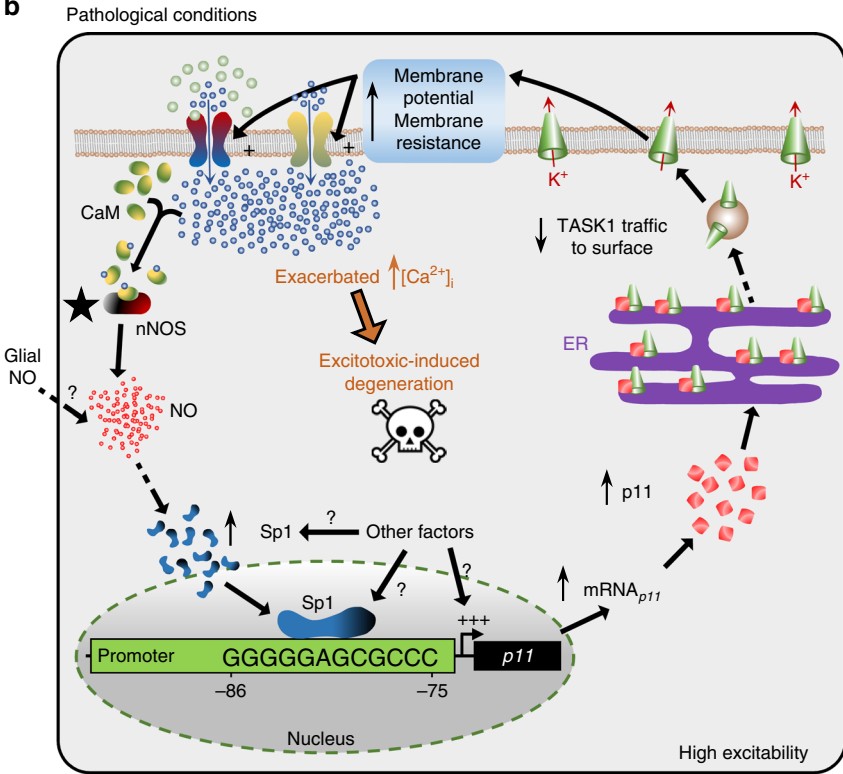

**Fig. 10** Schematic diagram modeling the role of the Sp1–p11–TASK1 triad in neurodegeneration. **a** Physiological expression of the transcription factor Sp1 maintains baseline levels of p11, which regulates IME by controlling TASK1 trafficking from the endoplasmic reticulum (ER) to the plasma membrane. p11–TASK1 complex dissociation is required to overcome ER retention and normal recycling of TASK1 to the surface[14]. Under healthy conditions, when neuronal IME is fine-tuned, moderated $Ca^{2+}$ influx, through GluRs and/or VSCCs, leads to physiological responses, which are controlled by mechanisms of intracellular $Ca^{2+}$ buffering and extrusion. **b** In neurological conditions, over-expression and/or de novo expression of nNOS could be the switch on (star) to initiate a positive feedback loop that contributes to the degeneration and death of sick neurons. Usually, NO production in the nervous system is directed by $Ca^{2+}$ influx driven by NMDA receptors. The calmodulin (CaM)–$Ca^{2+}$ interaction, which is now enabled to activate nNOS, allows for the coupling of pathological NO synthesis to $Ca^{2+}$ entry. NO reacts rapidly with the superoxide anion ($O_2^-$) to form the highly toxic product peroxynitrite ($ONOO^-$) and/ or promotes S-nitrosylation, triggering protein misfolding and aggregation, thus leading to neurotoxicity[8] (omitted for simplicity). In addition, a persistent pathological concentration of NO, involving the sGC/PKG-RhoA/ROCK pathway (omitted for simplicity), induces Sp1 and stimulates the human p11 promoter depending on, at least, a Sp1-binding site (GC-box). Subsequent p11 upregulation sequesters TASK1 to the ER, thus disrupting the normal recycling of this subunit to the surface. This TASK1 uptake/replenishment unbalance enhances IME, which, in turn, increases the opening probability of VSCCs and, GluRs in the presence of glutamate, thereby exacerbating $Ca^{2+}$ entry into the cell. $Ca^{2+}$ overload might contribute to neurodegeneration and, finally, to neuron death. Whether NO of glial origin[6] and/or other unidentified factors promote Sp1 expression/binding and/or p11 expression are not discarded and merit for further investigation

Evaluation of motor function was assessed by rotarod, footprint, runtime, and grip strength tests. Rotarod test was carried out to judge motor coordination, strength, and balance. Animals were daily trained for one week before data acquisition. Mice were placed individually on a cylinder (3.4 cm in diameter), rotating at a constant speed of 15 rpm. The longest latency without falling for each animal was recorded once per week, after three attempts to keep onto the rotarod (Rotarod LE8500, Panlab SA, Barcelona, Spain). Rotarod recordings began one week before surgery and an additional test was performed one day before surgery. The cut-off time was taken at 180 s. As starting point to construct the cumulative probability curve of symptoms onset, we used the first day in which a mouse

showed motor function deficits on the rotarod test (≥10% reduction in the time to fall) followed by progressive deterioration in performance in the next sessions.

Footprint analyses for step length and runtime were performed weekly starting one week after surgery. The hind feet were immersed into innocuous black ink and animals were placed on a conventional calibrated paper along a gangway of 50 cm length. Footprints were analyzed with respect to the step length as illustrated in Supplementary Fig. 6e. Additionally, the time duration required by the mice to run along the track was determined.

Muscle strength of forelimbs and hind limbs was monitored on a grip strength meter (Grip Strength Meter, BIOSEB, Chaville, France) twice a week for each mouse. The animal held by the tail, once ensured that its fore-limbs or hind-limbs were gripping a T-bar or a grid, respectively, were pulled upwards at a 60° angle. The grid and T-bar were connected to a force transducer and grip strength was measured in grams. The highest outcome of three trials per mouse was recorded for each session. Mean value per week for each mouse was taken to construct time course of grip strength for the two cohorts, cRNA-treated or siRNA$_{p11}$-treated SOD1-G93A mice.

Electromyography was performed by inserting a concentric needle electrode into the gastrocnemius muscle of anaesthetized mice (1.5% isoflurane mixed with 100% $O_2$). To ground the system, a monopolar needle electrode was inserted into the tail of the mouse. The electrical signals were amplified and filtered at a bandwidth of 10 Hz–10 kHz for display and digitization purposes. Spontaneous activity of the muscle was monitored and recorded for 30 min, after a period of 5 min of stabilization. Only spontaneous activity with an amplitude of at least 50 μV was considered as a significant signal. Mean and maximum frequency of fibrillation potentials were calculated from automatically selected 1 s-intervals all along the recording session (30 min). Thus, 1,800 data per animal and session were used to construct box-plots (Fig. 7l, m).

**Drugs and treatments**. SMN cultures were treated at different DIV depending on the experimental series. Oligonucleotides (cRNA, siRNA$_{p11}$; 2 μM each) and LVVs (LVV-shRNA$_{lacz}$, LVV-shRNA$_{p11}$, LVV-dnSp1; 2 μl in 350 μl of culture medium) were added to the culture medium at 1 DIV, and were washed out at 4 DIV. For patch-clamp recordings, SMNs (4 DIV) were initially perfused with normal aCSF to obtain baseline control data. Next, SMNs were superfused for 10 min with aCSF containing AEA (3 or 10 μM) before voltage responses were acquired again. Finally, a last round of acquisition was taken after a 10 min washout with AEA-free aCSF (Fig. 2f). For survival experiments, bupivacaine (40 μM, 60 min), isoflurane (0.8 mM, each 10 min for 70 min), AEA (0.03, 0.05, 0.1, 1, 3, 5 or 10 μM, 30 or 60 min), ethanol (0.07%, v/v, 60 min), AM281 (0.5 μM, 65 min), AEA+AM281, alone or in combination with glutamate (150 μM, 30 min) were added to the culture medium at 4–5 DIV (Fig. 2d, e, h, Supplementary Fig. 4c,d). For $Ca^{2+}$ imaging experiments, Fluo-4-charged SMNs were initially perfused (~5 min) with Krebs solution to obtain resting $[Ca^{2+}]_i$. Next, nifedipine (30 μM), ω-conotoxin GVIA (0.5 μM), ω-agatoxin IVA (0.1 μM), NBQX (20 μM), or (DL)-APV (50 μM) were added, for at least 5 min or until a new plateau was reached. Then, glutamate (150 μM) was co-added to the solution for 30 min. The Sp1-interfering agent Mit-A (30 nM, Serva Electrophoresis GmbH, Heidelberg, Germany) was added to the culture medium at 3 DIV for 24 or 36 h. Although, it cannot be fully discarded that Mit-A may impair the binding of another transcription factors with affinity for GC-rich sequences, here, we used this drug as a prospective pharmacological strategy with clinical potential for reducing, at least, Sp1 function.

dNSC34 cultures, after 7–9 days in differentiating medium, were treated with the following drugs: the broad-spectrum NOS inhibitor $N^ω$-nitro-L-arginine methyl ester (L-NAME, 100L μM, 48 h) or the inactive stereoisomer D-NAME (100 μM, 48 h); the long half-life NO donor (Z)−1-[2-(2-aminoethyl)-N-(2-ammonioethyl) amino]-diazen-1-ium-1,2-diolate-NO (DETA/NO) (0.03, 0.1, 0.5, 1 or 2 mM; 6, 12, 24, 48 or 72 h) or DETA (1 mM, 48 h); the specific soluble guanylyl cyclase (sGC) inhibitor 1H-[1,2,4]oxadiazolo[4,3-a]quinoxalin-1-one (ODQ; 20 μM, 48 h; Sigma-Aldrich); the specific protein kinase G inhibitor (PKG-i) guanosine, 3′,5′-cyclic monophosphorothioate, 8-4-chlorophenylthio-, Rp-isomer (Rp-8-pCPT-cGMPS; 10 μM, 48 h; Calbiochem); the specific inhibitor of ROCK (S)-(+)−2-methyl-1-[(4-methyl-5-isoquinolinyl)sulfonyl]-hexahydro-1H-1,4-diazepine dihydrochloride (H1152; 20 μM, 48 h; Tocris Bioscience); and Mit-A (30 nM, 48 h). DETA was obtained by adding DETA/NO (1 mM) to culture medium 48 h before treatment. At that time, almost all NO had been already released from DETA/NO. DETA/NO (1 mM) was also co-added together with ODQ, PKG-i or H1152 in some experiments.

Brainstem slices, at the level of the HN, were incubated with different combinations of drugs according to the aim of each experimental series (see above and main text). For patch-clamp recordings, drug treatments were either acute (10–25 min during patch-clamp recordings) or "chronic" (a minimum of 4 h before the recordings and during the recordings). To study expression levels of diverse molecules, drugs were applied for 6 h before HN microdissection. Control slices were for the same time period in standard aCSF. Acute treatments were as follow: the $Gα_q$ inhibitor YM-254890 (1 μM, 25 min; kindly provided by Astellas Pharma Inc., Tokyo), thyrotropin-releasing hormone (TRH) (10 μM, 10–15 min), the specific inhibitor of ROCK trans−4-[(1 R)-1-aminoethyl]-N-4-pyridinylcyclohexanecar boxamide dihydrochloride, Y27632 (10 μM, 15 min) were applied after chronic incubation with DETA/NO (1 mM, ≥ 4 h; Supplementary Fig. 8c). Drug combinations for chronic treatments were as follow: DETA/NO

(1 mM), DETA/NO+PKG-i (10 μM), DETA/NO+H1152 (20 μM), DETA/NO+ Y27632 (10 μM), DETA/NO+YM-254890, the Rho inhibitor C3 exoenzyme (C3 exo, 2.5 μg ml$^{-1}$), DETA/NO+C3 exo, dexamethasone (Dexa, 1 μM), retinoic acid (20 μM), DETA/NO+retinoic acid, L-NAME (100 μM).

Neonatal rats received a daily subcutaneous injection of L-NAME (90 mg kg$^{-1}$ day$^{-1}$) beginning on the day of nerve crushing. Alternatively, rat pups (P5) received a single injection of oligonucleotides[65]. Anesthetized animals (as above) were placed in a stereotaxic instrument (David Kopf Instruments, Tujunga, CA). siRNA$_{p11}$ or non-targeting siRNA (cRNA) (2 μg per rat; Accell, Dharmacon Inc., Lafayette, CO) in 2 μl of RNase-free PBS was injected at a rate of 0.5 μl min$^{-1}$ by means of a microsyringe (5 μl, Hamilton Company, Tokyo, Japan). Chemically modified siRNA used here has been reported to reduce the expression of target genes in neurons, but not glia, after intracerebroventricular administration[72]. The needle ending was visually guided by means of a surgical microscope to deliver oligonucleotides into the fourth ventricle. The target sequence for the siRNA$_{p11}$ was 5′-CUUCUGAGUUUUAUAUUGU-3′. Brainstem slices of treated rats were quickly extracted at P6–P8 for electrophysiological studies as above.

Adult mice (2-month-old) received a first intracerebroventricular injection of cRNA (5 μg per 5 μl) or siRNA$_{p11}$ (5 μg per 5 μl), in siRNA buffer (Dharmacon), just after implantation of the cannula infusion system. Oligonucleotides administration was weekly repeated under light anesthesia (0.5–1.0% isoflurane mixed with 100% $O_2$) just to avoid unexpected sudden movements of the animal during infusion. An infusion cannula with the appropriated outer diameter was filled with 5 μl of oligonucleotide-containing solution. Intracerebroventricular microinjection of the solution was performed slowly for a period of 10 min driven by an oil-filled tubing system connected to a Hamilton syringe. After the injection, the cannula was left in place for 5 min and then slowly retracted. Sham mice were similarly implanted with the chronic injection system and, weekly manipulated as the experimental group, but volume delivery was omitted.

In another experimental series, adult mice under deep anesthesia (as above) were placed in a stereotaxic device and surgically prepared to receive a single injection of LVVs or vehicle (sterile PBS)[73]. Briefly, a midline dorsal incision exposed the caudal dorsal medulla and, subsequently, glass micropipettes (~25 μm-tip diameter), connected to an oil-filled system, were visually advanced to perform microinjections of the LVVs (Fig. 4c). p11 upregulation in the HN began as soon as 1 day after XIIth nerve transection. In order to minimize feasible harmful effects of p11 upregulation, LVVs were microinjected 3 days before nerve injury since a delay of 3–4 days is expected for effective action of shRNA$_{p11}$[70]. Furthermore, this schedule increases the rate of success of the experiment with the aim to minimize the number of animals required. LVV injection into the HN involves a longer-lasting and more aggressive surgical approach than resection of the XIIth nerve. Therefore, to optimize experimental successfulness, we performed nerve transection in animals in which LVV application did not compromise animal integrity beyond that expected from surgery. Three days before nerve injury, a microinjection (0.5 μl) of LVV-shRNA$_{lacz}$, LVV-shRNA$_{p11}$, or the vehicle was performed in the midline and 0.7 mm below the dorsal surface of the medulla at the obex level. The intramedullar injection rate was 0.1 μl min$^{-1}$ and the micropipette remained in situ for 10 min before being slowly withdrawn at the end of injection. The incision was sutured and cleaned with an aseptic solution (povidone-iodine). 24 days after lentiviral infusion, animals were processed for mRNA and protein extraction or for histological inspection as described above.

In addition, 2-month-old SOD1-G93A mice were anesthetized and placed in a stereotaxic frame to perform intraspinal microinjections of LVV-shRNA$_{lacz}$, LVV-shRNA$_{p11}$, or LVV-dnSp1. The L1–L2 lumbar segment was exposed by laminectomy. Then, two microinjections (0.5 μl each, 2-mm-apart in the rostrocaudal axis) per side of LVV-containing solutions were carried out, as above, at 0.4 mm lateral to the midline and 1 mm below spinal cord dorsal surface (Fig. 9a). To verify the injection place, some wild-type animals received the same volume at the same coordinates of a 2% solution of Chicago sky blue 6B (Sigma) in PBS (Fig. 9a).

For treatment with Mit-A or Riluzole (Sigma-Aldrich, St. Louis, USA), drugs were dissolved in sterilized drinking water supplemented with sucrose (1%). Each animal received daily 5 ml of the drug/vehicle solution to assure that each mouse received the appropriated dosage (Mit-A, 30, 150, 300, or 3000 μg kg$^{-1}$ day$^{-1}$; Riluzole, 25 mg kg$^{-1}$ day$^{-1}$). Vehicle (1% sucrose) and Mit-A or Riluzole containing solutions were freshly prepared each 3 days. Combined treatment with Mit-A and Riluzole did not lead to additive or synergistic effects (not shown).

**Statistics**. Summary data are all presented as mean ± s.e.m. The number of analyzed specimens per experimental paradigm, and statistical tests applied to each data set are indicated in figure legends or in the "Results" section. Statistical analysis was performed using SigmaPlot (Systat Software, Inc.). The minimum significance level was set at $p < 0.05$. Statistical tests were employed for all data sets with similar variance. For comparison between two groups, normally distributed data were analyzed by unpaired Student's t-test, unless otherwise stated, while non-parametric data sets were assessed by Mann–Whitney test. One-way or two-way ANOVA followed by Holm–Sidak post hoc method were employed for comparison of three or more groups which passed normality test. Box-plots show median (white line) and the 25–75% range as box, the whiskers indicate 5–95% range. No data points were excluded from the statistical analysis unless otherwise noted.

Studies using SOD1-G93A mice were all blinded to researcher until data were grouped. The remaining experiments were not blinded unless otherwise noted.

**Reporting summary**. Further information on research design is available in the Nature Research Reporting Summary linked to this article.

## Data availability

The source data underlying Fig. 1d, e, g–i, k, l, 2b, d, e, g-j, 3d, e, i–l, 4d, e, h, j, 5b–e, g, h, j, 6a–i, 7a, b, d–h, l, m, 8, 9f, i and Supplementary Figs. 1, 2, 3c–f, i–k, 4b–d, 5a–c, e–i, 6a–d, f–i, 7, 8b–g, 9a–c, e, g, h, 10–12, 13a–c, e–i, 14 and 15 are provided as a Source Data file. In situ hybridization images were obtained from © 2004 Allen Institute for Brain Science. Allen Mouse Brain Atlas. Available from: http://mouse.brain-map.org/ (Fig. 4a) and from © 2008 Allen Institute for Brain Science. Allen Mouse Spinal Cord Atlas. Available from: http://mousespinal.brain-map.org/ (Figs. 3a, 9a, b). All other data, raw images and/or supplemental materials are available from the corresponding author upon reasonable request.

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

## Acknowledgements

Funding grants: SAF2008-01415 (MICINN/FEDER), SAF2011-23633 (MICINN), BFU2015-71422-R (MINECO/FEDER), PI14/00060 (ISCIII/FEDER) from Spain's Government, as well as P07-CTS-02606, P09-CTS-5445, and P11-CTS-7281 (CICE/FEDER) from Junta de Andalucía, Spain. We thank Dr. Douglas A. Bayliss (University of Virginia, USA) for kindly providing the knock-out mice, Drs. Carmen Castro (University of Cadiz, Spain) and Sergey Kasparov (University of Bristol, UK) for supervision on the initial western blotting experiments and on viral constructions and production, respectively, and Ms. Lucia Molanes, Ms. Eugenia Gomez and Mr. Antonio Torres for their skillful technical assistance. We thank Elaine Lilly, Ph.D. (Writer's First Aid), for English language revision.

## Author contributions

B.M.-L. conceptualized, designed the study, organized data, prepared figures and, wrote and reviewed the paper. L.G.-P., G.R.-B, V.G.-M., G.D.-V., F.P., A.G.-C., N.I and D.G.-F. performed experiments and data analysis. B.M.-L., A.C.-C., and R.M.S. obtained the funding; D.G.-F., supervised initial electrophysiological experiments. R.M.S. and A.G. supervised initial experiments using primary cultures of embryonic spinal motoneurons and calcium-imaging. A.C.-C. supervised experiments using the human p11 promoter. All authors participated in critical review of the manuscript.

## Additional information

**Competing interests:** The authors declare no competing interests.

