## [Peer Review File · Nature Communications]

Reviewers' comments:

Reviewer #1 (Remarks to the Author):

Overall the paper is strong and potentially important with regards to identifying and modulating effects of sp1/p11/task1 to reduce MN death in SOD1 mice.

P11 is a retention factor, which upon expression retains TASK1 in the ER. Quite a few earlier papers have shown interaction of P11 and TASK1 and loss of this interaction (KD of p11) leads to more membrane insertion and function of TASK1 (TASK1 currents). However, this manuscript shows knockdown of P11 results in transcriptional increase of TASK1 as opposed to just membrane insertion. Novel finding is identification of transcription factor SP1 as the orchestrator of p11 expression. In turn pharmacological modulation of SP1 activity leads to a protective, albeit mild, effect due to reduced expression of p11 followed by increase in TASK1. Flip side of the argument is modulating a TF as a therapeutic strategy is risky as it would impact on many other pathways.

Specific issues that need to be addressed for Nat Comm publication

1) Fig 1d -e : Increase in Task1 fluorescence is observed upon P11 knockdown in whole axon as opposed to their claim in neuropil, however, their conclusion of membrane insertion needs to be showed by biotinylation pulldown assay for membrane fractions and observe if there is an increase in TASK1 expression in membrane. Alternatively, authors need to measure or at least show TASK currents are present using pH alterations for key experiments.

2) Does SP1 directly regulate TASK1 expression? This can be evaluated through measuring promoter activity of TASK1 using SP1 and dominant negative SP1

3) Since SP1 is a transcription factor, altering its function could lead to change in many pathways which might contribute to the protective effect. Therefore to conclusively prove TASK1 is the downstream effector modulator, it would be important to perform an experiment to see if the rescue in SOD1 mice by modulating SP1 activity (MitA treatment) is attenuated in the absence of TASK1(SOD1;TASK1-/-mice) (Figure 4f).

Reviewer #2 (Remarks to the Author):

In this paper, the authors explore the possibility of modulating the triad Sp1/p11/TASK1 as a potential neuroprotective approach for the treatment of neurodegenerative diseases such as ALS. They show that p11 prevents insertion of TASK1 in the membrane and therefore increases excitability. Downregulation of p11 therefore reduces excitability and protects against glutamate-mediated Ca²⁺ dysregulation. They further show that nitrosative stress stimulates p11 expression via the transcription factor Sp1, and the interfering with the binding of Sp1 on the p11 promoter is a potential neuroprotective strategy. Therefore, they studied the effect of the anticancer drug Mit-A, which interferes with Sp1 binding on motoneuron excitability. Mit-A decreased excitability and Ca²⁺ dysregulation, and improved lifespan and behavioral scores of ALS mice.

This paper represents an impressive amount of work. The number and diversity of the experiments presented here are remarkable. However, in a way, the sheer amount of information does a bit of disservice to this paper. The constant reference to whole sections of texts in the supplementary information and 20 extra figures makes it very hard to parse through and fully understand the various experiments that are presented here. Note that this may not necessarily be the authors fault, and that it may be an unfortunate side-effect of the format of this journal. All in all, most of

the results of the paper are compelling and the paper should have broad appeal.

My main concerns with this paper is that the authors seem absolutely convinced that motoneuron hyperexcitability is a fundamental feature of ALS. Although motoneuron hyperexcitability has been reported (e.g. Van Zundert et al, J Neurosci 2008; Martin et al, Neurobiol Dis 2013), it seems today that the situation is more nuanced than the authors' description. Various authors argue that hyperexcitability might be a transitory state in ALS (Devlin et al, Nat Comm 2015; Kim et al, J Neurosci 2017). Other reports even suggest that hyperexcitability might even be protective in ALS (Leroy et al, eLife 2014; Saxena et al, Neuron 2013). In the present paper, the authors have studied the excitability of cultured cells only, despite the fact that they could have easily performed recording of hypoglossal motoneurons in slices, since they've used that very technique in the paper. Van Zundert et al (J Neurosci 2008), despite showing hyperexcitability in hypoglossal motoneurons, report no change in resting membrane potential and input conductance, contrary to the present results. What evidence do the authors have that the effects that they observe in vivo is mediated at the level of the motoneurons themselves? I believe this whole controversy on the role of hyperexcitability in ALS need to be, at the very least, discussed in the discussion section.

Minor comments:

- In Suppl Note 7, LVV-shRNA(p11) was injected 3 days before nerve resection. Why this particular timing? What happens if the injection is performed on the day, or shortly after the resection?
- In Suppl Fig 6, the data are represented by bar graphs, instead of the scatter with mean / boxplots that were used previously to depict the same kind of data. Why the change? Are the bar plots hiding something about the distribution of the data points?
- Since I assume there are no size restrictions on the suppl figures, most of them could be made much larger to be easier to read.

Reviewer #3 (Remarks to the Author):

In this manuscript, data are presented to advance the hypothesis that elevated levels of p11 in models of neurodegeneration (SOD, nerve injury) limit expression of TASK1, thereby leading to enhanced intrinsic excitability and motor neuron cell death. Furthermore, elevated levels of NO following nerve injury are linked to changes in motoneuronal TASK channel function via a mechanism according to which injury-induced nitrosative stress enhances Sp1 expression to upregulate p11 via a GC-box in the NO-responsive -96 to -70 fragment of the p11 promoter. In SOD or nerve-injured mice, treatment with MitA to inhibit Sp1-GC box interactions, decreased motoneuronal excitability and degeneration, and delayed or reduced symptoms; the beneficial effects in these models rivaled those of riluzole, a standard treatment for ALS.

This work provides some exciting new evidence that p11-mediated downregulation of the functional expression of a specific background channel may contribute to motor neuron degeneration in ALS and nerve injury models. It also suggests that re-purposing of MitA for treatment of ALS might be worth considering.

Overall, the manuscript covers a lot of territory in its attempt to provide a comprehensive accounting of this mechanism, spanning from electrical recordings from in vitro preparations, through manipulation of gene expression and signaling pathways in vitro and in vivo, and analysis of gene regulatory mechanisms, to testing potential therapeutic treatments based on this mechanism in preclinical models of motor neuron degeneration. That said, the presentation was extremely disjointed and difficult to follow, with data and text excluded from the main text and variously placed in supplemental notes and figures. For example, although the authors emphasized the applicability of their mechanism to two different models of motor neuron degeneration, the entire set of experiments with the nerve transection model was described in a single sentence of the main text and the bulk of the actual work was relegated to supplemental sections.

Other major issues with the experiments are provided below:

1. There are some apparent inconsistencies with respect to basal TASK channel contributions to excitability, calcium responses and cell survival across different experiments. For example, pH changes cause membrane hyperpolarization and decreased input resistance in hypoglossal motor neurons in slices, an effect which is enhanced slightly by p11 knockdown in vivo (Fig. S1d). However, there was no effect of pH changes on Em or Rn of SMNs in culture, and in these cells p11 knockdown had a major effect (Fig. S1g). What accounts for the difference in basal current in these two preparations? Even if p11 is hampering functional expression of TASK1, is there no role for TASK3 in SMNs?

In any case, these data suggest little basal TASK channel activity in the SMN preparation. Nevertheless, in subsequent experiments, various TASK channel blockers reduced SMN cell survival and exacerbated Glu-evoked excitotoxicity (Fig. S3). Moreover, AEA caused membrane depolarization, increased Rn and reduced cell survival in a TASK1-dependent manner (Fig. S3). Thus, unlike the experiments with pH changes, these experiments suggest there is basal TASK channel activity in SMN cells. This is confusing, and seems internally inconsistent with previous experiments that implied p11 was required for TASK1 functional expression.

2. The VGCC mechanisms described for Glu-induced calcium dynamics are also confusing and not well described. It appears that Ca deregulation is largely due to P/Q-type channels and independent of L- or N-type Ca channels under baseline conditions, but becomes strongly dependent on those latter Ca channel types after p11 knockdown (Fig S2f). What accounts for this shift in the relative contributions of these different Ca channel types once p11 expression is reduced?

3. The reported effects in the SOD model based on this siRNA knockdown approach, with 4th ventricle injections of siRNA constructs reducing p11 expression in the lumbar spinal cord, seems incredible. Is there another way to delete p11 in motor neurons (e.g., with conditional knockout models)? This would go a long way to increase confidence in the results. In addition, the western blot illustrating knockdown of p11 in SOD mice looks like there could have been an issue with the transfer (Fig S5c). Is this the best example?

4. What are the spinal and brainstem localization patterns of p11. The methods used for manipulating p11 expression allow no cell-type specificity so how certain can we be that any effects are cell autonomous due to p11 depletion in motoneurons? Can a loss of p11 and upregulation of TASK1 be demonstrated in motoneurons following siRNA (e.g., by histochemical approaches)?

5. What are the common features of the proposed mechanisms underlying neurodegeneration associated with ALS (SOD model) and nerve injury (hypoglossal nucleus model) that would be targeted by p11? is excitotoxicity a primary consideration in nerve injury that leads to use of Glu administration as a relevant test (Fig. S6f)? or, is that more likely a reflection of the loss of trophic support after cutting the nerve?

6. An excessive amount of information re. NO effects on cell excitability etc. is encompassed by the 2 sentences (6 lines of text) that must be accessed via Supplementary Note 8 and Figs. S7-S9). Some more effort should be expended in synthesizing and clarifying these data (e.g., distinct mechanisms of acute vs. chronic effects of NO, implications of dexamethasone results, and why all are necessary and relevant to the current work?) In addition, is the in vitro Glu excitotoxicity model dependent on nitrosative stress (i.e., is it similar to the in vivo condition, as described)?

7. The effect of Sp1dn or MitA on p11 expression seems to work constitutively, even without the enhancing effects of DETA/NO (e.g., Fig. S11, Fig. 4). This seems confusing, since the Sp1-GC box

interaction was identified as an NO-stimulated effect. Please clarify.

8. If MitA works in SMNs from SOD mice by increasing TASK expression, why is there no effect on Rn (Fig. 4g)? Were effects of MitA sensitive to Sp1dn? Also, the MitA effect on survival of G93A SMNs (~40%) was far less robust than on wild type SMNs (~100%) – compare Fig. 4e to Fig. 4j. Is that expected? why?

9. The results with MitA by comparison to riluzole are important. Is it possible that combined treatment with both would have even greater effects (perhaps additive or synergistic)?

Minor issues:

1. For the nerve injury model, changes in intrinsic excitability due to p11 knockdown were attributed to TASK channels by examining effects of changes in pH. This same test was not performed on SMNs from SOD mice, where TASK contributions to changes in Em and Rn were not evaluated experimentally.

2. What is a % change in membrane potential (e.g., see Fig. S8g, Fig. S9e)?

3. Fig. S10 and Supplementary Note 9: DETA/NO increased luciferase activity expressed from the p11 promoter, but how can this be definitively attributed to enhanced transcription (as opposed to DETA/NO effects on translation or luciferase stability)?

4. For the supershift analysis in Fig. 3g, there appears to be quantitatively a much greater loss in the unbound fraction than is accounted for in the shifted fraction?

5. What was the rationale for switching the cell model for in vitro work from SMNs to dNSC34s?

6. Were the antibodies validated against knockout lines (e.g., TASK1)?

Dear Reviewers,

We would like to thank you for your helpful comments on the manuscript. We are sincerely grateful to you for the time spent reading, commenting, and suggesting new experiments, which have greatly improved the quality of the work. We have tried our best to carefully consider and respond to the questions raised. Detailed answers to all the reviewers' comments are provided below. To facilitate the review process, changes in the text have been tracked with another color font, as well, a detailed list of new contributed figures and rearrangement of previous figures is added at the end of this document.

RESPONSE TO REVIEWERS

REVIEWER #1:

Overall the paper is **strong and potentially important** with regards to identifying and modulating effects of sp1/p11/task1 to reduce MN death in SOD1 mice. P11 is a retention factor, which upon expression retains TASK1 in the ER. Quite a few earlier papers have shown interaction of P11 and TASK1 and loss of this interaction (KD of p11) leads to more membrane insertion and function of TASK1 (TASK1 currents).

Reviewer: However, this manuscript shows knockdown of P11 results in transcriptional increase of TASK1 as opposed to just membrane insertion.

Response 1: We think that we have not been clear enough for this statement. We have tried to clarify this concern in the new version of our manuscript (lines 109-111). Note that, p11 up- or down-regulation induced by DETA/NO or siRNA_{p11}, respectively, did not result in a parallel change of TASK1 expression (**before:** Figs S1e, S7h & S8b-e; **now:** Figs 1i, 5c-e, S8g & S9a). Furthermore, in non-contributed data from an additional set of experiments, both, siRNA_{p11} and dexamethasone altered p11 levels in the motoneuron-like cell line NSC34, without changing TASK1 expression (see **figures A & B**, below). Therefore, we did not find any support for a regulatory action of p11 on TASK1 transcription.

Figure A: Effect of siRNA_{p11} on the expression of p11 and TASK1 in NSC34 cells. Average data were obtained from 3 independent experiments. Protein levels were relativized to α -tub expression. * $p < 0.05$, Mann-Whitney U test.

Figure B: Effect of dexamethasone treatment (100 nM, 72 h) on the expression of p11 and TASK1 in NSC34 cells. Average data (B) were obtained from 3 independent experiments. Protein levels were relativized to α -tub expression. * $p < 0.05$, Mann-Whitney U test.

Reviewer: Novel finding is identification of transcription factor SP1 as the orchestrator of p11 expression. In turn pharmacological modulation of SP1 activity leads to a protective, albeit mild, effect due to reduced expression of p11 followed by increase in TASK1.

Response 2: Results presented in Fig 3f,g (**now:** Fig 6f,g) strongly support a binding of Sp1 (and probably also Sp3) to the minimal NO-responsive element. Furthermore, in the current Fig S11b,c we show that interfering with Sp1 induces p11 downregulation. Altogether, these outcomes indicate that Sp1 controls p11 expression, although we cannot rule out the possibility that other transcription factors might regulate p11 expression.

On the other hand, the mechanism of action of Mit-A involves its binding to G-C rich sequences, thus interfering with transcription factor attachment. In this way, here we selected Mit-A, a pharmacological tool which interferes, at least, with Sp1 function, mainly by its clinical potential application, given that, FDA approved its use as an anticancer drug (lines 363-368). As we also stated in the manuscript, Mit-A-induced upregulation of TASK1 was an unexpected outcome (line 371). This finding might be due to the interference with the repressive control exerted by Sp1/Sp3 or other

unidentified factors with affinity to a GC-box, rather than to the regulation of TASK1 transcription by p11.

Reviewer: Flip side of the argument is modulating a TF as a therapeutic strategy is risky as it would impact on many other pathways.

Response 3: We agree with the reviewer at this point. However, we would like to highlight that the main reason for selecting Mit-A, which modulates Sp1, was that it is a FDA-approved drug with anticancer properties which is currently being tested in clinical trials for several cancers (<http://www.ClinicalTrials.gov>; Identifiers: NCT02859415, NCT01624090). Furthermore, the oral lower dose we administered (30 µg/kg/d) is in the lower range for its intravenous application in chemotherapy (25 µg/kg/d over 5-10 days)¹. In addition, we would expect that concentration of Mit-A that might be reached within the organism after oral administration was lower than that obtained with equivalent intravenous doses, thus reducing possible side-effects. It is noteworthy that Sp1 was found upregulated in motoneurons in the two neurodegenerative models we used (**before:** Fig S19c,d; **now:** Figs 7b & 8b,c), in which Mit-A treatment would be not expected to fully block Sp1 action, as usually assumed for any pharmacological treatment. Thus, partial inhibition of Sp1 function in conditions in which Sp1 is upregulated, could more selectively affect pathological than physiological Sp1-mediated processes. In agreement, we did not observe Mit-A associated side-effects (anorexia or bleeding tendency) in chronically-treated mice. This is now discussed in the new version of the manuscript (lines 568-573).

Specific issues that need to be addressed for Nat Comm publication

Reviewer: Fig 1d-e : Increase in Task1 fluorescence is observed upon P11 knockdown in whole axon as opposed to their claim in neuropil,

Response 4: We apologize for an unfortunate choice of terminology. In primary cultures of embryonic spinal cord motoneurons (SMNs), we have used the term “neuropil” to refer to areas containing neuronal processes (axons and dendrites). In our experimental conditions, SMNs are still in a developing stage, and therefore, it is difficult to discriminate between axons and dendrites without using appropriated markers. To avoid

confusion, we have now used the term “neurites” (line 109). In the photomicrographs shown in Fig 1d (**now**: Fig 1j), the presence of some spine-like protrusions suggests that they could probably be dendrites. However, to avoid a biased misinterpretation we refer them as “neurites”.

Reviewer: However, their conclusion of membrane insertion needs to be showed by biotinylation pulldown assay for membrane fractions and observe if there is an increase in TASK1 expression in membrane. Alternatively, authors need to measure or at least show TASK currents are present using pH alterations for key experiments.

Response 5: We appreciate reviewer’s concern to aware us on the importance of these results. We think reviewer claims for the results previously presented in Fig S1d,g (**now**: Fig 1g,m). Because of their relevance, they have been now contributed as main results in the new version of the manuscript. In this figure, we report pH-induced changes in membrane potential and input resistance of hypoglossal motoneurons and SMNs. Note that, in control and cRNA-treated SMNs at 3-4 DIV, pH-dependent changes are negligible. However, after siRNA_{p11} treatment, pH-induced changes are evident in wt and task3^{-/-} SMNs but, conclusively, not in task1^{-/-} SMNs. Therefore, these data support our assumption that p11 regulates *functional expression* of the channel at the plasma membrane (lines 118-120).

Reviewer: Does SP1 directly regulate TASK1 expression? This can be evaluated through measuring promoter activity of TASK1 using SP1 and dominant negative SP1

Response 6: Reviewer arises an interesting and relevant question. As it can be extracted from the Eukaryotic Promoter Database (EDB) for TASK1 (Kcnk3_1) promoter (https://epd.vital-it.ch/cgi-bin/get_doc?db=mmEpdNew&format=genome&entry=Kcnk3_1), there are several potential Sp1/Sp3 binding sites and GC-rich motifs in the -1000 to 100 bp promoter region (**Figure C**). Hence, it is highly likely that Sp1 regulates to some extent TASK1 expression.

[REDACTED]

However, in our hands, we have not obtained confident results after transfection of NSC34s with a plasmid expressing the dominant negative for Sp1 (dnSp1). Inspection of Fig S11b and **Figure D**, which depicts the results for this experiment, reveals that whereas p11 was clearly downregulated by dnSp1, a change in the expression of TASK1 was not consistent. Thus, although it cannot be discarded that Sp1 controls TASK1 promoter activity, it has not become evident in our experimental conditions.

Figure D. As Fig S11b, but immunoblot for TASK1 is also shown. Note that while p11 is clearly downregulated in the presence of dnSp1, results from TASK1 are not consistent. Proteins extracted from 3 independent experiments were processed in the western.

Within the scope of our work, chronic (4-6 h) treatment with DETA/NO led to p11 upregulation but did not affect TASK1 levels (**before:** Figs S7h & S8b-e; **now:** Figs 5c-e, S8g & S9a). Furthermore, p11 upregulation after nerve injury was dependent on NO synthesis (**before:** Fig S9d; **now:** Fig 5g). These outcomes provided the background to hypothesize on the feasible role of NO as a regulator of human p11 promoter activity (lines 274-310). Performance of these experiments prompted us to identify Sp1 as a feasible transcription factor mediating p11 induction under nitrosative stress, a common event occurring in most neuropathological conditions (lines 312-351). Finally, the finding that, dnSp1 did not alter TASK1 expression (**Figure D**), supports that Sp1 is not a determining factor in the control of constitutive activity of the TASK1 promoter. As this letter will be accessible online together with the manuscript, we have not included these results in the paper to avoid increasing the huge volume of results.

Reviewer: Since SP1 is a transcription factor, altering its function could lead to change in many pathways which might contribute to the protective effect. Therefore to conclusively prove TASK1 is the downstream effector modulator, it would be important to perform an experiment to see if the rescue in SOD1 mice by modulating SP1 activity (MitA treatment) is attenuated in the absence of TASK1 (SOD1;TASK1^{-/-}mice) (Figure 4f).

Response 7: We agree with the reviewer. Based on the mechanism of action of Mit-A, we cannot fully reject the possibility that other pathways could also contribute to its neuroprotective action. The mechanism of action of Mit-A involves its binding to GC-rich sequences, thus interfering with transcription factor binding. Therefore, here we used Mit-A, as a FDA-approved pharmacological tool to interfere with Sp1 function (lines 363-368). The experiment suggested by the reviewer is interesting and would be conclusive. For that reason, we have tried to perform it over the last six years. However, task1^{-/-} x task1^{-/-} crossbreeds often fail, and even when occasionally they are successful, only 3-5 pups are obtained at best. Because SOD1^{G93A} females are infertile, we started to crossbreed task1^{-/-} females with SOD1^{G93A} males. In this case, rate of success and number of descendants were even much lower and, when occurred, only 25% of descendants were task1^{+/-}/SOD1^{G93A} males. These were subsequently crossbred once again with task1^{-/-} females to finally obtain 25% of descendants task1^{-/-}/SOD1^{G93A} males. On that basis, the experiment proposed by the referee was not finally performed because of the unfeasibility to obtain, in a reasonable time-window, a homogeneous population of 20-30 animals to carry out parallel treatments with vehicle and Mit-A in two task1^{-/-}/SOD1^{G93A} groups randomly separated. Furthermore, it cannot be predicted how different genetic backgrounds of task1^{-/-} and SOD1^{G93A} would affect disease progression or Mit-A action on task1^{-/-}/SOD1^{G93A} mice.

As we contributed throughout the manuscript, Sp1/p11/TASK1 dysregulation in motoneurons is a common hallmark for the two neurodegenerative models we have used. Alternatively, in Figure S20 (**now:** Fig 8f), we show that Mit-A was neuroprotective for injured motoneurons in wt and task3^{-/-} mice, but, conclusively, not in task1^{-/-} mice. These outcomes strongly support that TASK1 is a relevant target for the neuroprotection reached under Mit-A treatment in the experimental model of motoneuron degeneration induced by axotomy. Furthermore, in the new contributed data, we show that Mit-A treatment

reduces p11 expression and increases TASK1 levels in lumbar spinal motoneurons (Fig S14c,d).

REVIEWER #2:

In this paper, the authors explore the possibility of modulating the triad Sp1/p11/TASK1 as a potential neuroprotective approach for the treatment of neurodegenerative diseases such as ALS. They show that p11 prevents insertion of TASK1 in the membrane and therefore increases excitability. Downregulation of p11 therefore reduces excitability and protects against glutamate-mediated Ca^{2+} dysregulation. They further show that nitrosative stress stimulates p11 expression via the transcription factor Sp1, and the interfering with the binding of Sp1 on the p11 promoter is a potential neuroprotective strategy. Therefore, they studied the effect of the anticancer drug Mit-A, which interferes with Sp1 binding on motoneuron excitability. Mit-A decreased excitability and Ca^{2+} dysregulation, and improved lifespan and behavioral scores of ALS mice.

Reviewer: This paper represents an impressive amount of work. The number and diversity of the experiments presented here are remarkable. However, in a way, the sheer amount of information does a bit of disservice to this paper. The constant reference to whole sections of texts in the supplementary information and 20 extra figures makes it very hard to parse through and fully understand the various experiments that are presented here. Note that this may not necessarily be the authors fault, and that it may be an unfortunate side-effect of the format of this journal. All in all, most of the results of the paper are compelling and the paper should have broad appeal.

Response 1: We thanks the reviewer for his/her positive comments on our paper. We agree that excessive information has been contributed in the paper. In the new version of the manuscript, we have tried our best to reorganize figures and text to gain in fluidity, readability, and understandability, according to the journal format.

Reviewer: My main concerns with this paper is that the authors seem absolutely convinced that motoneuron hyperexcitability is a fundamental feature of ALS. Although motoneuron hyperexcitability has been reported (e.g. Van Zundert et al, J Neurosci 2008; Martin et al, Neurobiol Dis 2013), it seems today that the situation is more nuanced than the authors' description. Various authors argue that hyperexcitability might be a transitory state in ALS (Devlin et al, Nat Comm 2015; Kim et al, J Neurosci 2017). Other reports even suggest than hyperexcitability might even be protective in ALS (Leroy et al, eLife

2014; Saxena et al, Neuron 2013). In the present paper, the authors have studied the excitability of cultured cells only, despite the fact that they could have easily performed recording of hypoglossal motoneurons in slices, since they've used that very technique in the paper. Van Zundert et al (J Neurosci 2008), despite showing hyperexcitability in hypoglossal motoneurons, report no change in resting membrane potential and input conductance, contrary to the present results. What evidence do the authors have that the effects that they observe in vivo is mediated at the level of the motoneurons themselves? I believe this whole controversy on the role of hyperexcitability in ALS need to be, at the very least, discussed in the discussion section.

Response 2: Reviewer's comment is appropriate, relevant, and pertinent. We have contributed further evidence that Sp1/p11 dysregulation occurs in motoneurons in the two degenerative models used in our work (Figs 3e, 4c,f,g,i, 7b, 8b,c & 9). In our previous version of the manuscript, we avoided the discussion on hyperexcitability controversy for space limitations. Following suggestions from referee, we have now taken it in full consideration, by discussing this concern in the new version of discussion (lines 497-522).

Minor comments:

Reviewer: In Suppl Note 7, LVV-shRNA(p11) was injected 3 days before nerve resection. Why this particular timing?

Response 3: We have selected this timing by two essential reasons. The first reason was that a delay of 3-4 days could be expected between LVV injection and actual/optimal action of shRNA_{p11} in transduced motoneurons². However, as shown in Fig S6c (**now:** Fig 4e), p11 upregulation in the hypoglossal nucleus began as soon as 1 day after XIIth nerve transection. Therefore, administration of LVVs 3 days before nerve injury sought the early interference with p11, thus minimizing the feasible harmful effects of p11 induction. The second reason was to increase the rate of success of the experiment with the aim to minimize the number of animals required. LVV injection into the hypoglossal nucleus requires for a much longer-lasting and more aggressive surgical approach than resection of the XIIth nerve. Therefore, to optimize recourses and compliance with ethical guidelines, we performed nerve transection in animals in which LVV application did not

compromise animal integrity beyond that expected from surgery. Finally, a lesser rate of success would have very likely resulted if LVV injection and nerve transection had been carried out in a same surgical session, which would have been even more stressful and long-lasting. A statement on this issue has been included in the methods section of the new version of the manuscript (lines 1108-1116).

Reviewer: What happens if the injection is performed on the day, or shortly after the resection?

Response 4: Reviewer is likely thinking in the clinical translation of the results; however, we have not performed these experiments by the aforementioned reasons. Alternatively, Mit-A resulted neuroprotective when applied from the same day of nerve transection (**before:** Fig S20; **now:** Fig 8ef). In this way, it is expected that the delay between Mit-A administration and subsequent alterations in p11/TASK1 expression was shorter than after LVV microinjection. Furthermore, new contributed data (Fig 8e) show that chronic administration of L-NAME, a NOS inhibitor, or Fasudil, a ROCK inhibitor, beginning 5 days after nerve injury, were both beneficial for axotomized motoneurons.

Reviewer: In Suppl Fig 6, the data are represented by bar graphs, instead of the scatter with mean / boxplots that were used previously to depict the same kind of data. Why the change? Are the bar plots hiding something about the distribution of the data points?

Response 5: In the new version of figures, all data have been represented as scatter/boxplots.

Reviewer: Since I assume there are no size restrictions on the suppl figures, most of them could be made much larger to be easier to read.

Response 6: In this new version of the paper, we have tried our best to accommodate figures as reviewer suggests.

REVIEWER #3:

In this manuscript, data are presented to advance the hypothesis that elevated levels of p11 in models of neurodegeneration (SOD, nerve injury) limit expression of TASK1, thereby leading to enhanced intrinsic excitability and motor neuron cell death. Furthermore, elevated levels of NO following nerve injury are linked to changes in motoneuronal TASK channel function via a mechanism according to which injury-induced nitrosative stress enhances Sp1 expression to upregulate p11 via a GC-box in the NO-responsive -96 to -70 fragment of the p11 promoter. In SOD or nerve-injured mice, treatment with MitA to inhibit Sp1-GC box interactions, decreased motoneuronal excitability and degeneration, and delayed or reduced symptoms; the beneficial effects in these models rivaled those of riluzole, a standard treatment for ALS.

This work provides some exciting new evidence that p11-mediated downregulation of the functional expression of a specific background channel may contribute to motor neuron degeneration in ALS and nerve injury models. It also suggests that re-purposing of MitA for treatment of ALS might be worth considering.

Reviewer: Overall, the manuscript covers a lot of territory in its attempt to provide a comprehensive accounting of this mechanism, spanning from electrical recordings from in vitro preparations, through manipulation of gene expression and signaling pathways in vitro and in vivo, and analysis of gene regulatory mechanisms, to testing potential therapeutic treatments based on this mechanism in preclinical models of motor neuron degeneration. That said, the presentation was extremely disjointed and difficult to follow, with data and text excluded from the main text and variously placed in supplemental notes and figures.

Response 1: We thank reviewer for the positive comments on our paper. We agree that too much of information has been contributed in the paper. According to the journal format, we have now included 10 main figures and extended the main text to accommodate data and text previously contributed as supplementary information. In the new version of the manuscript, we have tried our best to reorganize figures and text to gain in fluidity, readability, and understandability.

Reviewer: For example, although the authors emphasized the applicability of their mechanism to two different models of motor neuron degeneration, the entire set of experiments with the nerve transection model was described in a single sentence of the main text and the bulk of the actual work was relegated to supplemental sections.

Response 2: In our attempt to reduce extension of the main information, we initially relegated results focused on the nerve transection model to the supplementary information. Following the reviewer's suggestion, we have now included most of them as main information.

Other major issues with the experiments are provided below:

Reviewer: 1. There are some apparent inconsistencies with respect to basal TASK channel contributions to excitability, calcium responses and cell survival across different experiments. For example, pH changes cause membrane hyperpolarization and decreased input resistance in hypoglossal motor neurons in slices, an effect which is enhanced slightly by p11 knockdown in vivo (Fig. S1d). However, there was no effect of pH changes on E_m or R_N of SMNs in culture, and in these cells p11 knockdown had a major effect (Fig. S1g). What accounts for the difference in basal current in these two preparations? Even if p11 is hampering functional expression of TASK1, is there no role for TASK3 in SMNs?

Response 3: We understand the reviewer's concerns and we agree that additional information should be contributed. Data about the effect of pH on TASK-mediated currents have been now contributed as main information (Fig 1f,g,m; lines 83-97). The apparent inconsistencies that reviewer reasonably found could be in part explained by differences between preparations and in the stage of functional differentiation of motoneurons in each experimental model. Recordings of hypoglossal motoneurons were performed in slices obtained from P6-P9 rats. At this postnatal stage of differentiation, motoneurons consistently display functional expression of TASK channels³. However, in the new contributed data (Fig 1h), and as we now detail in the new version of the manuscript (lines 100-108), "*in our experimental conditions, IME of wild-type SMNs (SMNs^{wt}) changed over time in culture (Fig. 1h). Thus, whereas V_m and R_N remained steady throughout the first 3-4 days-in-vitro (DIV), IME of SMNs^{wt} progressively declined*

over the 4-6 DIV interval, and subsequently remained stable up to 8 DIV, the last time point tested (Fig. 1h). These outcomes are consistent with a progressive increase in surface expression of functional background K^+ channels at the 4-6 DIV interval. Therefore, next experiments were carried out in SMN^{wt} at 3-4 DIV (Fig. 1i), a time window over which plasma membrane expression of TASK is still low". This underlines the very different values of V_m and R_N obtained from hypoglossal motoneurons (Fig 1e) and SMNs (Fig 1l) under cRNA-treatment. Note that R_N of SMNs was almost an order of magnitude higher than those of hypoglossal motoneurons. This indicates that surface expression of background channels is much lower in SMNs, at this stage of functional differentiation, than in hypoglossal motoneurons. Anyway, we assume that perhaps the logical reviewer concern was due to our attempt to synthesize experimental protocols. As different tests were performed at the 3-4 or 4-5 DIV intervals, we initially summarized them as 4 DIV. We are very grateful to reviewer for alerting us on the inappropriate use of this abbreviation for results interpretation. We have carefully revised the paper on this issue.

Reviewer: In any case, these data suggest little basal TASK channel activity in the SMN preparation. Nevertheless, in subsequent experiments, various TASK channel blockers reduced SMN cell survival and exacerbated Glu-evoked excitotoxicity (Fig. S3). Moreover, AEA caused membrane depolarization, increased R_N and reduced cell survival in a TASK1-dependent manner (Fig. S3). Thus, unlike the experiments with pH changes, these experiments suggest there is basal TASK channel activity in SMN cells. This is confusing, and seems internally inconsistent with previous experiments that implied p11 was required for TASK1 functional expression.

Response 4: Please consider the preceding explanation (**response 3**). Experiments showed in Fig. S3 (**now:** Fig. 2c-h) were performed at 4-5 DIV, as now indicated in the experimental protocols.

Reviewer: 2. The VGCC mechanisms described for Glu-induced calcium dynamics are also confusing and not well described. It appears that Ca deregulation is largely due to P/Q-type channels and independent of L- or N-type Ca channels under baseline conditions, but becomes strongly dependent on those latter Ca channel types after p11

knockdown (Fig S2f). What accounts for this shift in the relative contributions of these different Ca channel types once p11 expression is reduced?

Response 5: We hope that this concern was now successfully clarified in the new version of the supplementary note (lines 29-37). “*Within the analyzed time window, contribution of L- and N-type channels to glutamate-induced Ca²⁺ deregulation was lesser than that of P/Q-type. The strengthening of L-/N-type dependence under siRNA_{p11} treatment could be the result of two mechanisms. First, p11 down-regulation induces V_m hyperpolarization, thereby reducing opening probability of non-fully inhibited L- and/or N-type channels under the drug concentrations used in our experiments. Second, if full inhibition of L- and/or N-type channels had been reached in our experiments, siRNA_{p11}-induced hyperpolarization would be expected to reduce Ca²⁺ entry via NMDARs and/or non-targeted VSCCs.*”

Reviewer: 3. The reported effects in the SOD model based on this siRNA knockdown approach, with 4th ventricle injections of siRNA constructs reducing p11 expression in the lumbar spinal cord, seems incredible.

Response 6: We have performed new experiments and contributed additional data (Fig. 3h; Suppl. Fig. 6c) to confer more reliability to the siRNA knockdown approach (lines 219-226). “*We firstly tested the efficacy of a weekly injection of crRNA or siRNA_{p11} into the 4th ventricle to affect lumbar motoneurons, beginning at the pre-symptomatic stage (2-month-old), once the skull was developed enough to implant the injection system (Fig. 3g). By using this procedure, we observed that siRNA_{p11} consistently reduced mRNA_{p11} and p11 protein in the lumbar spinal cord after 8 weeks of treatment (Supplementary Fig. 6a,b). Quantitative analysis of p11-immunolabelling in lumbar motoneurons revealed that the decline in labelling intensity was already evident 2 weeks after beginning of siRNA_{p11}-treatment (Supplementary Fig. 6c).*” Then, we demonstrated that the “*chronic administration of siRNA_{p11} in SOD1^{G93A} mice, which efficiently reduced p11-immunoreactivity in the lumbar spinal cord (Fig. 3h), ...*”. We now believe that the effectiveness of this experimental approach cannot be objectively questioned if we rely on the contributed experimental evidence. Furthermore, the chemically modified siRNA used in our work, “Accell siRNA”, has been reported to efficiently reduce the expression of target genes in neurons, but not glia, after ICV administration⁴. On the other hand, ICV

injection is a delivery technique that is minimally invasive, and Accell siRNA delivery has been proposed to have the potential for neurotherapeutic exploitation when it is necessary to reach vast areas of the brain⁵⁻⁷. Moreover, since gene knockdown was selectively achieved in differentiated mature neurons⁴, this strategy is suggested to be exploited in neuron-specific diseases, such as neurodegenerative disorders³.

Reviewer: Is there another way to delete p11 in motor neurons (e.g., with conditional knockout models)? This would go a long way to increase confidence in the results.

Response 7: Experiments suggested by reviewer would be conclusive and interesting enough to be addressed as a new project. Based on our results, it would be relevant to develop inducible conditional knockouts for Sp1 and/or p11, as well as an inducible conditional knock-in for TASK1 overexpression in motoneurons, to subsequently study their potential neuroprotective effects on the SOD1^{G93A} transgenic model of ALS. However, some considerations must be taken into account regarding the current models used as motoneuron-specific conditional knockouts (VAcHT:Cre or Hb9:Cre). The VAcHT:Cre delivers Cre to a **subset** (~ 50% at best) of adult motoneurons that selectively include slow (S) and fatigue-resistant (FR) motoneurons^{8,9}. Strikingly, larger soma-sized motoneurons, i.e. fast-fatigable (FF), are more vulnerable to neurodegeneration in ALS¹⁰⁻¹², while slow motor units tend to be spared in the disease^{11,13,14}. Therefore, even in the case that we were able to develop a viable conditional knockout for p11 (VAcHT:Cre/p11), it would be highly likely that neuroprotective effects would be difficult to detect in VAcHT-Cre/p11xSOD1^{G93A} mice, because *the targeted motoneurons would be the ones mainly spared in the disease*. To the extent that we know, it would be the reason by which this kind of approach has not been reported so far, even though VAcHT:Cre mice were generated in 2003⁹, and numerous proteins have been reported to be involved in motoneuron degeneration in ALS. On the other hand, HB9 expression in cells other than the spinal cord motoneurons could not be completely ruled out¹⁵. For instance, expression of HB9:Cre has also been observed in cells other than the ventral horn motoneurons¹⁶. In addition, a subset of Hb9-positive spinal cord interneurons, involved in rhythm generation during locomotor activity, has also been reported^{17,18}. Finally, larger dorsal horn neurons are apparent in the Hb9:Cre/Rosa26-FP mouse^{18,19}.

Given the state-of-the-art of currently available motoneuron-specific conditional knockouts, we alternatively performed intraspinal microinjections of *neuron*-specific lentiviral vectors directing the expression of a dominant negative for Sp1 (dnSp1) or shRNA_{p11} in 2-month-old SOD1^{G93A} mice (Fig. 4c, Fig. 9; lines 258-262; 459-477 and 1123-1130). Briefly, transduced lumbar motoneurons showed, in both cases, a significant reduction in p11-immunolabelling. Conclusively, the number of lumbar motoneurons in 4-month-old mice receiving LVV-dnSp1 or LVV-shRNA_{p11} was higher than in those injected with control lentivirus (LVV-shRNA_{lacz}). Thus, p11 depletion in lumbar motoneurons increased motoneuron survival in the SOD1^{G93A} mouse model of ALS, supporting a cell autonomous mechanism which involves Sp1/p11 as key players contributing to motoneuron degeneration.

Reviewer: In addition, the western blot illustrating knockdown of p11 in SOD mice looks like there could have been an issue with the transfer (Fig S5c). Is this the best example?

Response 8: This experiment has been repeated with similar results and western blot in Fig. S5c (**now:** Fig. S6b) has been replaced by another representative example. We have inspected all western blots shown in the paper, and some of them have been replaced by better examples (if possible) for the cropped and uncropped formats. In any case, replacement has not supported a change in the direction of results.

Altogether (**responses 6-8**), we think that new experiments and contributed data consistently confer additional confidence and strength to our results.

Reviewer: 4. What are the spinal and brainstem localization patterns of p11.

Response 9: New images from Allen Brain Atlas (in situ hybridization) and immunohistochemical analysis (Fig. 3a-f; Fig. 4a,b,e-g) are now contributed to describe expression patterns of p11 in lumbar spinal cord and brainstem in wild-type and neurodegenerative models (lines 185-188, 191-195, 240-241, and 252-255). As can be extracted from the study, motoneurons are the cell-type in both structures with higher levels of p11 expression.

Reviewer: The methods used for manipulating p11 expression allow no cell-type specificity so how certain can we be that any effects are cell autonomous due to p11 depletion in motoneurons?

Response 10: In the response to preceding concerns, we have detailed new experiments performed to gain in cell-type specificity (please see *responses 6-9*). The diversity of experimental approaches and new contributed data strongly support a cell autonomous effect of p11 depletion in delaying motoneuron degeneration in both experimental models. First, immunohistochemical characterization of p11 expression in wild type and neurodegenerative models (Fig. 3a-f; Fig. 4a,b,e-g). Second, reduction in p11 expression and neuroprotection observed in lumbar spinal motoneurons after chronic icv administration of siRNA_{p11} and, after intraspinal microinjections of neuron-specific LVV-dnSp1 or LVV-shRNA_{p11} (Fig. 4c, Fig. 9; Supplementary Fig. 6c). Third, experiments performed in primary cultures of spinal motoneurons all along the manuscript demonstrating that p11 depletion protects wild type and SOD1^{G93A} motoneurons against excitotoxic degeneration. However, as we recognize in the text in light of the new findings (lines 476-477), we cannot fully exclude the possibility that p11 upregulation in glial structures could also contribute in some degree to disease progression in the ALS model.

Reviewer: Can a loss of p11 and upregulation of TASK1 be demonstrated in motoneurons following siRNA (e.g., by histochemical approaches)?

Response 11: We have performed the suggested experiments and shown that p11-staining intensity in lumbar motoneurons 2 weeks after beginning of siRNA_{p11}-treatment was lower than that measured following cRNA-treatment (Supplementary Fig. 6c). However, we do not expect TASK1 upregulation following siRNA_{p11} (please see *response 1* to **reviewer #1**), but after Mit-A. Thus, administration of Mit-A for 2 weeks reduced and increased, respectively, p11 and TASK1-like immunolabelling in lumbar spinal motoneurons (Supplementary Fig 14c,d; lines 407-409).

Reviewer: 5. What are the common features of the proposed mechanisms underlying neurodegeneration associated with ALS (SOD model) and nerve injury (hypoglossal nucleus model) that would be targeted by p11?

Response 12: Schematic diagram of the proposed molecular pathways leading to neurodegeneration in both pathological conditions and a potential strategy to achieve neuroprotection are illustrated in figures 10 and 2k, respectively. Briefly, we have shown up-regulation of Sp1 and p11 in motoneurons preceding cell loss in both models. Sp1 induction seems to be a mechanism promoting p11 upregulation, which sequesters TASK1 to the ER, thus disrupting the normal recycling of this subunit to the surface. The consequent imbalance between TASK1 insertion and retrieval at the cell surface enhances IME, which, in turn, increases the opening probability of VSCCs and, GluRs in the presence of glutamate, thereby exacerbating Ca^{2+} entry into the cell. Ca^{2+} overload might contribute to neurodegeneration and, finally, to neuron death. However, although we have shown evidence for nitrosative stress as an upstream event promoting Sp1/p11 upregulation in motoneurons, we cannot exclude the possibility that other unidentified factors acting earlier and/or in parallel could also contribute to Sp1 and/or p11 upregulation and subsequent reduction in TASK1 expression at plasma membrane (Fig 10; lines 541-545; 1681-1683). Therefore, as we state in lines 168-171, “*p11 downregulation lowers motoneuron IME by promoting TASK1 insertion at the plasma membrane, attenuates Ca^{2+} influx through GluRs and VSCCs, and subsequently reduces neuronal vulnerability to excitotoxic degeneration (Fig. 2k).*”

Reviewer: is excitotoxicity a primary consideration in nerve injury that leads to use of Glu administration as a relevant test (Fig. S6f)? or, is that more likely a reflection of the loss of trophic support after cutting the nerve?

Response 13: Glutamate-mediated excitotoxicity has been considered to play an important role in the mechanisms of motor neuron death in ALS and after traumatic injury of a motor nerve²⁰⁻²⁸. As now we indicate in the manuscript (lines 123-127), “*It is well known that GluR overstimulation initiates a vicious cycle of intracellular Ca^{2+} deregulation, toxic Ca^{2+} overload, Ca^{2+} homeostasis disruption, and mitochondrial dysfunction that finally leads to the loss of neuronal function and cell death^{29,30}. To evoke this harmfulness mechanism, SMNs were exposed to a toxic dose of the excitatory neurotransmitter glutamate (Fig. 2a, Supplementary Fig. 3a,b).*” Furthermore, Vm variations affects Ca^{2+} entry via GluRs and VSCCs^{29,31,32}. Both, the ALS model and the resection of a motor nerve segment share excitotoxicity and hyperexcitability as

pathophysiological hallmarks^{25,27,28} (lines 174-179). Thus, our rationale was to use this test *in vitro* to evaluate the feasible neuroprotective actions of Sp1/p11 depletion against an excitotoxic insult.

Reviewer: 6. An excessive amount of information re. NO effects on cell excitability etc. is encompassed by the 2 sentences (6 lines of text) that must be accessed via Supplementary Note 8 and Figs. S7-S9). Some more effort should be expended in synthesizing and clarifying these data (e.g., distinct mechanisms of acute vs. chronic effects of NO, implications of dexamethasone results, and why all are necessary and relevant to the current work?)

Response 14: In the new version of the manuscript, Supplementary Note 8 has been included as main text (lines 271-304) with an explanation of why all are necessary and relevant (please also see **third paragraph** of the **response 6 to reviewer #1**). Previous data shown in Figs S7-S9 are now presented in Fig 5 and Suppl Figs 8 & 9. Acute and chronic effects of NO have been previously reported by others³. Dexamethasone treatment has been now presented as an experimental form to up-regulate p11 in the hypoglossal nucleus (Suppl Fig 1; lines 95-98).

Reviewer: In addition, is the *in vitro* Glu excitotoxicity model dependent on nitrosative stress (i.e., is it similar to the *in vivo* condition, as described)?

Response 15: NO from induced nNOS has been reported to mediate toxicity in primary cultures of SMNs deprived of neurotrophic factors^{33,34} and after FAS activation, which is potentiated by ALS-linked SOD1 mutations³⁵. However, nNOS was undetectable in control NTFs non-deprived SMNs³³. Given that in our experiments the excitotoxic stimulus was only present for 30 min, we do not expect that NO mediates glutamate-induced excitotoxicity in wild type SMNs. nNOS seems to be present in SMNs^{G93A} and, as Dorouchdi *et al* describe³⁶, “*inhibiting NOS rescued cultured motor neurons from excitotoxic death induced by adding glutamate to the culture medium*”. However, as we state above (please see **response 13**), we have used *in vitro* approaches to investigate the cell autonomous role of Sp1/p11 in neurodegeneration, as well as their potential relevance as molecular targets to protect against glutamate-induced excitotoxicity. For further reviews on NO involvement in excitotoxicity and neurodegeneration please see^{25,29,37-42}.

Reviewer: 7. The effect of Sp1dn or MitA on p11 expression seems to work constitutively, even without the enhancing effects of DETA/NO (e.g., Fig. S11, Fig. 4). This seems confusing, since the Sp1-GC box interaction was identified as an NO-stimulated effect. Please clarify.

Response 16: In the promoter activity experiments (**now:** Fig. 6), a baseline constitutive activity of the p11 promoter was always present. Our data indicate that DETA/NO stimulated/induced p11 promoter activity by, at least, a GC-box located at the -96 to -70 segment. That is why we named it as the “minimal NO-responsive element”, which resulted to be a Sp1-binding site (Fig. 6j). As we stated now in the text (lines 329-333) “*These findings did not discard other feasible NO-responsive and/or Sp1-binding sites upstream -96 to -70 segment. In this line, three additional Sp1 and a NO-responsive AP-1 binding sites upstream -96 have been identified^{43,44}. Thus, whether p11 promoter activity is regulated by other NO-responsive elements and/or by still unrevealed factors remain to be investigated.*” In no case we suggest that NO is required for p11 baseline expression. Indeed, only a pathological concentration of NO was able to induce/increase p11 promoter activity and p11 expression (Fig. 6).

Furthermore, we clarify in the text (lines 356-362) that “*Transient transfection of a plasmid directing expression of a truncated form of Sp1, which acts as a dominant negative (dnSp1), strongly reduced p11 in dNSC34s (Supplementary Fig. 11a,b). In the same line of evidence, a neuron-specific LVV directing the expression of dnSp1 (LVV-dnSp1) reduced mRNA_{p11}, strongly promoted functional expression of TASK-like channels, and almost fully blocked glutamate toxicity on SMNs^{wt} (Supplementary Fig. 11c-e). These data support that Sp1 controls constitutive expression of p11 in dNSC34s and SMNs^{wt}.*” Finally, in the discussion section (lines 532-535) we contributed published evidences supporting that NO could increase p11 promoter activity by promoting Sp1 expression and/or stimulating binding activity of Sp1. We hope that this concern has been now appropriately clarified.

Reviewer: 8. If MitA works in SMNs from SOD mice by increasing TASK expression, why is there no effect on Rn (Fig. 4g)?

Response 17: In previous Fig. 4g (**now:** Fig S13b), changes (in percent) in Vm and RN induced by Mit-A relative to vehicle-treated SMNs from Non-Tg (blue circles) and G93A (red diamonds) embryos are represented. As can be extracted from the plot, Mit-A induced a significant hyperpolarization of Vm and a reduction in RN in both SMN pools. We have now added new data showing that Mit-A promotes functional expression of TASK-like channels in SMNs^{G93A} (Fig S13c; line 393).

Reviewer: Were effects of MitA sensitive to Sp1dn?

Response 18: The reviewer's question would be relevant to be experimentally addressed in order to conclude about a Sp1 causative dependence of Mit-A effects on TASK1 expression. However, we suspect that Mit-A could regulate TASK1 by interfering not only with Sp1 (lines 363-368). Please see also **response 6 to reviewer #1**.

Reviewer: Also, the MitA effect on survival of G93A SMNs (~40%) was far less robust than on wild type SMNs (~100%) – compare Fig. 4e to Fig. 4j. Is that expected? why?

Response 19: From our point of view, it would be an expected result. First, Sp1/p11/TASK1 levels are different in SMNs^{G93A} and SMNs^{wt} and, therefore, the same dose of Mit-A might result in different levels of residual expression of the triad in both types of SMNs. Second, the multifactorial character of ALS; glutamate-induced excitotoxicity could be in part supported by mechanisms unaltered by Mit-A in the ALS model. Third, different genetic background of SMNs.

Reviewer: 9. The results with MitA by comparison to riluzole are important. Is it possible that combined treatment with both would have even greater effects (perhaps additive or synergistic)?

Response 20: We carried out this experiment in parallel with single drug Mit-A and Riluzole treatments (Suppl. Fig. 15g-m), but, combined treatment with both drugs did not lead to additive or synergistic effects (lines 430-431) (**Figure E**). Given the huge volume of results and figures, we decided to omit these results from the paper. However, if the reviewer considers them relevant enough, we will contribute these results as part of Suppl. Fig. 15g-m.

Figure E: Cumulative probability curve of survival for SOD1^{G93A} mice receiving the indicated treatments. The outcome after applying Long-Rank test, Kaplan-Meier analysis between Mit-A + Riluzole and vehicle conditions is stated in the plot. n = 14. Remaining conditions as in Supplementary Figure 15g-m.

Minor issues:

Reviewer: 1. For the nerve injury model, changes in intrinsic excitability due to p11 knockdown were attributed to TASK channels by examining effects of changes in pH. This same test was not performed on SMNs from SOD mice, where TASK contributions to changes in Em and Rn were not evaluated experimentally.

Response 21: We have performed this characterization in SMNs^{wt} treated with siRNA_{p11} (**now:** Fig. 1m), LVV-shRNA_{p11} (**now:** Suppl Fig 7b), LVV-dnSp1 (**now:** Suppl Fig 11d) and Mit-A (**now:** Suppl Fig 12d-g), with similar qualitative outcomes. In addition, we now contribute the effects of pH in SMNs^{G93A} in control condition and after Mit-A treatment (please see **response 17**). Therefore, we respectfully think that the additional study of siRNA_{p11}-induced effects suggested by the referee could be redundant. For compliance with ethical guidelines we have not performed the former experiment. However, if reviewer considers it relevant enough to support our main conclusions, we will perform it for a next version.

Reviewer: 2. What is a % change in membrane potential (e.g., see Fig. S8g, Fig. S9e)?

Response 22: These data are presented now in Fig 5j and Supplementary Fig. 9c. Changes in Vm are now expressed in mV. In the previous version, changes in Vm and RN were expressed as % to depict both data sets in the same plot.

Reviewer: 3. Fig. S10 and Supplementary Note 9: DETA/NO increased luciferase activity

expressed from the p11 promoter, but how can this be definitively attributed to enhanced transcription (as opposed to DETA/NO effects on translation or luciferase stability)?

Response 23: We understand the reviewer's concern. In this experiment and those presented in Figure 6, "*The human p11 promoter region (-1436 to +89) was cloned into the pGL4.10 luciferase Firefly reporter (pGL4.10-p11-Fir) vector and transfected into the differentiated motoneuron-like cell line NSC34 (dNSC34) together with the normalizer pGL4.74 luciferase Renilla (pGL4.74-TK-Ren) vector.*" as described in lines 314-317. Luciferase activity was represented as the ratio between Firefly/Renilla luciferase activities in each experiment (line 931). A similar protocol has been previously reported by others⁴⁵ to study NO effects on the PKG-I α promoter activity. Therefore, we think that these results may more likely reflect a change in luciferase activity rather than a change in its stability. Furthermore, luciferase activity was downstream the starting point for transcription (+89), and finally, DETA/NO increased mRNA_{p11} and p11 in NSC34s (Suppl Fig 10a), and via PKG-ROCK, increased mRNA_{p11} in SMNs^{wt} (Fig 6b,h,i). We think that, altogether, these data indicate that DETA/NO induces p11 promoter activity which results in the increase of p11 transcription. Anyway, we have substituted the terms "transcription activity" by "luciferase activity" in the text.

Reviewer: 4. For the supershift analysis in Fig. 3g, there appears to be quantitatively a much greater loss in the unbound fraction than is accounted for in the shifted fraction?

Response 24: We do not understand what the reviewer is referring to with this point. In Fig. 3g (**now:** Fig. 6g) just appears represented the complexes I and II. The unbound fraction (free probe) is not shown because it was deliberately left to run away for better showing of the complexes I and II.

Reviewer: 5. What was the rationale for switching the cell model for in vitro work from SMNs to dNSC34s?

Response 25: The motoneuron-like cell line NSC34 is easy to culture in order to obtain the great number of cells needed for this type of experiments, then reducing the number of animals used. Furthermore, lipofectamine was used to optimize plasmid transfection. In our hands, lipofectamine strongly affected viability of SMNs *per se*.

Reviewer: 6. Were the antibodies validated against knockout lines (e.g., TASK1)?

Response 26: Details of antibody validation are published on website of the corresponding commercial suppliers. Specific immunodetection of proteins by the antibodies used in this study has been indicated by demonstrating lack of immunoreactivity after application of antigen peptide, following protein knockdown, using knockout mice, or in controls cells not expressing respective proteins. All primary antibodies used in our work have been extensively used in numerous published papers. However, as the reviewer suggests, some controversy exists about specificity of most of commercially available anti-TASK1 antibodies since most of them exhibit reactivity in knockout animals. Thus, some considerations must be pointed out in this particular case.

For western blot analysis we have mainly used the antibody provided by Alomone Labs (Cat# APC-024, RRID:AB_2040132, lot. AN-08), and some experiments (not shown) were confirmed by using the antibody provided by Sigma-Aldrich (Cat# P0981, RRID:AB_260876). As TASK1 can be found in phosphorylated form⁴⁶, as homodimers, heterodimers (TASK1/3) and coupled with the adaptor proteins COPI, 14-3-3 and/or p11, it would not be surprising that several bands with different molecular weights could occasionally appear in western blotting. In our westerns, we have obtained a reproducible and well-defined band of ~53 KDa, absent in HEK293 (Suppl. Fig. 8g), the same as that previously reported to correspond to TASK1 protein⁴⁷. This immunoreactive band was strongly reduced⁴⁸ or fully absent⁴⁹ after treatment with a specific siRNA directed against TASK1, suggesting that this band correspond to TASK1. Furthermore, some key experiments were reinforced by qRT-PCR analysis, which confirmed changes in the same direction as those observed by western blotting.

Immunohistochemistry was performed using the antibody provided by Alomone for double immunolabelling with SMI32 to assure identification and analysis of lumbar motoneurons. In this way, a dramatic reduction in TASK1-immunolabelling in cells has been reported after treatment with an antisense deoxynucleotide probe targeting TASK1 and TASK3⁵⁰. Strikingly, immunohistochemistry performance with the lot. AN-08 from Alomone Labs clearly labelled SMNs^{wt} but immunolabelling was fully absent in SMNs^{task1^{-/-}} (**Figure F, Top**). On the contrary, an antibody obtained from Santa Cruz Biotechnology (Cat# sc-32065, RRID:AB_2280809) led to a staining pattern that was similar in samples from both genotypes (**Figure F, Bottom**). Whether specificity of the

Alomone's antibody is due to the lot. we used (AN-08), the experimental model, the developmental stage of motoneurons, or the different source of the knockout animals [Bayliss' lab (used here) vs Brickley's lab] should be clarified. Anyway, we now refer the signal due to the Alomone's antibody (lot. AN-08) as TASK1-like immunolabelling.

Figure F: Immunolabelling obtained after incubation with the two indicated anti-TASK1 antibodies (red) in $SMNs^{wt}$ and $SMNs^{task1-/-}$. (Top) The antibody from Alomone Labs (Cat# APC-024, RRID:AB_2040132, lot. AN-08) led to a strong immunostaining in $SMNs^{wt}$ which was fully absent in $SMNs^{task1-/-}$. Confocal setting was increased in gain for acquisition of TASK1-immunolabelling in $SMNs^{task1-/-}$ as compared with $SMNs^{wt}$ to strengthen background fluorescence. Nuclei were marked with DAPI (blue). (Bottom) Note that antibody from Santa Cruz Biotechnology (Cat# sc-32065, RRID:AB_2280809) led to similar immunostaining in the two samples. Both samples were processed in parallel. Solutions containing primary and secondary antibodies were common for both samples.

REFERENCES

1. Kofman, S., Perlia, C.P. & Economou, S.G. Mithramycin in the treatment of metastatic Ewing's sarcoma. *Cancer* **31**, 889-893 (1973).
2. Liu, B., Wang, S., Brenner, M., Paton, J.F. & Kasparov, S. Enhancement of cell-specific transgene expression from a Tet-Off regulatory system using a transcriptional amplification strategy in the rat brain. *J Gene Med* **10**, 583-592 (2008).
3. Gonzalez-Forero, D., *et al.* Inhibition of resting potassium conductances by long-term activation of the NO/cGMP/protein kinase G pathway: A new mechanism regulating neuronal excitability. *Journal of Neuroscience* **27**, 6302-6312 (2007).
4. Nakajima, H., *et al.* A rapid, targeted, neuron-selective, in vivo knockdown following a single intracerebroventricular injection of a novel chemically modified siRNA in the adult rat brain. *J Biotechnol* **157**, 326-333 (2012).
5. Gupta, A.K., Eshraghi, Y., Gliniak, C. & Gosain, A.K. Nonviral transfection of mouse calvarial organ in vitro using Accell-modified siRNA. *Plast Reconstr Surg* **125**, 494-501 (2010).
6. Larsen, H.O., Roug, A.S., Nielsen, K., Sondergaard, C.S. & Hokland, P. Nonviral transfection of leukemic primary cells and cells lines by siRNA-a direct comparison between Nucleofection and Accell delivery. *Exp Hematol* **39**, 1081-1089 (2011).
7. Gherardini, L., Bardi, G., Gennaro, M. & Pizzorusso, T. Novel siRNA delivery strategy: a new "strand" in CNS translational medicine? *Cell Mol Life Sci* **71**, 1-20 (2014).
8. Misawa, H., *et al.* Reappraisal of VAcHT-Cre: Preference in slow motor neurons innervating type I or IIa muscle fibers. *Genesis* **54**, 568-572 (2016).
9. Misawa, H., *et al.* VAcHT-Cre. Fast and VAcHT-Cre.Slow: postnatal expression of Cre recombinase in somatomotor neurons with different onset. *Genesis* **37**, 44-50 (2003).
10. Shaw, P.J. & Eggett, C.J. Molecular factors underlying selective vulnerability of motor neurons to neurodegeneration in amyotrophic lateral sclerosis. *J Neurol* **247 Suppl 1**, I17-27 (2000).
11. Pun, S., Santos, A.F., Saxena, S., Xu, L. & Caroni, P. Selective vulnerability and pruning of phasic motoneuron axons in motoneuron disease alleviated by CNTF. *Nat Neurosci* **9**, 408-419 (2006).
12. Saxena, S., *et al.* Neuroprotection through excitability and mTOR required in ALS motoneurons to delay disease and extend survival. *Neuron* **80**, 80-96 (2013).
13. Dengler, R., *et al.* Amyotrophic lateral sclerosis: macro-EMG and twitch forces of single motor units. *Muscle Nerve* **13**, 545-550 (1990).
14. Hegedus, J., Putman, C.T. & Gordon, T. Time course of preferential motor unit loss in the SOD1 G93A mouse model of amyotrophic lateral sclerosis. *Neurobiol Dis* **28**, 154-164 (2007).
15. Wu, L.S., Cheng, W.C. & Shen, C.K. Targeted depletion of TDP-43 expression in the spinal cord motor neurons leads to the development of amyotrophic lateral sclerosis-like phenotypes in mice. *J Biol Chem* **287**, 27335-27344 (2012).
16. Arber, S., *et al.* Requirement for the homeobox gene Hb9 in the consolidation of motor neuron identity. *Neuron* **23**, 659-674 (1999).
17. Wilson, J.M., *et al.* Conditional rhythmicity of ventral spinal interneurons defined by expression of the Hb9 homeodomain protein. *J Neurosci* **25**, 5710-5719 (2005).
18. Caldeira, V., Dougherty, K.J., Borgius, L. & Kiehn, O. Spinal Hb9::Cre-derived excitatory interneurons contribute to rhythm generation in the mouse. *Sci Rep* **7**, 41369 (2017).
19. Lee, S.K., Jurata, L.W., Funahashi, J., Ruiz, E.C. & Pfaff, S.L. Analysis of embryonic motoneuron gene regulation: derepression of general activators function in concert with enhancer factors. *Development* **131**, 3295-3306 (2004).
20. Cleveland, D.W. & Rothstein, J.D. From Charcot to Lou Gehrig: deciphering selective motor neuron death in ALS. *Nat Rev Neurosci* **2**, 806-819 (2001).

21. Wood, L.K. & Langford, S.J. Motor neuron disease: a chemical perspective. *J Med Chem* **57**, 6316-6331 (2014).
22. Van Den Bosch, L., Van Damme, P., Bogaert, E. & Robberecht, W. The role of excitotoxicity in the pathogenesis of amyotrophic lateral sclerosis. *Biochim Biophys Acta* **1762**, 1068-1082 (2006).
23. Bogaert, E., d'Ydewalle, C. & Van Den Bosch, L. Amyotrophic lateral sclerosis and excitotoxicity: from pathological mechanism to therapeutic target. *CNS Neurol Disord Drug Targets* **9**, 297-304 (2010).
24. Foran, E. & Trotti, D. Glutamate transporters and the excitotoxic path to motor neuron degeneration in amyotrophic lateral sclerosis. *Antioxid Redox Signal* **11**, 1587-1602 (2009).
25. Moreno-Lopez, B., Sunico, C.R. & Gonzalez-Forero, D. NO orchestrates the loss of synaptic boutons from adult "sick" motoneurons: modeling a molecular mechanism. *Mol Neurobiol* **43**, 41-66 (2011).
26. Gonzalez-Forero, D. & Moreno-Lopez, B. Retrograde response in axotomized motoneurons: nitric oxide as a key player in triggering reversion toward a dedifferentiated phenotype. *Neuroscience* **283**, 138-165 (2014).
27. Do-Ha, D., Buskila, Y. & Ooi, L. Impairments in Motor Neurons, Interneurons and Astrocytes Contribute to Hyperexcitability in ALS: Underlying Mechanisms and Paths to Therapy. *Mol Neurobiol* (2017).
28. King, A.E., Woodhouse, A., Kirkcaldie, M.T. & Vickers, J.C. Excitotoxicity in ALS: Overstimulation, or overreaction? *Exp Neurol* **275 Pt 1**, 162-171 (2016).
29. Lau, A. & Tymianski, M. Glutamate receptors, neurotoxicity and neurodegeneration. *Pflugers Arch* **460**, 525-542 (2010).
30. Lu, Y.M., Yin, H.Z., Chiang, J. & Weiss, J.H. Ca²⁺-permeable AMPA/kainate and NMDA channels: high rate of Ca²⁺ influx underlies potent induction of injury. *J Neurosci* **16**, 5457-5465 (1996).
31. Simms, B.A. & Zamponi, G.W. Neuronal voltage-gated calcium channels: structure, function, and dysfunction. *Neuron* **82**, 24-45 (2014).
32. Mayer, M.L., Westbrook, G.L. & Guthrie, P.B. Voltage-dependent block by Mg²⁺ of NMDA responses in spinal cord neurones. *Nature* **309**, 261-263 (1984).
33. Estevez, A.G., *et al.* Nitric oxide and superoxide contribute to motor neuron apoptosis induced by trophic factor deprivation. *J Neurosci* **18**, 923-931 (1998).
34. Estevez, A.G., *et al.* Role of endogenous nitric oxide and peroxynitrite formation in the survival and death of motor neurons in culture. *Prog Brain Res* **118**, 269-280 (1998).
35. Raoul, C., *et al.* Motoneuron death triggered by a specific pathway downstream of Fas. potentiation by ALS-linked SOD1 mutations. *Neuron* **35**, 1067-1083 (2002).
36. Doroudchi, M.M., Minotti, S., Figlewicz, D.A. & Durham, H.D. Nitrotyrosination contributes minimally to toxicity of mutant SOD1 associated with ALS. *Neuroreport* **12**, 1239-1243 (2001).
37. Gu, Z., Nakamura, T. & Lipton, S.A. Redox reactions induced by nitrosative stress mediate protein misfolding and mitochondrial dysfunction in neurodegenerative diseases. *Mol Neurobiol* **41**, 55-72 (2010).
38. Calabrese, V., Bates, T.E. & Stella, A.M. NO synthase and NO-dependent signal pathways in brain aging and neurodegenerative disorders: the role of oxidant/antioxidant balance. *Neurochem Res* **25**, 1315-1341 (2000).
39. Yuste, J.E., Tarragon, E., Campuzano, C.M. & Ros-Bernal, F. Implications of glial nitric oxide in neurodegenerative diseases. *Front Cell Neurosci* **9**, 322 (2015).
40. Altinoz, M.A. & Elmaci, I. Targeting nitric oxide and NMDA receptor-associated pathways in treatment of high grade glial tumors. Hypotheses for nitro-memantine and nitrones. *Nitric Oxide* **79**, 68-83 (2018).

41. Fujikawa, D.G. The role of excitotoxic programmed necrosis in acute brain injury. *Comput Struct Biotechnol J* **13**, 212-221 (2015).
42. Jia, M., Njapo, S.A., Rastogi, V. & Hedna, V.S. Taming glutamate excitotoxicity: strategic pathway modulation for neuroprotection. *CNS Drugs* **29**, 153-162 (2015).
43. Pawliczak, R., *et al.* p11 expression in human bronchial epithelial cells is increased by nitric oxide in a cGMP-dependent pathway involving protein kinase G activation. *J Biol Chem* **276**, 44613-44621 (2001).
44. Huang, X., *et al.* Characterization of the human p11 promoter sequence. *Gene* **310**, 133-142 (2003).
45. Sellak, H., *et al.* Sp1 transcription factor as a molecular target for nitric oxide-- and cyclic nucleotide--mediated suppression of cGMP-dependent protein kinase-Ialpha expression in vascular smooth muscle cells. *Circ Res* **90**, 405-412 (2002).
46. Kilisch, M., Lytovchenko, O., Arakel, E.C., Bertinetti, D. & Schwappach, B. A dual phosphorylation switch controls 14-3-3-dependent cell surface expression of TASK-1. *J Cell Sci* **129**, 831-842 (2016).
47. Inoue, M., Harada, K., Matsuoka, H., Sata, T. & Warashina, A. Inhibition of TASK1-like channels by muscarinic receptor stimulation in rat adrenal medullary cells. *J Neurochem* **106**, 1804-1814 (2008).
48. Leithner, K., *et al.* TASK-1 Regulates Apoptosis and Proliferation in a Subset of Non-Small Cell Lung Cancers. *PLoS One* **11**, e0157453 (2016).
49. Shinoda, K., *et al.* Genetic and functional characterization of clonally derived adult human brown adipocytes. *Nat Med* **21**, 389-394 (2015).
50. Hartness, M.E., *et al.* Combined antisense and pharmacological approaches implicate hTASK as an airway O₂ sensing K⁽⁺⁾ channel. *J Biol Chem* **276**, 26499-26508 (2001).

CHANGES IN FIGURES ARE AS FOLLOWS:

- New experiments and data are presented in Figures:

Fig 1h; Fig 3a,b,c,e,f,h; Fig 4a,b,c,f,g,i; Fig 7b; Fig 8e; Fig 9; Suppl Fig 1; Suppl Fig 6b,c; Suppl Fig 13c; Suppl Fig 14c,d

- Previous figures have been rearranged as follows:

Previous version	Current version	Previous version	Current version
Fig 1c,d,e	Fig 1e,j,k,l	Suppl Fig 5b-i	Suppl Fig 6a,b,d-i
Fig 1f-j	Fig 2a,b,i-k	Suppl Fig 6a-c,g-i	Fig 4c-e,h,i
Fig 2a, h-k	Fig 3d,i-l	Suppl Fig 6d-f	Suppl Fig 7a-c
Fig 2b,c,d,e-g	Suppl Fig 5b,c,d,h,I	Suppl Fig 7a	Fig 5a
Fig 3	Fig 6	Suppl Fig 7b-h	Suppl Fig 8a-g
Fig 4a-e	Suppl Fig 12a-c,k,l	Suppl Fig 8a-d	Fig 5b-e
Fig 4f,k,l	Fig 7c-e	Suppl Fig 8e-g	Suppl Fig 9a-c
Fig 4g-j	Suppl Fig 13b,d,h,I	Suppl Fig 9a,d,e,f,g	Fig 5f,g,h,i,j
Fig 5a-c,g-k	Fig 7f-h,i-m	Suppl Fig 9b,c,e,h,i	Suppl Fig 9d -h
Fig 5d-f	Suppl Fig 15a-c	Suppl Fig 10	Suppl Fig 10
Fig 6	Fig 10	Suppl Fig 11	Suppl Fig 11
Suppl Fig 1a-e,g	Fig 1b,c,d,f,g,i,m	Suppl Fig 12	Suppl Fig 12d-g
Suppl Fig 1f	Suppl Fig 2	Suppl Fig 13	Suppl Fig 12h-j
Suppl Fig 2	Suppl Fig 3	Suppl Fig 14a	Fig 7a
Suppl Fig 3a,d,e,g-j	Fig 2c-h	Suppl Fig 14b	Suppl Fig 13a
Suppl Fig 3b,c,f,k	Suppl Fig 4a-d	Suppl Fig 15	Suppl Fig 13e-g
Suppl Fig 4a	Suppl Fig 5a	Suppl Fig 16	Suppl Fig 14a,b
Suppl Fig 4b-d	Suppl Fig 5e-g	Suppl Fig 17,18	Suppl Fig 15
Suppl Fig 5a	Fig 3g	Suppl Fig 19,20	Fig 8a-d,f

Reviewers' comments:

Reviewer #2 (Remarks to the Author):

I commend the authors on the work they did refactoring this manuscript. It is now much easier to read and understand, and is therefore more convincing.

I am satisfied with the answers provided by the author to mine and the other reviewer's concerns.

Some very minor points could be addressed:

- Although I don't doubt the results of the "TASK-dependent changes" experiments (i.e. changes in RMP and RN with changes in pH), I would have preferred seeing plots of RMP / RN vs. pH and to see the change of slope associated with the different treatments rather than a single delta data point. I don't know what such figure would look like, and if the number of points/lines required would make it more confusing, so I'm leaving that as a suggestion for the authors to decide on their own what is the best representation.

- The term "I_h" for holding current could be easily misinterpreted since in my mind it refers to the hyperpolarization-activated cationic current mediated by HCN channels. Maybe "I_{holding}" would avoid any ambiguity

Reviewer #3 (Remarks to the Author):

This revised manuscript has been substantially improved, both by the reorganization of the presentation and by the inclusion of new data or clarifying text. It remains an exciting piece of work that provides evidence for a channel-dependent mechanism that can account for an intrinsic hyper-excitability of motor neurons that can contribute to degeneration in both an ALS or nerve injury model; it also includes proof-of-principle pharmacological experiments building on those mechanisms that show some efficacy in the mouse models.

Some remaining issues should be addressed:

1. Some additional data are presented to validate the knockdown approaches, and provide evidence for actions on motoneurons. This includes immunohistochemistry for p11 following 4th ventricle injections of siRNA (in Fig. S6c) and after brainstem or spinal injections of LVV constructs (Fig. 4i, Fig. 9c-f). These results are not particularly compelling, especially by comparison to changes shown in mRNA levels or by Western blot (e.g., compare Fig. S6b and S6c, or Fig. 4i to 4h). In Fig. S6c, why was the histochemistry performed at a different time point (2 wks instead of 8 wks, as for mRNA)?

One also wonders exactly how the immunohistochemical data were compared across different mice. Even with tissue processed in parallel, as stated in the Methods, there can be variations in how well the mice were perfused, the background fluorescence, etc., and all of these could lead to minor differences in fluorescence intensity such as those presented. Please provide additional details on this analysis.

For the LVV injection experiments, perhaps one can find infected and uninfected motor neurons to compare in the same field from the same section (e.g., for a new Fig. 9d,e).

2. The text describes the LVV-mediated knockdown in Fig. S7a as "efficient" (l. 262), but this looks to be ~30% at best. This language should be tempered. The effect on electrical properties and survival seem much more robust.

3. At other points, the reader is not given any sense of the size of effects in the text and this can

feel a bit misleading. For example, from the text it is not clear that the effect of Mit-A (300, ip) is so tiny (see Fig. 8f); the text just states that this higher dose of also showed neuroprotective effects (l. 450). There are other examples where the descriptions could be more circumspect.

4. In Fig. 3j (and elsewhere), what is the definition of "symptoms" and how was the onset determined?

5. Please change % to mV for presenting effects on membrane potential (in Fig. S12c,d,f; Fig. S13b; and anywhere else.)

Dear Reviewers,

We would like to thank you for your helpful comments on the manuscript. We are sincerely grateful to you for the time spent reading, commenting, and suggesting new experiments, which have greatly improved the quality of the work. We have tried our best to carefully consider and respond to the questions raised. Detailed answers to all the reviewers' comments are provided below.

Response to reviewers

REVIEWER #2:

I commend the authors on the work they did refactoring this manuscript. It is now much easier to read and understand, and is therefore more convincing.

I am satisfied with the answers provided by the author to mine and the other reviewer's concerns.

Some very minor points could be addressed:

Reviewer: Although I don't doubt the results of the "TASK-dependent changes" experiments (i.e. changes in RMP and RN with changes in pH), I would have preferred seeing plots of RMP / RN vs. pH and to see the change of slope associated with the different treatments rather than a single delta data point. I don't know what such figure would look like, and if the number of points/lines required would make it more confusing, so I'm leaving that as a suggestion for the authors to decide on their own what is the best representation.

Response 1: Reviewer's suggestion refers to plots like those presented as examples in SF12d,e. We have decided to represent these plots as delta mainly for clarity and for space minimization. In addition, we think that this representation form makes more intuitive and easier-to-see for readers the effect of treatments on "TASK-dependent changes".

Reviewer: The term "I_h" for holding current could be easily misinterpreted since in my mind it refers to the hyperpolarization-activated cationic current mediated by HCN channels. Maybe "I_{holding}" would avoid any ambiguity.

Response 2: We agree with reviewer; we have modified the term as suggested.

REVIEWER #3:

This revised manuscript has been substantially improved, both by the reorganization of the presentation and by the inclusion of new data or clarifying text. It remains an exciting piece of work that provides evidence for a channel-dependent mechanism that can account for an intrinsic hyper-excitability of motor neurons that can contribute to degeneration in both an ALS or nerve injury model; it also includes proof-of-principle pharmacological experiments building on those mechanisms that show some efficacy in the mouse models.

Some remaining issues should be addressed:

Reviewer: 1. Some additional data are presented to validate the knockdown approaches, and provide evidence for actions on motoneurons. This includes immunohistochemistry for p11 following 4th ventricle injections of siRNA (in Fig. S6c) and after brainstem or spinal injections of LVV constructs (Fig. 4i, Fig. 9c-f). These results are not particularly compelling, especially by comparison to changes shown in mRNA levels or by Western blot (e.g., compare Fig. S6b and S6c, or Fig. 4i to 4h).

Response 1: We understand reviewer's concern. In this context, different sensitivity of techniques can account for differential findings found by qRT-PCR, western blot and immunohistochemistry, which can be emphasized by difference in the specimens analyzed with each technique. For example, for qRT-PCR and western blot, analyzed samples consisted of microdissected HNs or lumbar spinal cords including other cells types besides motoneurons that may be also affected by LVVs or siRNAs. In contrast, by immunohistochemistry we exclusively quantified p11 expression within motoneuron perikarya, thus excluding other feasible sources of p11 (among them neuropil) that may be more affected by siRNA or LVVs. In addition, for comparison of Fig 4h and 4i, it should be taken into account that in 4i p11 is upregulated after lesion, while in 4h LVVs were injected in intact/uninjured mice, that is comparison of LVV effects between 4h and 4i is influenced by variables other than LVV injection. Furthermore, western blot shown in 4h was just exposed avoiding over-exposition of neocortex signal taken as an internal control, then it cannot be concluded from this experiment that protein for p11 was absent after LVV injection. When comparing S6b and S6c it should be taking into account the aforementioned arguments as well the different duration of siRNA administration (2 vs 8 weeks).

In Fig. S6c, why was the histochemistry performed at a different time point (2 wks instead of 8 wks, as for mRNA)?

Response 2: This experiment aimed to get evidence that siRNA administration into the 4th ventricle actually reaches lumbar motoneurons. With the purpose to minimize unnecessary animal suffering, according to ethical commitments, mice were perfused after 2 weeks of treatments, a time point at which maximal reduction of mRNA_{p11} was already reached in the lumbar spinal cord (S6a). Fig 3h shows that treatment with siRNA for 8 weeks drastically reduced p11 immunolabelling in the lumbar spinal cord of 4-month-old SOD1^{G93A} mice. Note the marked difference in p11 immunoreactivity existing between the two vacuolated motoneurons of animals treated either with cRNA or with siRNA_{p11} (Fig. 3h).

One also wonders exactly how the immunohistochemical data were compared across different mice. Even with tissue processed in parallel, as stated in the Methods, there can be variations in how well the mice were perfused, the background fluorescence, etc., and all of these could lead to minor differences in fluorescence intensity such as those presented. Please provide additional details on this analysis.

Response 3: As stated in Methods (lines 820-832), animals were processed in parallel throughout the experiment. They were sacrificed the same day sharing perfusion solutions. Analysis was performed from at least three different animals per condition, and changes in immunoreactivity (relative to controls) were directionally similar in all LVV-shRNA_{p11} or LVV-dnSp1 treated animals. Therefore, we believe that probability that these outcomes are the result of chance is low. Furthermore, treatments were blinded to researchers who analyzed immunolabeling and background fluorescence for each slice was subtracted. As an internal control in Fig. 9c-e, p11 labelling in glial-like cells was similar in the three conditions in spite that motoneurons showed different intensity of fluorescence.

For the LVV injection experiments, perhaps one can find infected and uninfected motor neurons to compare in the same field from the same section (e.g., for a new Fig. 9d,e).

Response 4: We respectfully do not fully share reviewer's argument on this concern. After 2 months of neuron-specific LVV injection, a GFP-positive motoneuron can be considered as transduced, however, the reciprocal could not necessarily apply. Stability of mRNA_{GFP}, GFP protein and/or level of GFP expression could contribute to false negatives. Therefore, comparison that reviewer proposes could underestimate LVV effects. This is the reason why we have chosen to show non-infected glial-like cells.

Reviewer: 2. The text describes the LVV-mediated knockdown in Fig. S7a as "efficient" (l. 262), but this looks to be ~30% at best. This language should be tempered. The effect on electrical properties and survival seem much more robust.

Response 5: The term "Efficiently" has been now omitted from the text (lines 228, 262 and 406) as suggested.

Reviewer: 3. At other points, the reader is not given any sense of the size of effects in the text and this can feel a bit misleading. For example, from the text it is not clear that the effect of Mit-A (300, ip) is so tiny (see Fig. 8f); the text just states that this higher dose of also showed neuroprotective effects (l. 450). There are other examples where the descriptions could be more circumspect.

Response 6: Now we describe Mit-A (300, ip) effects as "slight but significant" (line 449-450). We have looked for this type of descriptions and we have tried to be more circumspect if possible (lines 216, 253-254, 285-287, 407-408, 411 and 441).

Reviewer: 4. In Fig. 3j (and elsewhere), what is the definition of "symptoms" and how was the onset determined?

Response 7: We described this issue in the Methods section (lines 1011-1014). "As starting point to construct the cumulative probability curve of symptoms onset, we used the first day in which a mouse showed motor function deficits on the rotarod test ($\geq 10\%$

reduction in the time to fall) followed by progressive deterioration in performance in the next sessions.”

Reviewer: 5. Please change % to mV for presenting effects on membrane potential (in Fig. S12c,d,f; Fig. S13b; and anywhere else.)

Response 8: Membrane potential changes have been presented in mV in all figures following reviewer’s suggestion.

No further comments...